_Article_

# The DDHD2-STXBP1 interaction mediates long-term memory via generation of saturated free fatty acids

Isaac O Akefe [1,2,14], Saber H Saber [1,3,14], Benjamin Matthews [1,14], Bharat G Venkatesh [1], Rachel S Gormal [1], Daniel G Blackmore[1], Suzy Alexander[1], Emma Sierecki [4,5], Yann Gambin[4,5], Jesus Bertran-Gonzalez[6], Nicolas Vitale [7], Yann Humeau[8], Arnaud Gaudin [1], Sevannah A Ellis [1], Alysee A Michaels[9,10], Mingshan Xue [9,10,11], Benjamin Cravatt[12], Merja Joensuu [1,3 ✉], Tristan P Wallis [1 ✉] & Frédéric A Meunier [1,13 ✉]

## Abstract

**The phospholipid and free fatty acid (FFA) composition of neuronal membranes plays a crucial role in learning and memory, but the mechanisms through which neuronal activity affects the brain's lipid landscape remain largely unexplored. The levels of saturated FFAs, particularly of myristic acid (C14:0), strongly increase during neuronal stimulation and memory acquisition, suggesting the involvement of phospholipase A1 (PLA1) activity in synaptic plasticity. Here, we show that genetic ablation of the PLA1 isoform DDHD2 in mice dramatically reduces saturated FFA responses to memory acquisition across the brain. Furthermore, DDHD2 loss also decreases memory performance in reward-based learning and spatial memory models prior to the development of neuromuscular deficits that mirror human spastic paraplegia. Via pulldown-mass spectrometry analyses, we find that DDHD2 binds to the key synaptic protein STXBP1. Using STXBP1/2 knockout neurosecretory cells and a haploinsufficient STXBP1[+/−] mouse model of human early infantile encephalopathy associated with intellectual disability and motor dysfunction, we show that STXBP1 controls targeting of DDHD2 to the plasma membrane and generation of saturated FFAs in the brain. These findings suggest key roles for DDHD2 and STXBP1 in lipid metabolism and in the processes of synaptic plasticity, learning, and memory.**

**Keywords** Lipids; Phospholipase A1; Free Fatty Acids; Myristic Acid; Learning and Memory
**Subject Categories** Membranes & Trafficking; Metabolism; Neuroscience

## Introduction

Activity-dependent changes to the brain's lipid landscape are widely believed to play a role in neuronal activity, learning, and memory (Alrayes et al, 2015; Barber and Raben, 2019; Bruce et al, 2017; Carta et al, 2014; Zhu et al, 2016). Uncovering the mechanism by which molecular and cellular interactions between lipids and specific proteins at the synapse contribute to memory formation and consolidation remains crucial in establishing a deeper understanding of synaptic plasticity and bases for neurological diseases (Wenk and De Camilli, 2004). In addition to acting as building blocks for synaptic vesicles and neuronal plasma membranes, brain lipids are involved in diverse neuronal functions including scaffolding (Lee et al, 2021), neurodevelopment (Zhu et al, 2016), signalling (Falomir-Lockhart et al, 2019), neuroinflammation (Giacobbe et al, 2020), neurometabolism (Bruce et al, 2017), and cognition (de Mendoza and Pilon, 2019; Derbyshire, 2018; Egawa et al, 2016; Joffre, 2019; Snowden et al, 2017; Tracey et al, 2018). Dynamic regulation of the brain-lipid landscape via the coordinated activity of phospholipases and other enzymes is crucial for mediating appropriate membrane curvature, surface chemistry, fluidity and fusogenicity of the phospholipid membrane bilayer during vesicular trafficking, exocytosis, endocytosis and synaptic plasticity underpinning memory formation (Akefe, 2022; Akefe et al, 2023; Alrayes et al, 2015; Barber and Raben, 2019; Bruce et al, 2017; Carta et al, 2014; Zhu et al, 2016). The importance of phospholipases in this process is exemplified in hereditary spastic paraplegia (HSP; Strümpell-Lorrain disease) in which a mutation in the _DDHD2_ gene of phospholipase A1 (PLA1) is associated with neuromuscular and cognitive dysfunction (Akefe, 2022; Alrayes et al, 2015; Inloes et al, 2014; Joensuu et al, 2020b; Murala et al, 2021), autism (Matoba et al, 2020), schizophrenia (Park et al, 2021),

[1]Clem Jones Centre for Ageing Dementia Research, Queensland Brain Institute, The University of Queensland, St Lucia, QLD 4072, Australia. [2]Academy for Medical Education, Medical School, The University of Queensland, 288 Herston Road, 4006 Brisbane, QLD, Australia. [3]Australian Institute for Bioengineering and Nanotechnology, The University of Queensland, Brisbane, St Lucia, QLD 4072, Australia. [4]School of Medical Science, University of New South Wales, Randwick, NSW 2052, Australia. [5]EMBL Australia, Single Molecule Node, University of New South Wales, Sydney 2052, Australia. [6]Decision Neuroscience Lab, School of Psychology, UNSW Sydney, Sydney, Australia. [7]Institut des Neurosciences Cellulaires et Intégratives, UPR-3212 CNRS - Université de Strasbourg, Strasbourg, France. [8]Interdisciplinary Institute for Neuroscience, CNRS UMR 5297, Université de Bordeaux, Bordeaux, France. [9]Department of Neuroscience, Baylor College of Medicine, Houston, TX, USA. [10]The Cain Foundation Laboratories, Jan and Dan Duncan Neurological Research Institute at Texas Children's Hospital, Houston, TX, USA. [11]Department of Molecular and Human Genetics, Baylor College of Medicine, Houston, TX, USA. [12]The Department of Chemistry and The Skaggs Institute for Chemical Biology, The Scripps Research Institute, La Jolla, CA, USA. [13]The School of Biomedical Sciences, The University of Queensland, St Lucia, QLD 4072, Australia. [14]These authors contributed equally: Isaac O Akefe, Saber H Saber, Benjamin Matthews. ✉E-mail: m.joensuu@uq.edu.au; t.wallis@uq.edu.au; f.meunier@uq.edu.au

intellectual disability (Alrayes et al, 2015), and mental retardation (Alrayes et al, 2015; Darios et al, 2020; Inloes et al, 2014; Joensuu et al, 2020b). Although this establishes a potential causal link between PLA1 and cognitive function, the mechanisms by which phospholipases affect the lipid landscape to mediate synaptic plasticity that is conducive to memory acquisition in the brain are unknown.

Phospholipids and their free fatty acid (FFA) metabolites (Kihara et al, 2014) have been shown to bind key exocytic proteins such as syntaxin-1A (STX1A), and Munc18-1 (STXBP1) (Jang et al, 2012; Latham et al, 2007; Lee et al, 2004), to regulate the rate of synaptic vesicle exo- and endocytosis. Phospholipids and FFAs also function alone, as signalling molecules, or as reservoirs of lipid messengers which in turn, regulate and interact with several signalling cascades, contributing to synaptic structure and function (Montaner et al, 2018; Paoletti et al, 2016; Wang et al, 2016). Polyunsaturated fatty acids (PUFAs) which contain more than one double bond in their acyl tail, such as arachidonic acid (AA; C20:4) and docosahexaenoic acid (DHA; C22:6), play a vital role in soluble N-ethylmaleimide-sensitive factor attachment protein receptor (SNARE)-mediated synaptic transmission (Sidhu et al, 2016). Specifically, the non-covalent interaction of AA has been shown to allow STX1A to form the SNARE complex (Connell et al, 2007). Furthermore, FFAs are capable of regulating neurotransmission by modulating membrane curvature, fluidity, and fusogenicity, consequently making the membrane architecture more amenable to vesicle fusion (Graham and Kozlov, 2010). Cholesterol, diacylglycerol (DAG), and phosphatidic acid (PA) are cone-shaped lipids that can also induce negative membrane curvature to promote membrane fusion (Ammar et al, 2013).

During neurological processes such as neurotransmitter release, long-term potentiation and subsequent synaptic plasticity, the targeted activity of phospholipases plays a key role by catalysing the hydrolysis of phospholipids to generate FFAs thereby changing the local environment of the targeted bilayer (Akefe et al, 2023; Joensuu et al, 2020b; Puchkov and Haucke, 2013) and consequently impacting neurotransmission (Goldschmidt et al, 2016). Phospholipase A2 (PLA2) hydrolyses the fatty acyl chain on the sn-2 position of canonical phospholipids to generate unsaturated FFAs such as AA, which further initiate a cascade of bioactive signalling molecules in many biological systems including neuroinflammation, neurotransmitter release and long-term potentiation (Higgs and Glomset, 1996; Inloes et al, 2014).

With the advent of unbiased lipidomic approaches, the involvement of PLA1 family members was suggested as secretagogue stimulation of neuroexocytosis was shown to trigger a major increase in primarily saturated FFAs in cultured neurons and neurosecretory cells (Narayana et al, 2015). Further, the acquisition/consolidation of long-term fear memory in rats also correlates with strong increases in saturated FFAs (Wallis et al, 2021). These results pointed to a role in memory acquisition of PLA1 family members via the production of saturated FFAs and dynamic alteration of the phospholipid landscape in the brain. The importance of PLA1 to neuronal function is further substantiated in genetic disorders in which the prominent DDHD2 isoform of PLA1 is absent or mutated. While DDHD2 is highly expressed in the central nervous system, the understanding of its role in neuronal function and plasticity is limited (Alrayes et al, 2015; Inoue et al, 2012; Joensuu et al, 2020b).

To investigate the function of DDHD2 in mediating synaptic plasticity and memory formation, we used a *DDHD2* knockout mouse model of hereditary spastic paraplegia (HSP) (Inloes et al, 2014). We employed a number of neurobehavioural paradigms to track the onset and progression of both neuromotor and cognitive decline throughout the lifespan of the *DDHD2*$^{-/-}$ mice. These longitudinal experiments were conducted in 3mo young mice that were largely asymptomatic, and again at 12mo, when the animals exhibit typical symptoms of neuromuscular dysfunction. We assessed the FFA changes in response to an instrumental conditioning paradigm that depends on reward-based associative memory. We used a rapid isotope-based multiplexing FFA analysis pipeline (FFAST (Narayana et al, 2015; Wallis et al, 2021)) to quantify, in conditioned animals, the responses of 19 FFAs across 5 brain regions involved with cognitive and neuromotor functions. We found that reward-based instrumental conditioning can drive brain region-specific changes in the brain lipidome, characterised by increases in saturated FFAs, analogous to the response reported for a fear-based memory paradigm (Wallis et al, 2021). In contrast, *DDHD2* knockout mice exhibited a significant reduction in the response of saturated FFAs (especially myristic acid) even prior to the onset of memory impairment. This suggests that saturated FFA responses driven by DDHD2 may be a feature of memory acquisition in general and are likely coupled to the activity of proteins involved in synaptic function. We demonstrated that DDHD2 primarily interacts with STXBP1 (also referred to as Munc18-1), a key pre-synaptic exocytic protein with chaperone functions (Arunachalam et al, 2008; Han et al, 2011). We revealed that this interaction controls the transport of DDHD2 to the plasma membrane and the subsequent generation of saturated FFAs. The role of STXBP1 in neuronal FFA metabolism was further demonstrated by the reduction of FFA levels in STXBP1 knockout neurosecretory cell lines and across the brains of heterozygous STXBP1 knockout mice which is a model of STXBP1 encephalopathy (Chen et al, 2020). Together these data suggest a novel role for the *DDHD2* isozyme of PLA1 in regulating FFA changes associated with memory, via interaction with STXBP1.

## Results

### Instrumental conditioning drives region-specific changes in the brain lipidome

The predominant increase in saturated FFAs that occurred during fear conditioning suggested that this response was associated with the process of memory formation in the brain (Wallis et al, 2021). To assess if this was a general response to learning, we tested whether it was associated with other memory paradigms. Instrumental conditioning is a long-term learning process that depends on associative memories linking a behavioural response with reward and is known to initiate molecular and cellular mechanisms liable for memory formation in the posterior striatum and prefrontal cortex, amongst other regions (Balleine, 2019). Over the entire 17 days of training, the instrumental animals learned to push a lever for a food reward, while the control animals received no reward regardless of lever pushing (Fig. 1A). This was reflected by a considerably greater number of lever presses ($p < 0.001$) in the instrumental mice compared to the control cohort which exhibited essentially no lever presses (Fig. 1B–E).

    

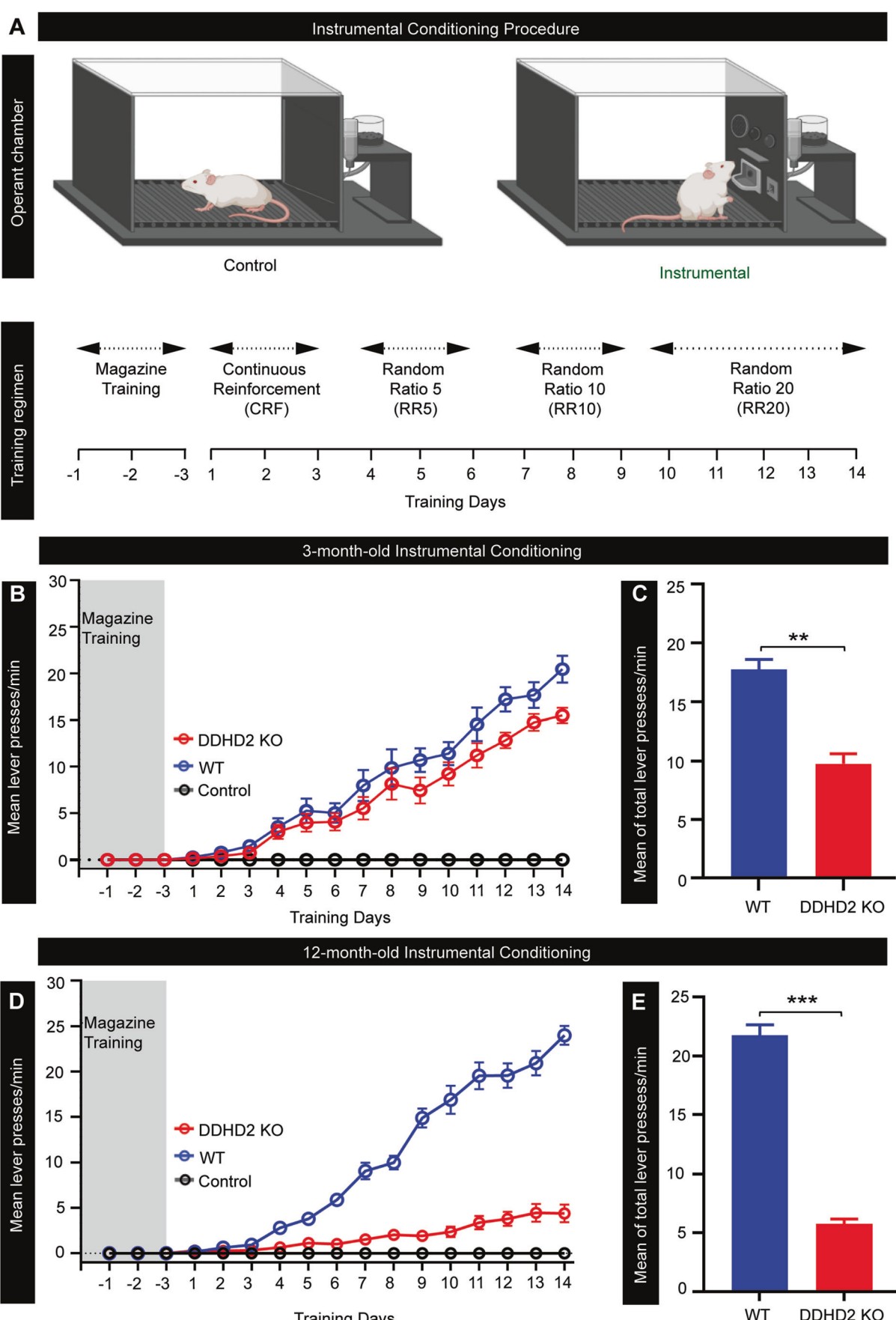

**Figure 1. Behavioural responses to instrumental conditioning in *DDHD2*+/+ vs *DDHD2*−/− mice.**

Instrumental conditioning was performed on cohorts of *DDHD2*+/+ (WT) or *DDHD2*−/− (KO) mice vs controls, respectively. (A) Illustration of the procedure for operant lever conditioning in instrumental vs control mice; $N = 20$ (the figure was created using Biorender.com). (B) Line graph showing mean lever presses per minute in 3mo WT vs KO mice. (C) Bar graph showing mean of total lever presses per minute in *DDHD2*+/+ vs *DDHD2*−/− mice at 3mo. (D) Line graph showing mean lever presses per minute in 12mo WT vs KO mice. (E) Bar graph showing mean of total lever presses per minute in *DDHD2*+/+ vs *DDHD2*−/− mice at 12mo. Data information: In (C, E), the significance of the difference between each group ($n = 20$ in each cohort of instrumental and control animals) as determined by unpaired *t*-test is indicated by asterisks **$p < 0.01$, ***$p < 0.001$. Error bars represent the cumulative standard error of the mean (SEM) for all groups and parameters. Source data are available online for this figure.

FFAs from each condition were solvent extracted from homogenised tissue, independently derivatized using FFAST and combined with labelled FFA standards prior to multiplex tandem liquid chromatography-mass spectrometry (LC-MS/MS). FFAST fragment peak intensities relative to the internal standard were used to determine the absolute abundance of the 19 FFAs in each condition (Fig. 2A; Appendix Tables S1–S3).

Consistent with the preserved intrinsic architecture of the 5 brain regions examined, we uncovered considerable heterogeneity in the concentration of FFAs across the assayed brain regions, with the highest concentrations relative to tissue weight in the prefrontal cortex, and lowest concentrations in the cerebellum. Following instrumental conditioning, we uncovered an overall increase in the FFA levels across the different brain regions of instrumental mice (Fig. 2B). Among the FFAs that increased following instrumental conditioning, saturated FFAs, particularly stearic acid (C18:0), palmitic acid (C16:0), myristic acid (C14:0), and hexanoic acid (C6:0) constituted the majority, while polyunsaturated FFAs such as arachidonic acid (C20:4), and monounsaturated FFAs such as oleic acid (C18:1), and palmitoleic acid (C16:1), represented the predominant unsaturated FFAs across all brain regions (Fig. 2B). We also observed a significant reduction in octanoic acid (C8:0), lignoceric acid (C24:0), and erucic acid (C22:1) across the different brain regions (Appendix Figs. S1 and S2). In agreement with previous results obtained in auditory fear-conditioned rats (Wallis et al, 2021), our data revealed that instrumental conditioning drives an overall increase in FFA levels across the key brain regions, with the most significant changes seen in saturated FFAs (Appendix Fig. S2A).

## Disruption of the *DDHD2* gene drives age-dependent reduction in long-term memory performance and saturated FFAs levels (including myristic acid)

The mammalian PLA1 family of enzymes consists of both intracellular and extracellular isoforms, all of which contain a DDHD domain. Among the 13 mammalian PLA1 isoforms, only three are intracellular: iPLA1α (also known as DDHD1 and phosphatidic acid-preferring PLA1; PAPLA1), iPLA1β (also known as p125), and iPLA1γ (also known as DDHD2) (Inoue et al, 2012). Activation of the DDHD2 domain has previously been linked with an increased generation of reactive oxygen species and apoptosis (Maruyama et al, 2018), mitochondrial dysfunction and mitochondrial phospholipid remodelling (Yadav and Rajasekharan, 2016), lipid trafficking, and signalling (Baba et al, 2014; Lev, 2004). DDHD2 is thought to utilise phosphatidic acid (PA) as its preferred substrate, and by doing so generates lysophosphatidic acids (LPAs) and saturated FFAs (Baba et al, 2014; Higgs and Glomset, 1996). We therefore sought to elucidate the effect of knocking out the *DDHD2* gene, which codes for an endogenous PLA1 that has a role

in the generation of FFAs during longer-lasting memory acquisition in vivo (Inloes et al, 2014; Inloes et al, 2018). Instrumental conditioning was carried out using separate cohorts of *DDHD2*−/− vs *DDHD2*+/+ litter mates at 3mo and 12mo. Data obtained from instrumentally conditioned *DDHD2*−/− mice exhibited a trend towards memory impairment at 3mo (asymptomatic young animals; Fig. 1B,C), and a statistically significant memory deficit at 12mo (symptomatic animals; Fig. 1D,E).

Using FFAST LC-MS/MS (Narayana et al, 2015) as described above, we quantified the abundance of 19 targeted FFA species from 5 brain regions of instrumentally conditioned *DDHD2*+/+ and *DDHD2*−/− mice at 3mo and at 12mo, and by doing so uncovered a significant decrease in the levels of most FFAs across all targeted brain regions in *DDHD2*−/− mice (Fig. 2B and Appendix Fig. S2B). Principal component analysis (PCA) was used to dimensionally reduce the average FFA profiles from 19 dimensions to 2, which revealed that the FFA profiles from instrumentally conditioned *DDHD2*+/+ and *DDHD2*−/− mouse cohorts cluster separately from their respective control mouse cohorts, at both 3mo and 12mo (Appendix Fig. S1A–C). Examination of the underlying components using scatter plots of absolute change vs fold change showed that the observed clustering was predominantly driven by saturated FFAs such as myristic acid (C14:0), palmitic acid (C16:0), and stearic acid (C18:0), and to a lesser extent, unsaturated FFAs docosahexaenoic acid (C22:6), linolenic acid (C18:3), and arachidonic acid (C20:4), which increased the most in instrumentally conditioned *DDHD2*−/− mice (Appendix Figs. S1D–G and S2A,B).

In response to instrumental conditioning, targeted LC-MS/MS of phospholipids PA, PE, PS, and PC species (Fig. 3) across the different brain regions of 12mo mice showed marked decreases in a number of myristoyl-containing phospholipid analytes especially in the dorsal hippocampus as depicted by the dotted boxes in Fig. 3B–D. However, no clear-cut difference was observed across the different brain regions of 12mo *DDHD2*+/+ and *DDHD2*−/− mice, between the phospholipid species containing saturated FFAs vs those containing unsaturated FFA (Fig. 3A–K; Appendix Fig. S2C,D). Overall, our data suggests that DDHD2 substantially contributes to lipid metabolism through the generation of saturated FFAs, and that its ablation triggers a major reduction in the concentration of these saturated FFAs across different regions of the brain, resulting in a blunted response following instrumental conditioning tasks.

## PLD1 knockout does not affect contextual fear conditioning

The involvement of PLA1 and PLA2 in generating FFAs from phospholipid substrates has been documented. PA generated by PLD1 is one of the possible substrates of PLA1 (PA-PLA1/ DDHD1). The requirement for PLD-synthesized PA in calcium-

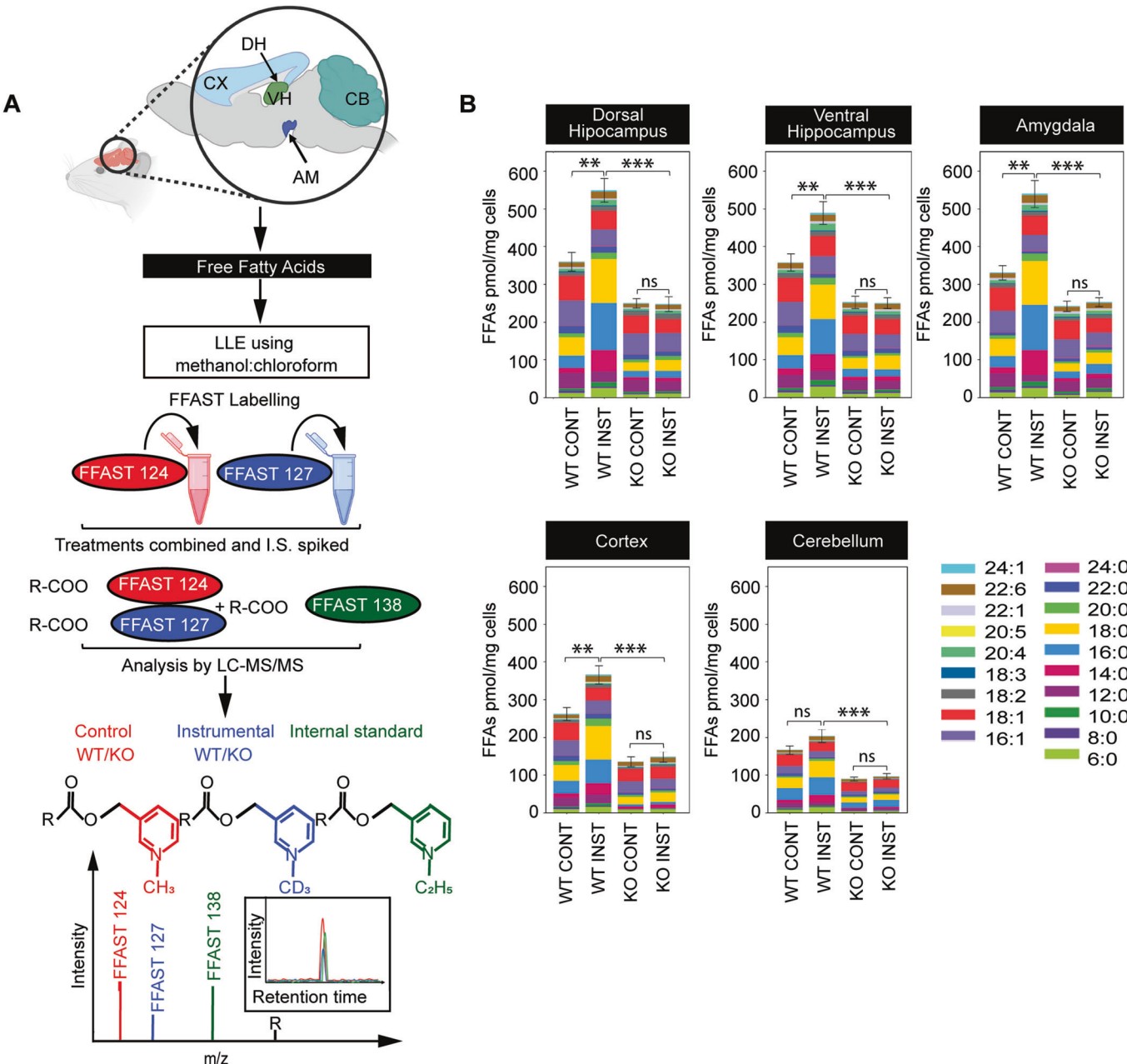

**Figure 2. Total FFA levels across different brain regions of 12-month-old instrumentally conditioned *DDHD2*⁺/⁺ vs *DDHD2*⁻/⁻ mice.**

(**A**) Schematic workflow for Free Fatty Acid (FFA) extraction from different brain regions including cortex (CX), ventral hippocampus (VH), amygdala (AM), dorsal hippocampus (DH) and cerebellum (CB), and subsequent derivatization using FFAST, and LC-MS/MS analysis. For each brain region, FFAs were extracted from the two different experimental conditions (instrumental or context control mice) and labelled separately at the carboxy-terminus using FFAST for subsequent multiplexing LC-MS/MS analysis, together with FFAST-labelled internal standards (as per Narayana et al, 2015 (Narayana et al, 2015)). This workflow was replicated 6 times, to establish FFA abundance in each of the 6 mice used for each experimental condition. (**B**) Bar plots showing the profile of 19 FFAs obtained by FFAST LC-MS/MS across 6 brains of 12-month-old control vs instrumentally conditioned *DDHD2*⁺/⁺ vs *DDHD2*⁻/⁻ mice with analytes shown by acyl chain composition. Data information: In (**B**), the significance of the difference between each group ($n = 6$ biological replicates) as determined by unpaired $t$-test with Holm–Sidak post hoc correction is indicated by asterisks **$p < 0.01$, ***$p < 0.001$, ns = not significant. Error bars represent the cumulative standard error of the mean (SEM) for all groups and parameters. Source data are available online for this figure.

regulated exocytosis in neuroendocrine and endocrine cells is a well-known mechanism (Bader and Vitale, 2009), and its contribution to neuronal signalling and membrane trafficking processes has also been documented (Humeau et al, 2001; Raben and Barber, 2017). However, knocking out PLD1 was recently shown not to affect contextual fear conditioning (Santa-Marinha et al, 2020) suggesting that PA is not a major substrate for the generation of saturated FFAs involved in memory acquisition. To verify this, we used a PLD1 KO mouse model in a contextual fear memory paradigm and examined the effect on phospholipid and

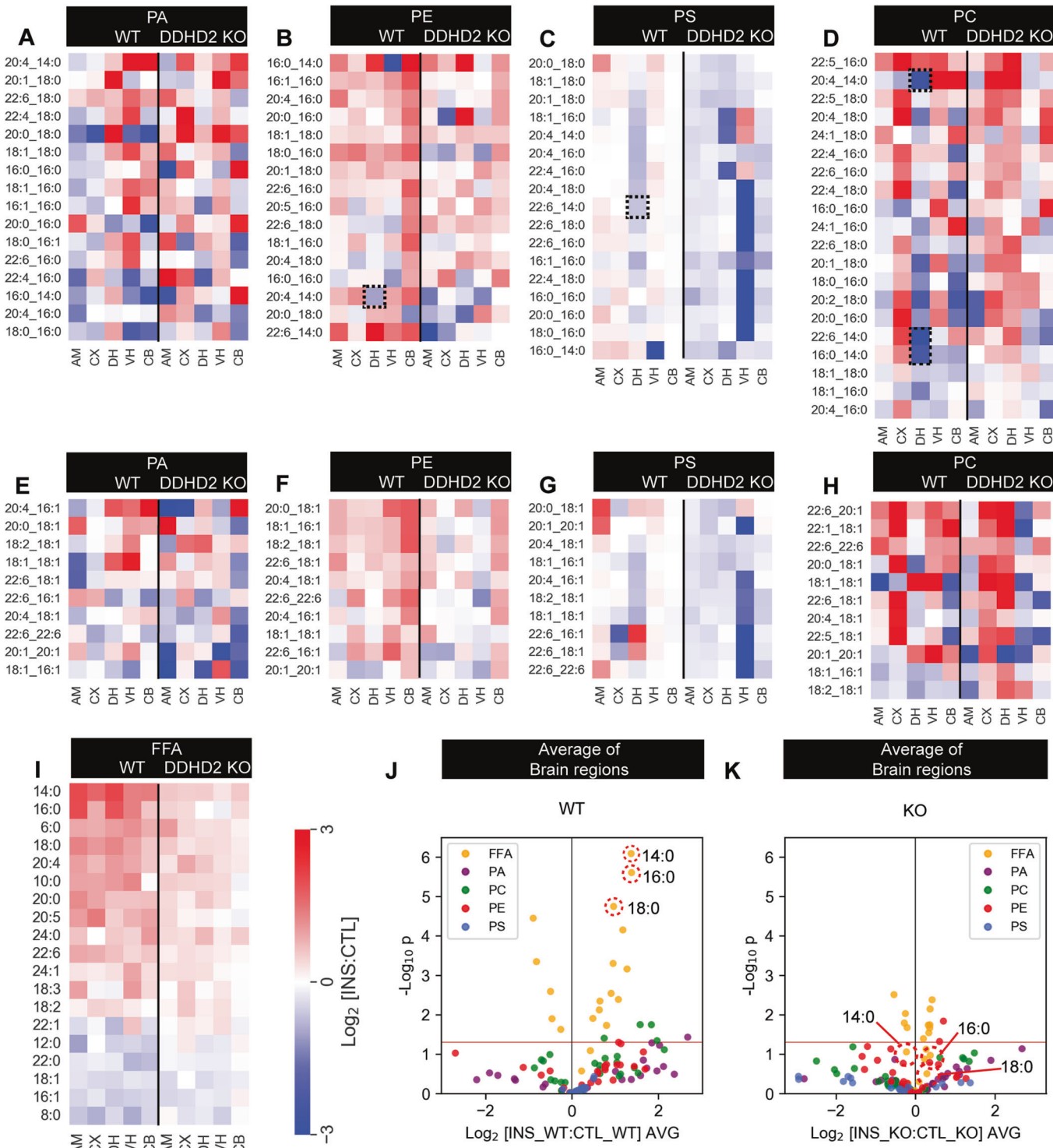

FFA levels. We used FFAST and targeted LC-MS/MS phospholipidomics to profile the mouse brain lipidome immediately following conditioning, and at the 24-hr timepoint representing long-term memory consolidation.

Our results reveal that while PLD1 KO exacerbated the fear response in these animals, no changes to their long-term fear memory retention were observed (Appendix Fig. S3A,B) in agreement with the previous report (Santa-Marinha et al, 2020). As expected, the PLD1 KO brain extracts exhibited major decreases in PA and PE (Appendix Fig. S4). This strongly suggests that DDHD1 is not involved in memory acquisition. Further, PLD1 knockout drove a heterogenous FFA response of saturated FFAs but not the major decrease that would be expected if PA was the primary substrate (Appendix Fig. S4A,B). Another intriguing

**Figure 3. Phospholipid and FFA profile in 12mo mice.**

Lipid response to instrumental conditioning across the brain of DDHD2 knockout (KO) and wild type (WT) animals. (A–H) Phospholipid responses (A–D) phospholipids containing C14:0, C16:0 and C18:0 acyl chains. (E–H) phospholipids not containing C14:0, C16:0 and C18:0 acyl chains. (I) Fold-change response of FFAs to instrumental conditioning at 12mo. (J, K) Average response of FFAs and phospholipids across all the brain regions in cohorts of $DDHD2^{+/+}$ (WT) or $DDHD2^{-/-}$ (KO) mice: AM—central amygdala, CX—Cortex, VH—ventral hippocampus, DH—dorsal hippocampus, CB—cerebellum. Lipids: PA—phosphatidic acid, PC—phosphatidylcholine, PE—phosphatidylethanolamine, PG—phosphatidylglycerol, PS—phosphatidylserine. Dotted boxes show myristoyl-containing phospholipids decreasing in the dorsal hippocampus. Data information: In (A–I), each pixel in the heatmap represents the average change in total abundance of all lipids of a given class across the 6 measured brain regions (n = 6 biological replicates). In (J, K), each dot on the volcano plot represents the average change in abundance of a single lipid analyte across 5 measured brain regions in control vs instrumentally conditioned $DDHD2^{+/+}$ vs $DDHD2^{-/-}$ mice. Analytes below the red line represent those whose change in abundance was not statistically significant (two-tailed $t$-test $p > 0.05$). Source data are available online for this figure.

observation in our results is the significant increase in specific PC, PE, and PS species 24 hr post contextual fear conditioning, suggesting that PLD1 KO membrane perturbations lead to rapid lipidomic changes thereby inducing compensatory responses from other enzymes. It is therefore likely that DDHD2 is the primary PLA1 involved in generating saturated FFAs in response to memory acquisition.

## $DDHD2^{-/-}$ drives progressive decline in spatial memory, motor function, and explorative behaviours

DDHD2 knockout is a model for human spastic paraplegia (HSP), a disorder characterised by progressive intellectual disability and neuromuscular deficits (Inloes et al, 2014). Having demonstrated that DDHD2 knockout mice exhibit cognitive impairment in a specific instrumental learning paradigm that correlated to their lack of saturated FFAs, we therefore further investigated the impact of $DDHD2$ KO on cognition and neuromuscular function by carrying out a longitudinal assessment (3mo to 12mo) of the age of onset and progression of these deficits. Motor coordination and strength were assessed using rotarod and grip strength tests, respectively. The onset of impairment in motor coordination was observed at 5mo in $DDHD2^{-/-}$ animals and from this time point progressively declined throughout the period of the study (Fig. EV1A). In comparison to the wild-type animals, $DDHD2^{-/-}$ mice demonstrated a progressive decline in motor strength, with an onset at 6mo (Fig. EV1B). These results suggest that the $DDHD2$ KO drives a progressive decline in motor function in mice beginning from 5mo, which further corroborates previous reports suggesting that the ablation of the $DDHD2$ gene elicits the manifestation of HSP and its associated neuromotor dysfunction symptoms (Blackstone, 2018; Murala et al, 2021).

Using automated activity monitoring, we also longitudinally assessed open-field locomotor performance, alongside explorative and vigilance activity. 3mo $DDHD2^{-/-}$ mice exhibited a significant decrease in vertical time, vertical counts, jump time, and jump counts when compared to $DDHD2^{+/+}$ mice. The observed decrease was progressive and more significant in 12mo mice (Fig. EV1C–G). The progressive decrease in these neuromuscular parameters suggests impaired vigilance, exploration, and escape response. However, there was no significant difference in the resting time and ambulatory distance of $DDHD2^{+/+}$ and $DDHD2^{-/-}$ mice.

To further investigate cognitive effects of DDHD2 knockout, we adopted a longitudinal novel object location (NOL) test to investigate short-term spatial memory impairment. The NOL behavioural paradigm relies on the innate instincts of mice to explore novel locations more than familiar ones (Blackmore et al,

2022). Results demonstrated a significant decrease in the time that $DDHD2^{-/-}$ mice spent exploring the NOL compared to the wild-type mice from 3mo onwards, until the termination of the study (Fig. EV1H). This suggests that disruption of the $DDHD2$ gene drives age-dependent impacts on spatial memory. Taken together the results suggest that $DDHD2^{-/-}$ drives progressive age-dependent alteration in mice behaviour and consequently DDHD2 is critically involved in memory and neuromuscular function via production of saturated FFAs.

## STXBP1 is a major DDHD2-binding protein

The mechanism by which DDHD2 activity mediates instrumental and spatial memory is currently unknown. The fact that saturated FFA increases are activity-dependent makes it likely that DDHD2 performs its function at the synapse. Although it is known that DDHD2 is expressed in neurons (Inloes et al, 2014), the mechanisms through which it is trafficked to the synaptic compartment where it modifies the lipid landscape are unclear. We hypothesized that the production of FFAs is associated with synaptic activity, likely through interaction with a resident synaptic protein. To investigate this, we carried out a series of co-immunoprecipitation pull-down assays from neurosecretory pheo-chromocytoma (PC12) cells to identify molecules interacting with DDHD2 (Appendix Fig. S5). Untargeted high-resolution tandem mass spectrometry (HRMS) analysis of peptide digests from a DDHD2 pull-down identified 150 proteins at a 1% false discovery rate. The number of proteotypic peptides with an identification confidence score above the 95% acceptance threshold, and the level of coverage these peptides have with the sequence of the identified protein, were together used as an approximate measure of the abundance of each identified protein. Reassuringly, using this metric, DDHD2 was the most abundant protein identified in the pull-downs—with 206 unique proteotypic peptides identified, covering 80.4% of the DDHD2 sequence (Uniprot accession: D3ZJ91). The second most abundant protein identified was Syntaxin-binding protein 1 (STXBP1, also known as Munc18-1; Uniprot accession: P61765) with 94 peptides identified, covering 76.4% of the protein sequence. This suggests that STXBP1 is a primary binding partner of DDHD2. STXBP1 is a synaptic protein that is critically involved in the priming of secretory and synaptic vesicles (Jiang et al, 2023) and is an essential component of the presynaptic neurotransmitter release machinery—with its knockout leading to a complete loss of neurotransmitter release and perinatal death (Martin et al, 2013; Verhage et al, 2000).

To explore this interaction further, we performed a functional association analysis by searching the list of identified proteins

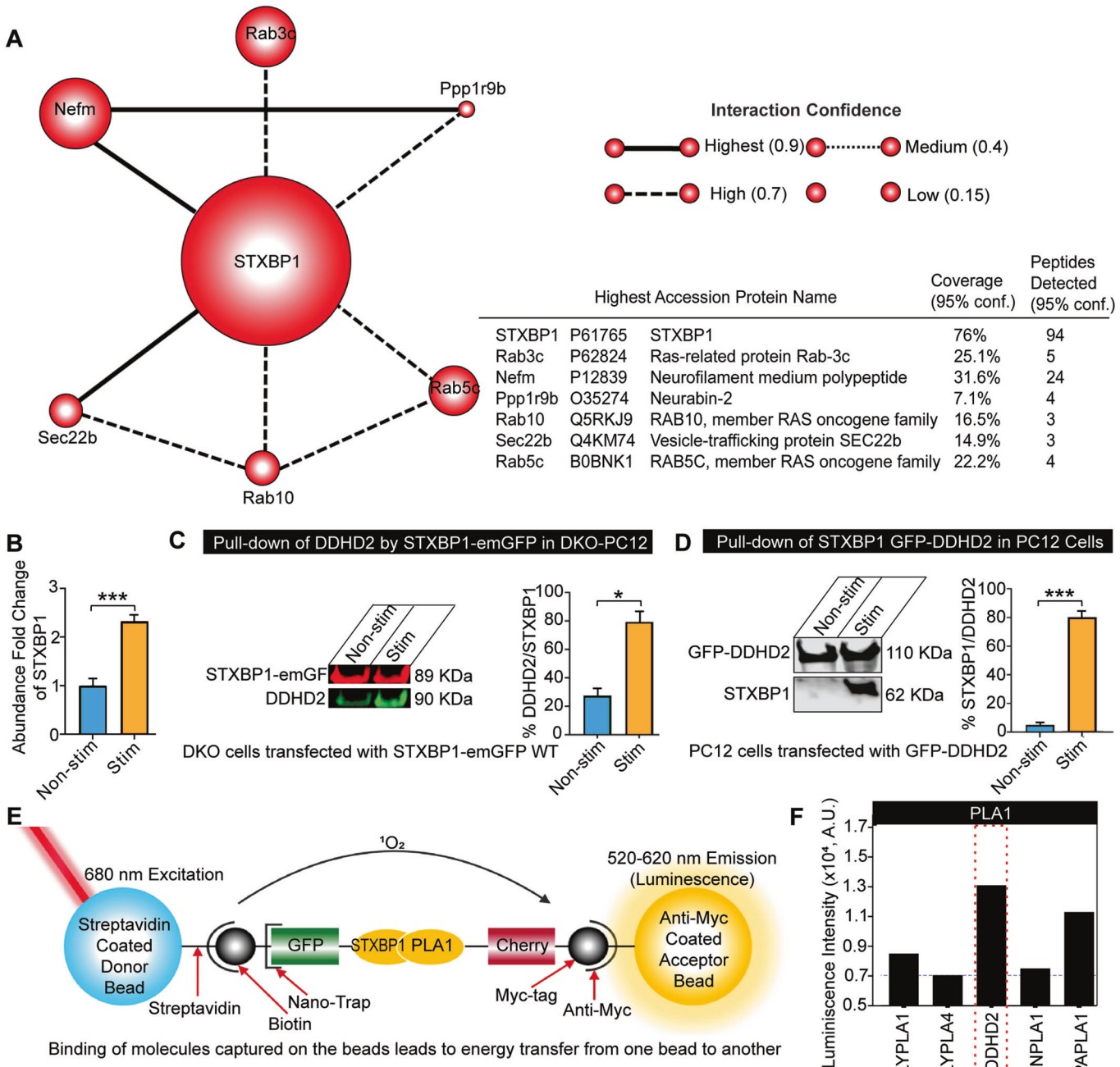

against the Search Tool for Retrieval of Interacting Genes/Proteins (STRING) database (Szklarczyk et al, 2021). This revealed that among the proteins that were co-purified with DDHD2 under a resting state (unstimulated), the primary interactome of STXBP1 consisted of six significant interactions including Rab3c, Rab10, Rab5c, Nefm, Ppp1r9b and Sec22b (Fig. 4A). The primary role of these proteins involves vesicle and membrane organisation, suggesting that STXBP1 may be acting as an interface between DDHD2 and membrane-organising structures. Gene ontology (GO) cellular component analysis revealed other proteins co-purifying with DDHD2 to a lesser degree (Appendix Fig. S5). To further validate the interaction between STXBP1 and DDHD2, a targeted multiple reaction monitoring mass spectrometry (MRM-MS) assay

specific for seven proteotypic peptides for STXBP1, and for four proteotypic peptides for DDHD2, was employed. This assay was applied to the DDHD2 pull-down, providing a parallel mass spectrometric measurement confirming the HRMS identifications of both DDHD2 and STXBP1, and was also applied to a STXBP1 pull-down from PC12 cells, which confirmed the cognate presence of DDHD2 (Appendix Fig. S5). This suggests that DDHD2 and STXBP1 undergo a biologically significant interaction. To assess whether the binding between DDHD2 and STXBP1 was altered in response to cellular stimulation, the targeted MRM-MS assay was employed to quantitatively assess the relative amount of STXBP1 that co-purified with DDHD2 in response to high $K^+$ stimulation prior to pull-down. This resulted in a 2.3-fold significant increase in

**Figure 4.   Identification of STXBP1 (Munc18-1) as a major DDHD2 interactor.**

High-resolution mass spectrometry (HRMS) analysis of an anti-GFP pull down from PC12 cells transfected with DDHD2-GFP. (**A**) STXBP1 and its direct interaction partners identified using STRING (https://string-db.org/) functional association analysis of proteins co-precipitating with DDHD2. STXBP1 was identified as the primary interaction partner of DDHD2. The relative size of each protein icon represents the percentage of sequence coverage with high quality (>95% identification confidence score) proteotypic peptides detected by HRMS. (**B**) Targeted MRM mass spectrometry showing relative abundance of STXBP1 peptides co-precipitated with DDHD2 in high vs low $K^+$ stimulated cultures as described in (**A**). (**C**) Binding of STXBP1 with DDHD2. Immunoblot of STXBP1-emGFP (89 KDa, red) pulldown of DDHD2 (90 KDa, green) and (**D**) GFP-DDHD2 (110 KDa) pulldown of STXBP1 (62 KDa) in non-stimulated and BaCl$_2$-stimulated PC12 cells. Densitometric quantifications of the immunoblots are shown as percentage of DDHD2 to STXBP1 and STXBP1 to DDHD2, respectively. Results are shown as mean ± SEM from four independent experiments. (**E**) Schematic of proximity-based ALPHAScreen analysis using tagged in vitro translated proteins to illustrate STXBP1$^{WT}$ interactions with phospholipases. The biotin-coupled GFP-nanotrap recruits the GFP-tagged proteins (STXBP1$^{WT}$) and binds with streptavidin-coated donor beads. Myc-tagged proteins (phospholipases, all expressed at the same concentration) bind to anti-myc-coated acceptor beads. The donor bead bound to STXBP1$^{WT}$ releases singlet oxygen ($^1O_2$) upon illumination at 680 nm, which only diffuses 200 nm in solution. Upon reaching an acceptor, $^1O_2$ reacts with derivatives in the acceptor bead containing phospholipases and luminescence is detected at 520–620 nm. The luminescence observed indicates the relative strength of the binding. (**F**) ALPHAScreen STXBP1$^{WT}$ interactions with phospholipases. The strong interaction of STXBP1 and DDHD2 is highlighted with a red box. Data information: In (**A**), the thickness and hatching of the lines connecting each protein represent the confidence score of the strength of the interaction as determined by the STRING knowledge base matching algorithm. In (**B–D**), the significance of the change between the different experimental conditions as determined by two-tailed unpaired Student's *t*-test, *$p < 0.05$, ***$p < 0.001$. Source data are available online for this figure.

STXBP1 associated with DDHD2 immediately after stimulation compared to unstimulated controls, indicating that an activity-dependent increase in binding was occurring (Fig. 4B). To further confirm this binding, we used gene-edited neurosecretory cells lacking STXBP1/2 (PC12-DKO) (Kasula et al), which we transfected with STXBP1-emGFP. Cells were stimulated using high $K^+$ and solubilized for pull-down via GFP-trap. We further confirmed that DDHD2 was detected in the pull-down fraction and that it was increased in response to stimulation (Fig. 4C). Conversely, we expressed GFP-DDHD2 and performed GFP-Trap, and detected STXBP1 by western blotting, which was significantly increased following stimulation (Fig. 4D).

Finally, we investigated whether DDHD2 directly binds to STXBP1 in vitro, using tagged proteins in the proximity-based Amplified Luminescent Proximity Homogeneous Assay Screen (ALPHAScreen) (Martin et al, 2013; Sierecki et al, 2013). In this assay, streptavidin-coated donor beads bind a biotin-coupled GFP-nanotrap that recruits GFP-tagged STXBP1$^{WT}$ protein. The acceptor beads coated with anti-myc antibody bind to myc-tagged DDHD2 and other myc-tagged PLA1 isoforms (Fig. 4E). Protein-protein interactions are detected via energy transfer luminescence. The ALPHAScreen assay confirmed that STXBP1 directly interacts with DDHD2, and to a lesser extent, with PAPLA1, PNPLA1, and LYPLA1 and 4 isoforms (Fig. 4E,F).

## STXBP1 controls DDHD2 transport to the plasma membrane

Having demonstrated that STXBP1 binds to DDHD2, we next sought to investigate the functional significance of this interaction. STXBP1 was originally described and named due to its significant interaction with syntaxin1A (STX1A) to mediate vesicular priming and STX1A transport to the plasma membrane in neurosecretory cells (Martin et al, 2013; Rickman et al, 2007). We therefore first investigated whether STXBP1 could be responsible for the transport of DDHD2 to the plasma membrane via a similar mechanism to STX1A.

We performed immunostaining of neurosecretory PC12 cells and found that DDHD2 was largely localized to the periphery of the cells, suggestive of plasma membrane localisation (Fig. 5A–D). Further, DDHD2 plasma membrane localisation was similar to that of STX1A (Fig. 5E–H). We used gene-edited PC12 cells genetically engineered to knock out STXBP1/2 (DKO PC12) (Jiang et al, 2023; Kasula et al, 2019) to investigate the role of STXBP1 in the transport of DDHD2 to

the plasma membrane and found that in the absence of STXBP1/2, the plasma membrane targeting of both DDHD2 and STX1A was severely impacted (Fig. 5I–K). Analysis of DDHD2 with STX1A in PC12 cells showed high level of plasma membrane targeting which was ablated in DKO PC12 cells (Fig. 5L). Re-introducing GFP-tagged wild-type STXBP1 into DKO PC12 cells rescued the plasma membrane localization of both DDHD2 and STX1A (Fig. 5M–R), indicating that STXBP1 has a central role in the transport of DDHD2 to the plasma membrane. We also checked whether STXBP1 could control the expression levels of DDHD2, and vice versa, in vitro (Appendix Fig. S6). The relative protein expression levels were not significantly different between PC12 cells and STXBP1/2 DKO PC12 cells when assessed by western blot assay and PCR (Appendix Fig. S6A,B), suggesting that STXBP1 controls the targeting of DDHD2 to the plasma membrane, but not its expression. Taken together, our data demonstrate that STXBP1 acts as a chaperone targeting both STX1A and DDHD2 to the plasma membrane.

Various PLA1 isoforms have been implicated in regulating the secretory pathway at the ER and Golgi interface (Bechler et al, 2012). We therefore wondered whether DDHD2 could also function at the level of the early secretory pathway. We performed electron microscopy of the secretory pathway in cultured E16 hippocampal neurons from C57BL/6J and DDHD2$^{-/-}$ mice at DIV21 which revealed evidence of arrested secretory vesicles suggestive of a transport defect in the ER-Golgi intermediate compartment (ERGIC). Additional perturbation of the early secretory pathway was demonstrated by the enlarged Golgi apparatus lumen and dilated tubulo-vesicular ERGIC as well as an accumulation of budding vesicles in the ERGIC of DDHD2$^{-/-}$ neurons compared to C57BL/6J neurons (Fig. 6A). There was also a significant reduction in the mean grey intensity of ERGIC53 and GM130 in hippocampal neurons from DDHD2$^{-/-}$ neurons compared to C57BL/6J neurons (Fig. 6B–E). These findings strongly implicate DDHD2 in the early secretory pathway and suggest that at least some of the dysfunction observed in HSP may be due to transport defects in key synaptic proteins.

We used SV2A as a synaptic vesicle marker and first confirmed that DDHD2 was enriched at the synapse by immunocytochemistry in DIV21 wild-type C57BL/6J neurons (Fig. 7A). We further imaged (Fig. 7B) and quantified the mean intensity of SV2A spots in both WT and DDHD2$^{-/-}$ hippocampal neurons and observed a significant reduction in the mean intensity of SV2A spots in DDHD2$^{-/-}$ neurons compared to C57BL/6J neurons (Fig. 7C). This

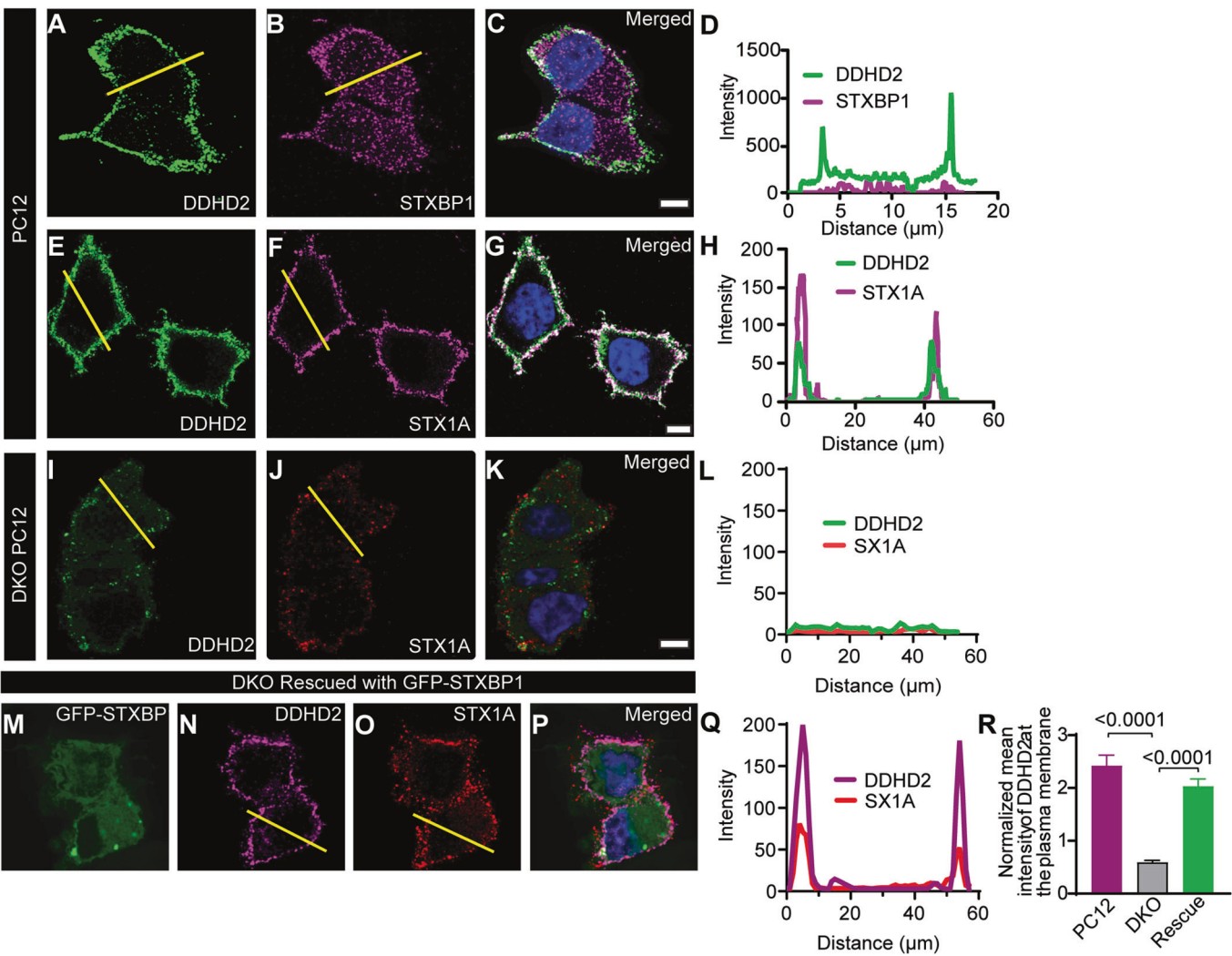

**Figure 5. STXBP1 (Munc18-1) plays a key role in the transport of DDHD2 to the plasma membrane.**

Representative image of BaCl$_2$-stimulated PC12 cells immunostained for (**A**) endogenous DDHD2 and (**B**) endogenous STXBP1. The merged image of the channels is shown in (**C**), and the fluorescence intensity profile of DDHD2 (green) and STXBP1 (magenta) from the indicated yellow line is shown in (**D**). Representative images of BaCl$_2$-stimulated PC12 cells immunostained for (**E**) endogenous DDHD2 and (**F**) endogenous STX1A. The merged image of the channels is shown in (**G**), and the fluorescence intensity profile of DDHD2 (green) and STX1A (magenta) from the indicated yellow line is shown in (**H**). Representative images of BaCl$_2$-stimulated STXBP1/2 double knockout (DKO) PC12 cells stained for (**I**) endogenous DDHD2 and (**J**) endogenous STX1A. The merged image of the channels is shown in (**K**), and the DDHD2 intensity with STX1A in DKO PC12 cells is shown in (**L**). DKO PC12 cells following rescue with GFP-STXBP1 is shown in (**M–P**). (**Q**) Quantification of mean intensity (±SEM) of DDHD2 on the plasma membrane in WT PC12, DKO PC12 cells and DKO PC12 cells rescued with GFP-STXBP1. (**R**) Normalised mean intensity of DDHD2 on the plasma membrane. Data information: In (**R**), the significance is tested using ordinary one-way ANOVA multiple comparison test. Scale bars are 5 μm. $n = 10$–22 cells from 3 independent experiments and error bars represent the cumulative standard error of the mean (SEM) for all groups and parameters.

suggested that DDHD2 could control the transport of synaptic components. In support of this, electron micrographs showed a significant reduction in the number of synaptic vesicles in DDHD2$^{-/-}$ neurons compared to C57BL/6J neurons (Fig. 7D,E). We further checked whether DDHD2 could also affect the expression of other synaptic components including STXBP1, synaptotagmin 1, and SNAP-25. However, the relative protein expression levels were not significantly different between DDHD2$^{-/-}$ neurons compared to C57BL/6J neurons when assessed by western blot assay (Appendix Fig. S6C,D) suggesting that DDHD2 controls the trafficking of synaptic components rather than their expression.

## STXBP1 binding to DDHD2 controls FFA levels in neurons and neurosecretory cells

Having demonstrated that STXBP controls the targeting of DDHD2 to the plasma membrane we tested whether STXBP could in turn control DDHD2 activity-dependent increase in FFAs in both neurosecretory cells and neurons. We first stimulated WT and DKO PC12 cells and used FFAST to compare the FFA response to secretagogue stimulation. Our results showed that stimulation of PC12 cells led to a significant increase in predominantly saturated FFAs, particularly C14:0, C16:0, and C18:0 (Fig. 8A) as previously detected in chromaffin cells (Narayana et al, 2015). In DKO PC12

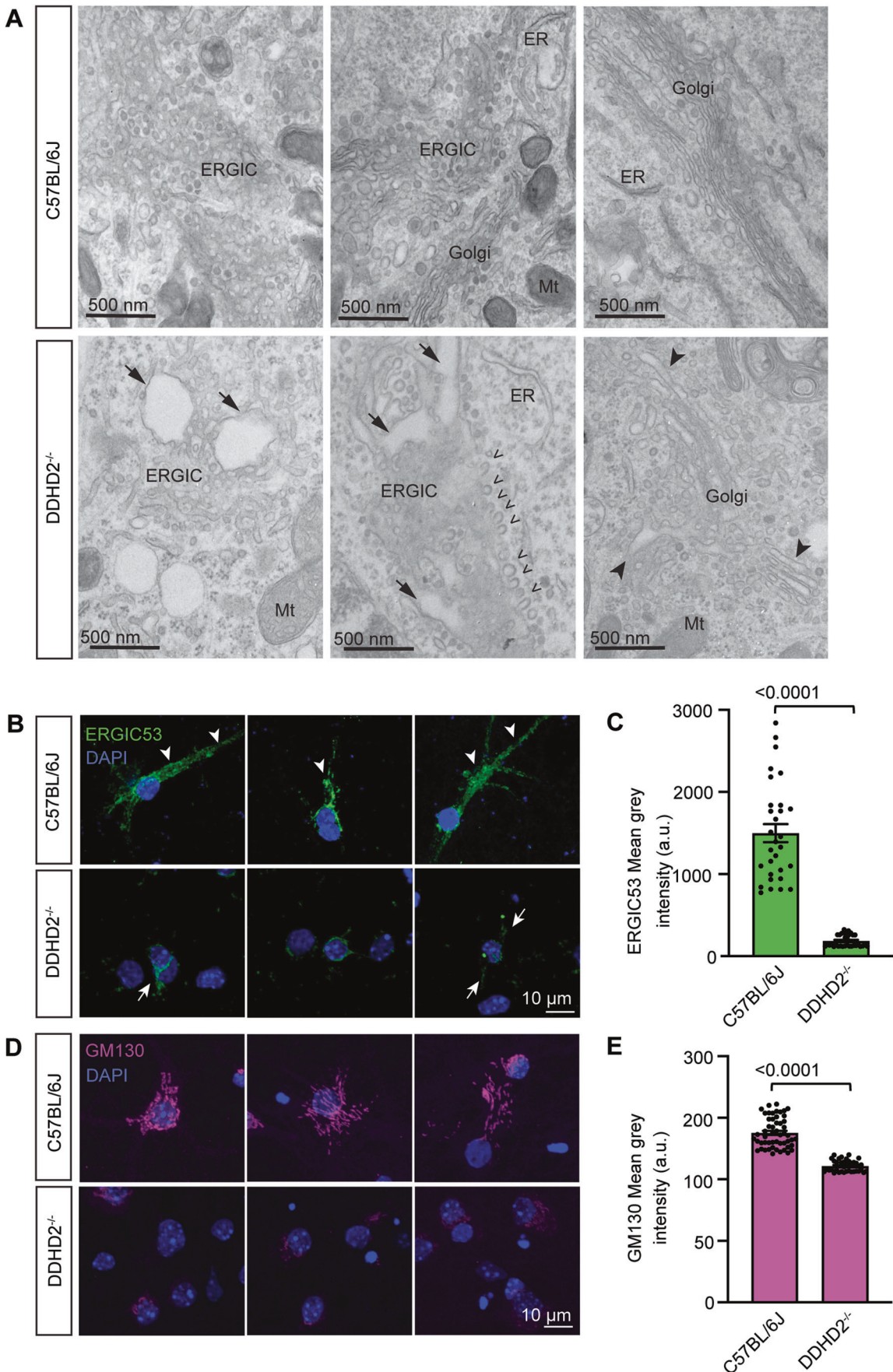

**Figure 6.  DDHD2 knockout perturbs the secretory pathway.**

(A) Selection of representative electron micrographs of the secretory pathway in cultured E16 hippocampal neurons from C57BL/6J and DDHD2$^{-/-}$ mice at DIV21. Images show enlarged Golgi apparatus lumen (arrowheads) and dilated tubulo-vesicular ERGIC (arrows), as well as an accumulation of budding vesicles (open arrowheads) in the ERGIC of DDHD2$^{-/-}$ neurons compared to C57BL/6J neurons. Mitochondria (Mt) and endoplasmic reticulum (ER) are indicated for reference. (B) A selection of representative maximum projections of cultured E16 hippocampal neurons from C57BL/6J and DDHD2$^{-/-}$ mice at DIV21 immunostained against endogenous ERGIC53 (green) and stained with DAPI (blue). Arrows point at ERGIC53 distributed from the perinuclear space to the somatodendritic area. (C) Quantification of ERGIC53 mean grey intensity in C57BL/6J and DDHD2$^{-/-}$ neurons. (D) A selection of representative maximum projection of cultured E16 hippocampal neurons from C57BL/6J and DDHD2$^{-/-}$ mice at DIV21 immunostained against endogenous GM130 (magenta) and stained with DAPI (blue). (E) Quantification of GM130 mean grey intensity in C57BL/6J and DDHD2$^{-/-}$ neurons. Data information: In (A), statistical testing of non-normally distributed data was done using Mann–Whitney U test. $n = 30$–53 regions of interest (ROIs) from 3 independent experiments, In (C, E), the significance of the difference between each group as determined by unpaired t-test is indicated as <0.0001, and the error bars represent the cumulative standard error of the mean (SEM) for all groups and parameters. $n = 30$–55 cells from 3 independent experiments.

cells lacking STXBP1 and 2, the basal (unstimulated) FFA levels were significantly reduced compared to WT PC12 cells, suggesting that STXBP1 is required for maintaining basal FFA levels in PC12 cells (Fig. 8A). In addition, the activity-dependent increase in saturated FFAs was abolished in stimulated DKO PC12 cells, indicating that STXBP1 controls the production of saturated FFAs in response to secretagogue stimulation (Fig. 8A). In agreement with previous studies using STXBP-1/2 double-knockdown neurosecretory cells (Han et al, 2010; Han et al, 2009), DKO PC12 cells were also unable to secrete exogenous neuropeptide-Y (NPY) in response to depolarising stimulus in a NPY-human placental alkaline phosphatase (NPY-hPLAP) release assay (Appendix Fig. S6E). Taken together, both baseline lipid homeostasis and activity-dependent FFA generation correlated with functional exocytic release and are likely controlled by STXBP1 (Appendix Fig. S6-E).

We next sought to establish whether the FFA lipid profile can be rescued upon re-expression of STXBP1$^{WT}$ in DKO PC12 cells. Our results revealed that re-expression of STXBP1$^{WT}$ completely rescued the FFA basal levels, demonstrating that STXBP1 is necessary and sufficient to control saturated FFAs levels (Fig. 8B). Since STXBP1 is a key regulator of vesicular priming in both secretory and synaptic vesicles (Deak et al, 2009; Kasula et al, 2016), we assessed whether STXBP1 missense mutants with altered priming and STX1A binding were similarly able to rescue the FFA profile. Both the re-expression of STXBP1$^{\Delta317\text{-}333}$, a priming deficient loop mutant that prevents the opening of STX1A and SNARE assembly (Kasula et al, 2016; Martin et al, 2013), and STXBP1$^{F115E}$, a hydrophobic pocket mutant (HPM) that ablates the binding of the SNARE complex with STXBP1 (Han et al, 2010; Han et al, 2011; Malintan et al, 2009), were able to restore the levels of saturated FFAs (Fig. 8B). This demonstrated that DDHD2 transport to the plasma membrane is underpinned by STXBP1 but is independent of STXBP1's function in vesicular priming. STXBP1 is the main isoform in neurons and neurosecretory cells. However, to explore the possible role of STXBP2 in FFA metabolism, we compared the basal levels of FFAs in STXBP1 single knockout (MKO) cells (Jiang et al, 2023) to the FFA responses of PC12 cells and observed that the basal FFA levels in MKO cells were also significantly reduced, confirming that STXBP1 plays a major role in the regulating the generation of FFAs (Appendix Fig. S7).

While neurosecretory cells are a tractable system for studying aspects of neurosecretion, ultimately, we sought to establish the roles of DDHD2 and STXBP1 in neurons. To confirm that synaptic DDHD2 is necessary for the activity-dependent FFA response, we treated PC12 cells and hippocampal neurons with a pharmacological inhibitor of DDHD2, KLH45, prior to stimulation and FFAST analysis. Acute

pharmacological inhibition of DDHD2 abolished the activity-dependent increase in saturated FFAs (Fig. 8C,D) in both PC12 cells and neurons. The acute inhibition of DDHD2 fully prevented the FFA response to stimulation but did not affect the basal level of FFAs. DDHD2 KO neurons exhibited significantly reduced basal FFA levels which did not respond to stimulation. We tested the selectivity of KLH45 and demonstrated that no further FFA decrease was observed in DDHD2 KO neurons (Fig. 8D). Importantly, knockout of either STXBP1 or DDHD2 resulted in similarly low basal FFA levels. A critical difference between pharmacological (acute) and genetic ablation (chronic) of DDHD2 is the complete absence of synaptic DDHD2 in the latter preventing accumulation of FFAs in synaptic membranes. The significant discrepancy in basal FFA levels between acute and chronic DDHD2 inhibition may therefore stem from differential depletion of this synaptic reserve of FFAs, although we cannot rule out an extrasynaptic reserve.

Comparison of the FFA response in the PC12 cells treated with KLH45, showed that the activity-dependent FFA changes that were observed across the different conditions were largely driven by saturated FFAs as expected for DDHD2 inhibition (Fig. EV2A).

## Effect of synaptic maturation on the activity-dependent changes in the free fatty acid landscape

Having demonstrated that memory acquisition is a major driver of FFA production in the brain and that synaptic activity controls FFA production in neurons in culture, we sought to assess whether synaptic maturation and glia could influence the FFA response. Neuron-glia interplay is highly regulated to co-ordinate a plethora of neurometabolic activities ensuring brain homeostasis. We aimed to determine the extent to which glial cells contribute to both basal and activity-dependent changes observed in FFAs. We profiled the FFA changes in neuronal cell cultures from the amygdala, cortex and hippocampus in the presence and absence of arabino-furanosyl cytosine (ara-C) (Schwieger et al, 2016) to prevent mitotic glial cells (mostly astrocytes) proliferation (Fig. EV3A,B). Inhibition of glial proliferation reduced the overall FFA levels but did not change the activity-dependent increase in the saturated FFA (Fig. EV3C–E). These data demonstrate that glia contribute to some extent to the basal levels of FFAs but that neuronal synapses drive the activity-dependent response.

## STXBP1 haploinsufficiency leads to FFA deficits in the brain

We herein demonstrate that saturated FFAs are required for memory acquisition and are tightly regulated at the synapse by the

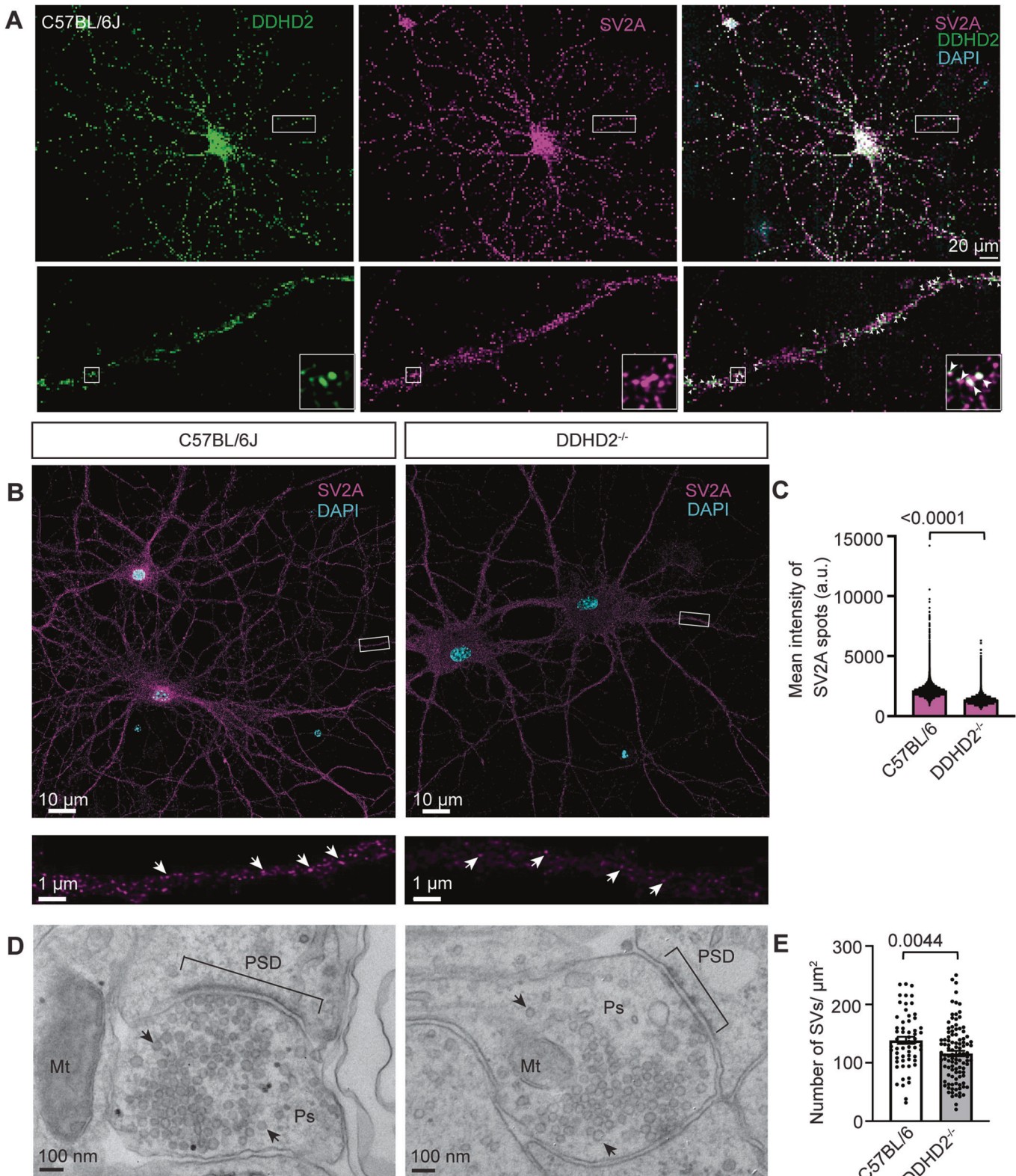

interaction between DDHD2 and STXBP1. De novo pathogenic mutations of the *STXBP1* gene lead to developmental epileptic encephalopathies (DEE)—a set of neurodevelopmental conditions associated with intellectual disability (STXBP1-DEE) (Abramov

et al, 2021; Chen et al, 2020; Guiberson et al, 2018; Miyamoto et al, 2017). STXBP1 encephalopathy encompasses a range of neurodevelopmental conditions including autism, intellectual disability (mental retardation), cognitive impairment, and movement

**Figure 7. DDHD2 knockout neurons show reduced SV2A intensity and SV numbers.**

(A) Representative maximum projection of cultured E16 hippocampal neurons from C57BL/6J at DIV21 co-immunostained for DDHD2 (green), SV2A (magenta) and DAPI (blue). Boxed areas are shown magnified in the bottom, and arrowheads point to co-localisation of DDHD2 in SV2A-positive synapses. (B) Representative maximum projection of cultured E16 hippocampal neurons from C57BL/6J and DDHD2$^{-/-}$ mice at DIV21 immunostained for SV2A (magenta) and DAPI (blue). (C) Quantification of SV2A mean spot intensity in C57BL/6J and DDHD2$^{-/-}$ neurons. (D) Representative electron micrographs of presynapses (Ps) cultured E16 hippocampal neurons from C57BL/6J and DDHD2$^{-/-}$ mice at DIV21. Post-synaptic density (PSD), synaptic vesicles (arrow) and mitochondria (Mt) are indicated for reference. (E) Quantification of number of SVs in C57BL/6J and DDHD2$^{-/-}$ neurons. Data information: In (C, E), $n = 5$ acquisitions per condition from 2 independent experiments in (C), with 94 and 147 presynapses quantified from C57BL/6J and DDHD2$^{-/-}$, respectively, from 3 independent experiments in (E). Statistical testing of normally distributed data was done using Student's $t$-test.

disorder (Chai et al, 2016; Lanoue et al, 2019; Saitsu et al, 2010; Stamberger et al, 2016; Tavyev Asher and Scaglia, 2012), and is thought to stem from haploinsufficiency altering the levels of functional STXBP1 protein and gain of toxic function (Lanoue et al, 2019). We therefore hypothesized that haploinsufficiency would lead to reduced saturated FFA production that could at least partially explain dysregulation of synaptic activity and associated cognitive defects. We used a genetically distinct haploinsufficient mouse model (STXBP1$^{+/-}$) showing a 40–50% decrease in the levels of STXBP1 protein, which recapitulates the cognitive, motor, seizure, and psychiatric phenotypic hallmarks of human encephalopathy (Abramov et al, 2021; Chen et al, 2020). As anticipated, STXBP1 haploinsufficient animals showed significant reduction in STXBP1 protein levels with no significant decrease in the levels of DDHD2 protein (Figs. 9A,B and EV2C; Appendix Fig. S8A,B). FFAST LC-MS/MS analysis of 19 FFAs from 5 different brain regions from the STXBP1$^{+/-}$ mouse revealed a significant decrease in the total FFA levels in heterozygous mice compared to WT animals (Fig. 9C; Appendix Fig. S9A,B). Further analysis of this data using PCA and partial least square discriminant analysis (PLSDA) of the average FFA profile across the brain showed that the heterozygous animals have a distinct FFA profile (Fig. 9D; Appendix Fig. S9C), largely driven by saturated FFAs (C14:0, C16:0 and C18:0) and to a lesser extent monounsaturated FFAs (C16:1 and C18:1) which were significantly decreased compared to WT animals (Fig. 9C–E; Appendix Fig. S9B,C). Together, our results point to a novel role for STXBP1-DDHD2 interaction in the regulation of the saturated brain FFAs involved in memory acquisition.

## Discussion

This study utilised a multidisciplinary approach to uncover the mechanisms underlying changes in the FFA landscape across key regions of the brain that are involved in memory formation. We used targeted FFAST lipidomics coupled with a reward-based long-term memory paradigm to demonstrate that the DDHD2 isoform of PLA1 is critical for driving saturated FFA responses to memory acquisition, with its knockout, a model for human spastic paraplegia, causing ablation of saturated FFAs and leading to the progressive decline in memory and neuromuscular performance. We report a novel interaction between DDHD2 and STXBP1—a key regulator of the exocytic machinery and demonstrate that STXBP1 controls the transport of DDHD2 to the plasma membrane and the subsequent generation of saturated FFAs (particularly myristic acid) in response to stimulation in neurons and neurosecretory cells. The activity-dependent generation of

saturated FFAs is solely driven at the synapse. Further, we show that it is DDHD2, rather than the DDHD1/PLD1 pathway, that mediates memory acquisition. Finally, we show that STXBP1 heterozygote mice, a model for early infantile encephalopathy, have markedly reduced saturated FFAs potentially contributing to the complex phenotype of the human disease including mental retardation and autism. Together this data demonstrates that DDHD2 regulates FFA changes that are associated with memory via interaction with STXBP1.

Consistent with previous studies which showed that saturated FFAs predominate the FFA response in both in vitro stimulation of neuronal cultures (Narayana et al, 2015) and during fear memory acquisition in vivo (Wallis et al, 2021), our observation that instrumental conditioning also induced changes in the saturated FFA landscape of the brain indicates that regardless of the paradigm used, localised activity-dependent increases in saturated FFAs in the brain are a general feature of memory acquisition. Instrumental conditioning induced the greatest FFA responses in the dorsal hippocampus and prefrontal cortex, both of which are involved in reward-based memory (Balleine, 2019; Balleine and O'Doherty, 2010). This is in contrast to fear conditioning where the highest FFA response was elicited in the amygdala, the known centre for emotion processing (Wallis et al, 2021). This strongly suggests that while the saturated FFA response to memory acquisition is consistent, the spatial distribution of the responses depends on the regions involved in each specific type of memory. In the case of instrumental conditioning, this involves the dorsal hippocampus, prefrontal cortex, and the posterior dorsomedial striatum which form a critical hub for the circuitry that encodes reward-based memory in the brain (Lex and Hauber, 2009). It is therefore expected that FFA changes indicative of synaptic plasticity would be concentrated in these areas. While the contribution of the cerebellum to non-motor function has been suggested (Timmann and Daum, 2007), the lower FFA response observed in this brain region may be ascribed to its primary involvement in motor processes (Statton et al, 2018) and consequently, its lower relevance to synaptic plasticity in response to instrumental conditioning. Although polyunsaturated fatty acids (PUFAs) such as AA, DHA, and eicosapentaenoic acid (EPA), previously associated with membrane fluidity (Fukaya et al, 2007), neuronal signalling and SNARE complex formation (Falomir-Lockhart et al, 2019; Garcia-Martinez et al, 2018; Rickman and Davletov, 2005), as well as memory (Inoue et al, 2019), also changed in our study, the overall response was dominated by saturated FFAs. The significant reduction in octanoic acid (C8:0), lignoceric acid (C24:0), and erucic acid (C22:1) which we observed across the different brain regions may be attributed to their energy-contingent usage or their metabolism as a precursor for generating

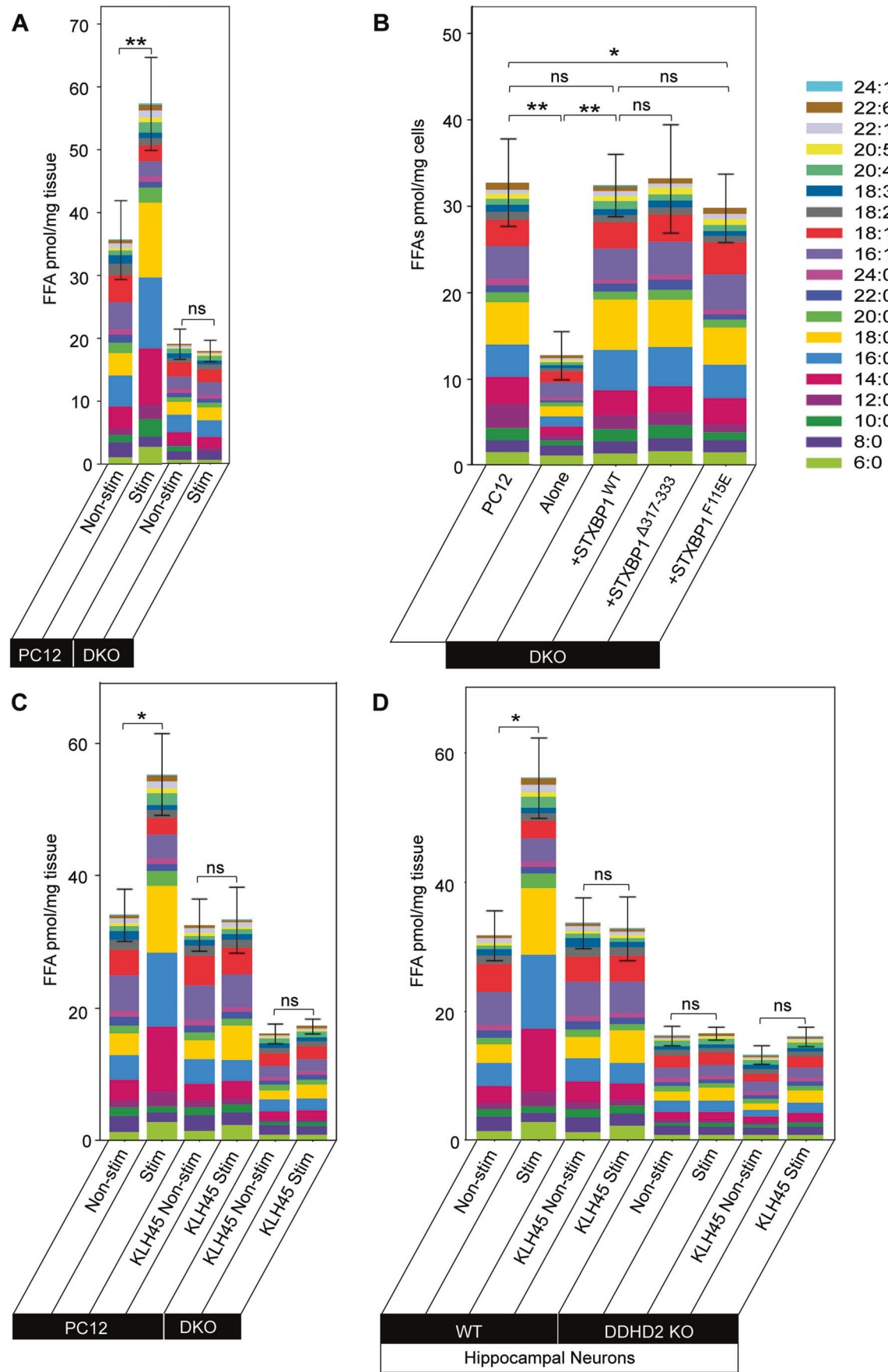

**Figure 8. STXBP1 (Munc18-1) controls the generation of saturated FFAs in vitro and in vivo.**

FFA abundance determined using FFAST, as detailed in Fig. 1. (A) Stacked bar plot showing profiles of FFAs in PC12 cells and STXBP1/2 double knockout (DKO) PC12 cells following stimulation. WT PC12 cells and DKO PC12 cells were stimulated (Stim) by depolarization for 15 min in high $K^+$ (60 mM) buffer. Control unstimulated (Non-stim) cells were treated for 15 min in low $K^+$ (2 mM) buffer. The significance of the change in average FFA abundance between PC12 Non-stim/Stim and DKO Non-stim/Stim as determined by $t$-test with Holm–Sidak post hoc correction is indicated by asterisks **$p < 0.01$. ns = not significant. (B) Basal FFAs levels in PC12 cells, STXBP1/2 DKO PC12 cells, DKO cells transfected with STXBP1$^{WT}$, DKO cells transfected with STXBP1$^{\Delta317-333}$ (Loop mutant), and DKO cells transfected with STXBP1$^{F115E}$ (hydrophobic pocket mutant) (pmol/mg tissue). (C) Profile of FFA responses to stimulation in PC12 cells following pharmacological inhibition of DDHD2 using KLH45 at 25 nM, for 4 h (±stimulation), and in DKO PC12 cells. (D) Profile of FFA responses to stimulation in DDHD2 WT hippocampal neurons following pharmacological inhibition of DDHD2 using KLH45 (±stimulation), and in DDHD2 KO hippocampal neurons. Data information: In (A–D), the significance of the change in FFA abundance between the different experimental conditions ($n = 3$ biological replicates) as determined by one-way ANOVA with Holm–Sidak post hoc correction is indicated by asterisks *$p < 0.05$, **$p < 0.01$, ns = not significant. Error bars represent the cumulative standard error of the mean (SEM) for all groups and parameters. Source data are available online for this figure.

other FFAs in response to memory acquisition (Andersen et al, 2021).

Looking into the mechanism by which these saturated FFAs are generated during memory acquisition, we demonstrated that the DDHD2 isoform of PLA1 is activated during these processes, supporting the idea that PLA1 plays a critical role in memory formation via the regulation of saturated FFA generation and dynamically modifying the phospholipid landscape of the brain (Higgs and Glomset, 1996; Inloes et al, 2014). The substrate–product relationship of DDHD2 activity is complex, as relatively few phospholipids contain a myristoyl (C14:0) acyl chain on the *sn-1* position. Most of the phospholipid fatty acyl chains are either C16:0 or C18:0, and although these FFAs also increase in response to conditioning, their levels are consistently lower compared to the amount of C14:0 generated. It is therefore reasonable to speculate that DDHD2 exhibits a relatively high level of selectivity for the C14:0 acyl chain phospholipids. Additional work will be needed to address this issue. Further, a decrease in unsaturated FFAs was detected in *DDHD2* knockout mice which could be attributed to the presence of non-canonical phospholipid substrates that do not have a saturated *sn-1* and an unsaturated *sn-2* acyl chain configuration (Thomas et al, 2006; Wang and Hsu, 2022). Alternatively, it may also stem from the reduced specificity of the PLA1 enzyme in cleaving both *sn-1* and *sn-2* positions of the fatty acyl chain of the phospholipid substrates. In line with this hypothesis, genetic dysregulation affecting DDHD2 expression (Inloes et al, 2014) and its localisation, greatly impacts saturated FFA levels and memory acquisition. We further found that the DDHD1/PLD1 pathway was not involved in generating saturated FFAs in response to memory acquisition as PLD1 ablation did not affect contextual fear memory acquisition, in good agreement with previous studies (Santa-Marinha et al, 2020) and had limited effect on the saturated FFAs response. Our findings therefore add weight to the theory that saturated FFAs are critical in memory formation and contribute to the mechanism(s) underpinning the retention of memory and cognitive function (Zamzow et al, 2019). In this view, it is worth considering that saturated FFAs are produced metabolically in contrast to many PUFAs that require dietary supplementation, and it seems unlikely that evolution would have selected an externally derived nutrient for such a critical function.

Beyond establishing a link between saturated FFA levels and brain function, our study provides relevant insights into the onset and progressive impairment of memory performance in $DDHD2^{-/-}$ mice. The discovery of a trend in memory impairment coupled with changes in the brain FFA landscape of otherwise asymptomatic 3mo mice, prior to the onset of a decline in motor function at 5mo,

suggests that impairment in memory function precedes the onset of motor dysfunction in $DDHD2^{-/-}$ mice. The impairment of memory performance associated with the DDHD2 knockout reported in our study could, at least in part, explain the intellectual disability associated with HSP, and recent studies showing that ablation of the *DDHD2* gene alters neural processing (Inloes et al, 2014; Joensuu et al, 2020b; Richmond and Smith, 2011) also strongly advocate for the significant impact of saturated FFAs in synaptic function and plasticity.

Our observed decline in motor function corroborates earlier reports associating disruption of the *DDHD2* gene with classical signs of slowly progressing spastic paraparesis characterised by weakness, hyperreflexia, neuromotor dysfunction predominantly affecting the lower limbs, and ultimately aberrant gait (Blackstone, 2018; Inloes et al, 2014; Inloes et al, 2018; Joensuu et al, 2020b; Parodi et al, 2017). The observation that deficits in motor coordination preceded decline in motor strength suggests that the premotor cortex, which is responsible for some aspects of motor control including preparation for movement and sensory guidance of movement, may be impacted more significantly or earlier than the primary motor cortex, which mainly contributes to the generation of neural impulses that control the execution of movement (Cisek and Kalaska, 2005). Hence, motor coordination may be a preferable diagnostic test for the early detection of HSP. The observed motor deficit may be attributable to changes in the FFA landscape of the cerebellum in genetically ablated *DDHD2* animals (Janssen et al, 2015). Notably, the observation of no significant difference in the ambulatory distance of $DDHD2^{+/+}$ and $DDHD2^{-/-}$ mice, further suggest that the impairment in motor function was not severe enough to impact on the ability of the mice to press the lever, and hence the reduced lever presses can only be attributed to memory deficits.

Another important aspect of our study was the discovery that DDHD2 strongly interacts with STXBP1/Munc18-1 suggestive of a potential synaptic targeting effect. Indeed, we found that STXBP1 controls the transport of DDHD2 to the plasma membrane and the generation of saturated FFAs. This effect is similar to that reported for the transport of syntaxin1A, a SNARE protein involved in neuroexocytosis, by STXBP1 (Han et al, 2011; Han et al, 2009; Malintan et al, 2009; Medine et al, 2007; Rickman et al, 2007). Using a neurosecretory cell line engineered to knock out STXBP1/2 we demonstrated that DDHD2 is unable to reach the plasma membrane, suggesting a block in the anterograde secretory pathway. In support of this, DDHD2 was found at the synapse of WT neurons, and DDHD2 knockout neurons exhibited significant ERGIC disruption with a knock-on effect on the Golgi apparatus

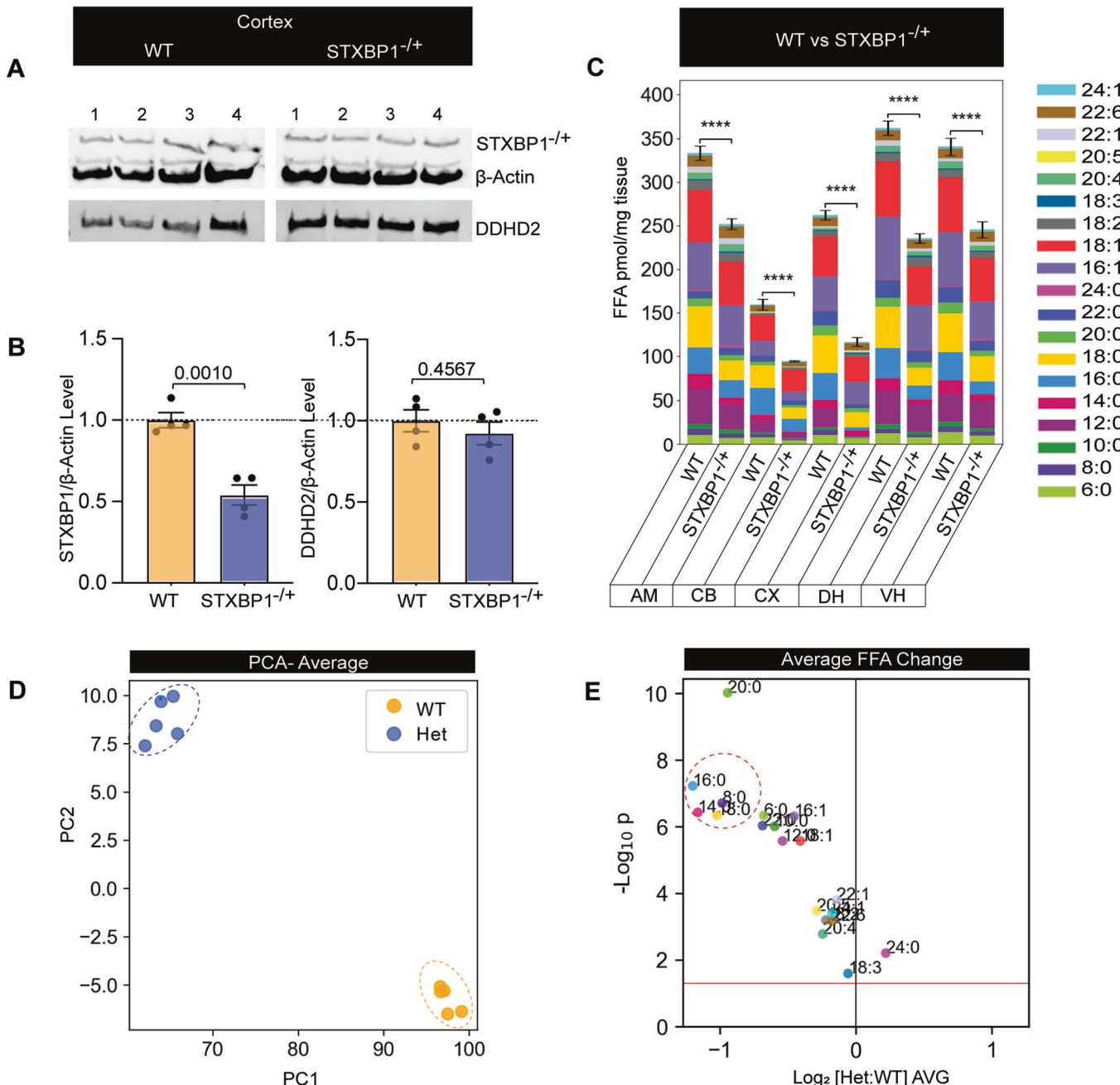

**Figure 9. Profile of STXBP1 protein expression levels and FFA response in STXBP1 haploinsufficiency mouse brain.**

Western blotting analysis from WT and STXBP1 heterozygous (STXBP1$^{-/+}$) mouse brains to assess expression levels of STXBP1 and DDHD2. (A) Western blot of cortical protein extracts blotted using anti-Mouse STXBP1 and anti-β-Actin (top), as well as Rb-DDHD2 (bottom), are shown. (B) Analysis of the relative protein level of STXBP1 or DDHD2 (relative to β-Actin) was analysed from cortical samples. Values are presented as mean ± SEM. Student's $t$-test was used to compare protein expression from WT to STXBP1$^{-/+}$ brains, $N = 4$ mice each. (C) Bar plots showing the total FFA levels across different brain regions (CB; cerebellum, CX; cortex, DH; dorsal hippocampus, VH; ventral hippocampus, AM; amygdala) of WT versus STXBP1$^{-/+}$ (KO) mice. (D) Principal component analysis of the average FFA profile across the brain. (E) Volcano plot showing average response across the brain of individual FFAs in WT versus KO mice. Data information: In (B), values are presented as mean ± SEM. Student's $t$-test was used to compare protein expression from WT to STXBP1$^{-/+}$ brains, $n = 4$ mice each. In (C), the significance of the change in FFA abundance between the different experimental conditions ($n = 5$ biological replicates) as determined by one-way ANOVA with Holm–Sidak post hoc correction is indicated by asterisks, ****$p < 0.0001$. In (E), each dot on the volcano plot represents the average change in abundance of a single analyte across 5 measured brain regions. Analytes below the red line represent those whose change in abundance was not statistically significant (two-tailed $t$-test $p > 0.05$). Error bars represent the cumulative standard error of the mean (SEM) for all groups and parameters. Source data are available online for this figure.

and ultimately a reduced number of synaptic vesicles. The latter could stem from such anterograde transport defects but could arise from altered synaptic vesicle recycling. Further work will be needed to address this point.

Furthermore, STXBP1/2 knockout reduced basal FFA levels and abolished the activity-dependent saturated FFA response. STXBP1 also controls priming of synaptic vesicles, however, our demonstration that FFA responses could be rescued using WT STXBP1 and two STXBP1 mutants known to affect vesicular priming indicates that the STXBP1 control of saturated FFA responses is independent of vesicular priming.

We show that STXBP1 targets DDHD2 to the presynapse, where it dynamically modifies the lipid landscape to generate saturated FFA metabolites during memory formation. Consistent with a major role of saturated FFAs in learning and memory, we also found that haploinsufficient STXBP1 mice have greatly reduced saturated FFAs which may potentially explain their poor cognitive performance (Chen et al, 2020) and that of STXBP1 encephalopathy patients (Abramov et al, 2021; Lanoue et al, 2019; Stamberger et al, 2016). This finding also suggests the possible involvement of saturated FFAs in the pathophysiological mechanisms underlying DEE resulting from STXBP1 mutations. Further, previous studies have shown that mutations in the *DDHD2* gene regulate Golgi/ER membrane trafficking and further support its critical role in essential cellular processes associated with memory function. The observation of arrested vesicles along the Golgi/ER interface of hippocampal neurons from *DDHD2*$^{-/-}$ mice is suggestive of a transport defect which may also be contributing to the memory deficit observed in these animals. This result corroborates previous reports which used electron microscopy to detect an accumulation of lipid droplets and multilamellar bodies which can alter autophagy-lysosome function, and render the cell more susceptible to apoptosis, subsequently resulting in memory dysfunction (Garcia-Sanz et al, 2018). Our data indicate that the interplay between STXBP1 and DDHD2 is critical for anterograde transport and presynaptic function in generating saturated FFAs. Disruption of either player results in altered FFA response and memory defects.

Glial cells are the predominant cell type in the mammalian brain, contributing 50–60% of the brain's total mass, and are critical in maintaining the function and integrity of the surrounding neurons. This view is supported by the findings of closely-tied spatial proximity (Perea et al, 2014), bidirectional communication (Verkhratsky, 2010), and neurometabolic coupling of astrocytes with neurons at the pre- and post-synapse (Dienel, 2017; Lee et al, 2021). Through this association, the various perineural glial subtypes function to maintain the critical physiology and firing ability of the adjoining neurons (Heller and Rusakov, 2017) by providing the requisite structural and metabolic support, homeostatic balance, and defence against oxidative stress (Allen and Eroglu, 2017; Siracusa et al, 2019), consequently increasing the number of mature, and functional synapses on the neurons (Pfrieger, 2010; Pfrieger and Barres, 1997; Ullian et al, 2001). Although some studies have suggested a possible lipid-mediated interaction involving neurons and glial cells, there is limited information on the lipid metabolic interaction between the two cell types. We confirmed that the synapse is the locus for FFA production by culturing neurons from multiple brain regions and demonstrating that stimulation elicited a clear saturated FFA

increase in all cases. Further, inhibiting glial proliferation in these cultures slightly impacted the basal FFA levels, however, the activity-dependent increase in FFAs remained unaltered.

This suggests that while glia, particularly astrocytes, are involved in lipid metabolism in the brain, they are not contributing to the FFA responses driving memory.

Although FFAs can mediate the process of memory consolidation via several mechanisms including the modulation of membrane properties, post-translational targeting of proteins to interact with membranes and other proteins, or via other lipid signalling pathways, how these saturated FFAs affect synaptic function is currently unknown. It is likely that protein acylation occurring via acyl-CoA intermediates is a key player in this process (Seo et al, 2022). Considering that myristic and palmitic acids are highly increased in response to memory acquisition, the protein lipidation driven by increased substrate availability could contribute to the establishment of synaptic plasticity. More work is also needed to assess this important question.

Together, our findings demonstrate for the first time that the interaction between DDHD2 and STXBP1 is critical for long-term memory by regulating the generation of FFAs at the synapse. Consequently, DDHD2 may be an important pharmacological target to alter saturated FFAs in the context of memory in ageing and dementia. A better understanding of the DDHD2-regulated lipid pathways may offer critical insights into the mechanisms of synaptic plasticity and therapeutic strategies for cognitive disorders.

# Methods

## Ethical considerations and animals

For all experimental procedures, the care and use of animals was carried out in line with the protocols approved by the Animal Ethics Committee of The University of Queensland (2017/AE000497, 2018/AE000508, 2021/AE000971, 2020/AE000352 and 2022/AE00073) and by the Institutional Animal Care and Use Committee at Baylor College of Medicine (protocol AN-6544).

## Key resources

HPLC/analytical grade reagents were used throughout. 1,1-carbonydiimidiazole, triethylamine, iodomethane, iodomethane-d3, iodoethane, iodoethane-d5, iodopropane, formic acid, citric acid, methanol, disodium hydrogen phosphate, chloroform, ammonium formate, acetonitrile, 1-butanol, and analytical standards for saturated and unsaturated fatty acids were purchased from Sigma-Aldrich. All lipid extractions were performed in 2 mL polypropylene LoBind safe-lock tubes (Eppendorf).

Cell culture reagents were purchased from Life Technologies. Mouse anti-STXBP1 antibody was purchased from BD Biosciences. pCMV-STXBP1-emGFP, pCMV-STXBP1, NPY-hPLAP, and NPY-mCherry were prepared as previously described (Arunachalam et al, 2008; Martin et al, 2013; Tomatis et al, 2013). STXBP1$^{\Delta317\text{-}333}$ and STXBP1$^{F115E}$ mutants were made using the quick-change lightning site-directed mutagenesis kit (Strategene, USA) and the mutational primer

5′-GACTTTTCCTCTAGCAAGAGGATGATGCCCCAGTAC-CAGAAGGAGC-3′, as previously described (Martin et al, 2013).

All constructs were sequenced at The Australian Genome Research Facility, located at The University of Queensland.

## Experimental model and subject details

*DDHD2*$^{-/-}$ mice generated in a C57BL/6 background using standard gene targeting techniques (Inloes et al, 2014) were sourced from the Scripps Research Institute in the United States. The animals were maintained on a 12 h/12 h light/dark (LD) cycle at between 21–22 °C and housed in duos with access to standard mouse chow (in Dresden: Ssniff R/M-H; catalogue # V1534 and in Brisbane: Specialty Feeds, catalogue # SF00-100) and ad libitum autoclaved water.

*STXBP1*$^{+/-}$ mice were generated as described previously (Chen et al, 2020) and housed in an Association for Assessment and Accreditation of Laboratory Animal Care International-certified animal facility at Baylor College of Medicine on a 14 h/10 h LD cycle. All procedures to maintain and use mice were performed in strict accordance with the recommendations in the Guide for the Care and Use of Laboratory Animals of the National Institutes of Health and were approved by the Institutional Animal Care and Use Committee at Baylor College of Medicine (protocol AN-6544).

## Behavioural experiments

### Instrumental conditioning test

#### Animals
All behavioural and lipidomics experiments were performed using a cohort of age- and sex-matched *DDHD2*$^{-/-}$ and their *DDHD2*$^{+/+}$ litter mates (3 and 12 months old, C57BL/6 background), while all other experiments were carried out using neurons from *DDHD2*$^{-/-}$ and C57BL/6 WT animals (Inloes et al, 2014) housed in 12 h/12 h LD environment with diet restricted to 85% of their free-feeding weight. Every effort was made to minimize the number of animals used and their suffering. A total of 80 animals were used (20 mice per group) for behavioural experiments, while 6 animals were sacrificed per group, with brain samples collected for lipidomics analysis.

PLD1$^{-/-}$ mice were originally generated through the removal of exons 13 (that codes for the first HKD motif for PLD1) and 14, using a Cre-lox/FLP-FRT recombination system, as previously described (Dall'Armi et al, 2010) at the Université de Bordeaux, France. Littermate mice were used for the experiments which were of age 3–6 months old and had C57BL6 background housed in 12/12 LD with ad libitum feeding. Every effort was made to minimize the number of animals used and their suffering. The experimental design and all procedures were in accordance with the European guide for the care and use of laboratory animals and the animal care guidelines issued by the animal experimental committee of Bordeaux Universities (CE50; A5012009).

#### Instrumental conditioning apparatus
Instrumental conditioning training procedures occurred in sound and light-resistant operant chambers (MED Associates) as previously described (Matamales et al, 2017). In brief, the chambers were illuminated with a 3 W, 24 V house light. Each chamber

contained a recessed feeding magazine in the centre of the chamber wall on one end. The magazine was connected to two pellet dispensers that could individually deliver grain and purified pellets (20 mg Dustless Precision Pellets; Bioserve Biotechnologies, 3.35 kcal/g) into the magazine when activated. Either side of the magazine contained retractable levers. Med-PC (MED Associates) software was used to direct the insertion and retraction of the levers, illumination of the chamber, and delivery of the pellets. This software also recorded the number of lever presses, magazine entries, pellets delivered and duration of the experiment (30 min). Activity was monitored using a camera and D-View Cam Software (D-link Corporation; Taiwan).

#### Behavioural procedures
Behavioural training of the mice was carried out as previously described (Matamales et al, 2017). In brief, the mice were trained to press a lever (left or right) to obtain a food reward (20 mg Dustless Precision Pellets; Bioserve Biotechnologies, 3.35 kcal/g). A total of 10 mice were trained per session and two conditions were used: instrumental and control. The instrumental animals were presented with a lever and a reward contingent with the number of lever presses, while the control animals were only exposed to the box but not presented with a lever and there was no reward delivery. Each session ran for 30 min or until the animals earned 20 rewards (whichever came first).

#### Magazine training
Each mouse (*DDHD2*$^{+/+}$ and *DDHD2*$^{-/-}$) was allotted to a specific operant chamber, which was maintained throughout the experimental period. For magazine training during the first 3 days (days −1 to −3), the instrumental group animals received a maximum of 20 pellets per mouse for the 30 min period. The chamber was illuminated to indicate the onset of the session and extinguished at the expiration. Following the training, both levers were retracted, and the mouse allowed to freely explore the chamber.

#### Instrumental training
Instrumental mice were trained on a continuous reinforcement schedule (CRF) where the delivery of the rewarding outcome was contingent on the corresponding lever-pressing action (one reward delivered for each lever press) for first 3 days of instrumental training (days 1–3). Thereafter, a random ratio (RR) schedule was initiated to reduce the chance of obtaining a reward. The mice were trained on RR5 for 3 days (outcome delivery for each action was at a probability of 0.2; training days 4–6), then RR10 for 3 days (probability 0.1; training days 7–9), and lastly RR20 for five days (probability 0.05; training days 10–14). Control mice were not presented a lever, and hence, no reward was delivered. After the 17 days of training, the memory performance for each mouse was ascertained by computing the lever press rate (lever presses per minute).

#### Fear conditioning
Mice were housed individually in a ventilated area before the start of behavioural training. To avoid excessive stress during the experiments, animals were handled every day before the start of the experiment during a week. On day 1, animals were transferred to the conditioning context (context A) for habituation. On day 2, we proceeded with the conditioning phase. The protocol typically

consisted of 3× foot shock of 0.6 mA for 2 s with 60 s time interval between shocks. Discriminative contextual fear memory was tested 24 h after conditioning by analysing the freezing levels in context B vs. context A (Tests B/A). Freezing behaviour was quantified automatically in each behavioural session using a CCD camera connected to automatic freezing detection software (Ugo Basile, Italy). To test for animal exploration and activity, the animal displacement in the context was traced and analysed with software programmed and provided by Dr. Jiyun Peng (Fudan University, Shanghai, China).

### Novel object location (NOL) test

The NOL paradigm is a variant of the novel object recognition (NOR) test (Lueptow, 2017) which affords assessment of spatial memory based on the ability of the animal to encode both the object's feature as well as the spatial location information of the object (Massey et al, 2003). In this paradigm, all mice were habituated to the room as well as to two identical objects placed in opposite locations of a white box (30.5 × 30.5 cm) for 10 min. After 24 h, one of the objects was moved to a novel location and the mice were allowed to explore the objects in the different location for 10 min while being videoed. The time that the mice spent exploring the NOL (discrimination index), defined as exploration with the nose, <1 cm from the object, was quantitatively assessed using software (Ethovision) (Blackmore et al, 2022).

### Activity monitoring test

Using an automated activity monitoring device (Med Associates), we assessed open-field locomotor performance, alongside explorative and vigilance activity. Briefly, each mouse was placed in the centre of an arena (40 × 40 × 30 cm) and allowed to freely explore in the presence of ~70-lux illumination throughout the testing period. Parameters including vertical time, resting time, jumping time, vertical counts, jump counts, and ambulatory distance, were recorded over a 30-min trial period (Fan et al, 2023; Pandit et al, 2019).

### Grip strength test

The grip strength for both forelimb and hind limb was assessed monthly (between 3 and 12 months of age) using a Digital Force Gauge (Ugo Basile, System; San Diego Instruments Inc., San Diego, CA). Each mouse was held close to the base of its tail and placed on the rectangular bar of the grip strength device, ensuring that all four paws made contact before gently pulling at a consistent angle, to automatically record the peak force in Newtons before the animal's grip was broken. A total of 10 trials per mouse were carried out and the mean peak force of the 10 trials was used for subsequent analysis (Pandit et al, 2019).

### Motor coordination test

Motor coordination was assessed monthly using an accelerating Rota-Rod device (Ugo Basile, Comerio, Italy) by determining the latency period from when the mouse is placed on the accelerating rotating rod device to the initial fall (the length of time the mouse lingered on the rolling rod). A total of three trials was carried out for each mouse, at an acceleration mode of 5 to 30 rpm over 60 s. For the three individual trials, the latency to the first fall was averaged and used as the mean latency, while a ceiling latency of 180 s was recorded if the mouse persisted beyond this period (Pandit et al, 2019).

### Tissue collection

To determine the FFA response, ~4 h following instrumental conditioning the mice were sacrificed under deep anaesthesia (intraperitoneal injection of 0.2 ml phenobarbitol (300 mg/ml)) and transcardially perfused with ice-cold phosphate-buffered saline (PBS) pH 7.4 to flush out blood and chill the brain prior to snap freezing in liquid nitrogen. Tissue was then cryodissected from 5 different frozen brain regions (Cortex—CX, ventral hippocampus—VH, dorsal hippocampus—DH, amygdala—AM, and cerebellum—CB) to minimize post-mortem and ischemic lipid metabolism (Wallis et al, 2021). The animals were swiftly sacrificed, and their brains were excised, carefully wrapped in aluminium foil and snap-frozen in liquid nitrogen (subsequently stored at −80 °C). Frozen brains were mounted on the block of a cryostat (NX70, Thermo Fisher Scientific) running at −20 °C, where 80–100 μm thick tissue slices were cut and transferred to glass slides (Merck; S8902). Slides were then transferred to an Olympus SZ51 microscope next to the cryostat, and suitable brain areas were quickly dissected from each frozen slice and placed on dry ice in 2 ml tubes (Eppendorf). Each brain was split into 70 slices on average. Material was dissected from roughly 20 sequential brain slices in each tube, which represented a specific brain region. Until the lipid extraction, all samples were kept at −80 °C. STXBP1$^{+/-}$ mouse tissues were treated in a similar way and the extraction operations were also carried out in frigid temperatures to minimize ischemic and post-mortem lipid metabolism, and at no stage prior to this were brain samples allowed to thaw.

### Synthesis of FFAST isotopic-coded differential tags

From commercially available $CH_3I/CD_3I/C_2H_5I$ and 3-hydroxymethyl-pyridine reagents, 3-hydroxymethyl-1-methylpyridium iodide (FFAST-124), 3-hydroxymethyl-1-methyl-d3-pyridium iodide (FFAST-127), and hydroxymethyl-1-ethylpyridium iodide (FFAST-138) tags were produced. The synthesis of FFAST derivatives was carried out as described previously (Koulman et al, 2009; Narayana et al, 2015). In brief, 200 mg of iodomethane, iodomethane-D3, and iodoethane were each mixed with 100 mg of 3-hydroxy-methyl-pyridine. Under gaseous $N_2$, the resulting solution was heated to 90 °C and microwaved for 90 min at 300 W in a CEM Discover microwave reactor. The resultant solution was then dried after being rinsed with 100% diethyl ether. Using 1H nuclear magnetic resonance (NMR) spectroscopy, the purity of all the derivatives was determined to be >95%.

### Free fatty acid (FFA) extraction and FFAST labelling of brain tissue

In the cold room, frozen brain tissue samples were homogenised for 5 min in 0.5 mL of HCl (0.1 M) using a tissue homogenizer ultrasonic processor (Vibra-Cell, Sonics Inc., USA). 200 mL homogenate was treated with 0.6 mL ice-cold chloroform and 0.4 mL ice-cold methanol:12 N HCl (96:4 v/v, supplemented with 2 mM $AlCl_3$). After the mixture had been thoroughly vortexed, 0.2 mL ice-cold water was added, and tubes centrifuged at 4 °C for 2 min at $12,000 \times g$ in a refrigerated microfuge (Eppendorf 5415 R). The upper phase was discarded, and the lower phase tube was dried in a vacuum concentrator (Genevac Ltd). Dried extracts were redissolved in 100 μL acetonitrile. The derivatization strategy was designed for molecules with free carboxylic acid groups and followed a previously published procedure (Narayana et al, 2015).

Briefly, 100 μL of FFA extracts in acetonitrile were combined with 50 μL of 1,1-carbonyldiimidiazole (1 mg/mL in acetonitrile) and incubated at room temperature (RT) for 2 min. Following this, 50 μL of either FFA extracts tagged with FFAST-124 or FFAST-127 (50 mg/mL in acetonitrile, 5% triethylamine) was added. The combinations were then mixed for 2 min before heating for 20 min at 50 °C in a water bath. Finally, 100 μL of each isotopically labelled sample was combined and dried in a vacuum concentrator. They were then redissolved in 200 μL of an internal standard solution (2.5 μM in acetonitrile) made by derivatizing the 19 FFA standards with the FFAST-138 label. The samples were then placed in an auto-sampler vial and analysed using liquid chromatography tandem mass spectrometry (LC-MS/MS). Prior to analysis, all samples were maintained at −20 °C.

## FFA LC-MS/MS analysis

LC-MS/MS analysis was performed on a Shimadzu Nexera UHPLC equipped with an Poroshell 120 CS-C18, 2.1 × 100 mm column with 2.7 μm particle size, linked to an AB Sciex 5500QTRAP tandem mass spectrometer fitted with an ESI Turbo V source. Analyst ® 1.5.2 software (AB Sciex) was used for instrument control data collection while Multiquant software (AB Sciex) was used for the multiple reaction monitoring (MRM) data analysis. Chromatographic separations were performed on 1 μL sample injection volume using a gradient system consisting of solvent A (0.1% formic acid (v/v)) and solvent B (100% acetonitrile with 0.1% formic acid (v/v)) was used to perform liquid chromatography (LC) at 0.450 mL/min heated to 60 °C). The gradient conditions consisted of an initial isocratic step of 15% B for 1 min followed by a gradient to 100% B over 9 min. The column was flushed at 100% B for 2 min and then reduced back to 15% to re-equilibrate for 2 min for a total run time of 14 min. The first 0.5 min of the LC run was switched to waste to remove any excess underivatized FFAST tags.

Mass spectrometric data acquisition was performed using positive mode ionisation in MRM mode (Appendix Tables S1–S3). Ion source temperature was set at 400 °C, and ion spray voltage set to 5500 V. The source gases setting consisted of curtain gas, GS1 and GS2 set to 45, 40 and 50 psi, respectively. The collision energy, declustering potential, and collision cell exit potentials were all set at 50, 100, and 13 V, respectively, for all transitions.

## Phospholipid extraction

Frozen brain tissue samples were homogenized with five volumes of cold citrate buffer (30 mM) containing 1.25 μg/mL internal standard mixture (PC 17:0/17:0, PE 17:0/17:0, PS 17:0/17:0, PA 17:0/17:0 and PG 17:0/17:0) for 5 min using a tissue homogenizer in the cold room. Phospholipid extraction was performed essentially according to the well-established method of Bligh and Dyer (Bligh and Dyer, 1959). In brief, to 100 μL of the homogenate, 0.9 mL of ice-cold chloroform: methanol (1:2 v/v) was added and vortexed for a minute. To this, 0.3 mL of chloroform followed by 0.3 mL of 0.1 N HCl were added. The mixture was vortexed and incubated for a minute and then centrifuged for phase separation at 10,000 rpm in a benchtop microfuge for 5 min at 4 °C. The lower organic phase was collected and stored at −80 °C for further analysis.

## Phospholipid LC-MS/MS analysis

Analysis was performed on the Shimadzu Nexera UHPLC/5500QTRAP tandem mass spectrometer system described above. Chromatographic separation was performed on a Kinetex 2.6 μm HILIC 100 Å column (150 × 2.1 mm) (Phenomenex, USA). The instrument was operated in negative ion mode under multiple reaction-monitoring conditions. The Turbospray temperature was set to 300 °C, the curtain gas flow to 30 psi, and the ion spray voltage was −4500 V. The collision energy (CE) was set to −50 for all phospholipid species. The declustering potential (DP) and collision cell exit potential were optimized and were set to 100 and 13, respectively. The precursor masses of each phospholipid species in every class were selected as Q1 and the fatty acyl chains generated in negative ion mode were selected as Q3, respectively (Appendix Tables S1–S3). HILIC chromatography elution conditions were set at 400 μl/min, using a gradient system consisting of solvent A (95% acetonitrile with 5% 50 mM ammonium formate pH 3.55) and B (50% acetonitrile, 45% water with 5% 50 mM ammonium formate pH 3.55). After equilibration with 95% B, the gradient elution used was 95–85% B (1 min), 85–70% B (10 min), 50% B (2 min) and 50–95% B (1 min). The column was equilibrated at 95% B for 10 min before the next run. The injection volume of all the samples was 5 μl. The abundance of the species within each phospholipid class was determined relative to a single standard for each class. A 10 μg/mL standard mixture of PC 12:0/13:0, PE 17:0/14:1, PS 17:0/14:1 and PA 12:0/13:0 was prepared in Mobile phase A. They were serially diluted into 6 steps to prepare calibrators (0.15 μg, 0.312 μg, 0.625 μg, 1.25 μg, 2.5 μg and 5 μg/mL). The lower limit of detection (LLOD) and lower limit of quantitation (LLOQ) was up to 0.15 μg/mL and 0.60 μg/mL on column for all monitored PLs. To each calibrator, an internal standard mixture containing PC 17:0/17:0, PE 17:0/17:0, PS 17:0/17:0 and PA 17:0/17:0 was added to a final concentration of 1.25 μg/mL. Calibration lines were analysed as triplicates ($n = 3$) and a standard curve for each phospholipid was plotted as the ratio between analyte and internal standard peak area ratio of the same lipid class. Calibration curves were calculated by linear regression without weighting.

## Cell cultures

PC12 cells were cultured in Dulbecco's modified Eagle's media (DMEM) supplemented with glucose (4500 mg/L) and containing 30 mM NaHCO₃, 5% foetal bovine serum, 10% horse serum, 100 units/ml penicillin/streptomycin, and 100 g/ml Kanamycin, and maintained as described previously (Han et al, 2009; Martin et al, 2013). STXBP1/2 DKO PC12 cells were created using CRISPR (Kasula et al.). Production of STXBP1$^{F115E}$, STXBP1$^{Δ317-333}$ and STXBP1$^{WT}$ recombinant proteins were produced as previously described (Christie et al, 2012; Deak et al, 2009; Kasula et al, 2016; Martin et al, 2013).

## Immunofluorescence microscopy

PC12 and STXBP1 DKO PC12 cells were transfected with STXBP1$^{WT}$, STXBP1$^{Δ317-333}$, and STXBP1$^{F115E}$ constructs and immunolabelling was carried out as previously described (Malintan et al, 2009; Martin et al, 2013). Following permeabilization of cells using 0.1% Triton X-100, cells were imaged using a Zeiss LSM510 confocal microscope.

C57BL6/J and DDHD2$^{-/-}$ hippocampal neurons at DIV21 were stimulated with High K$^+$ buffer or incubated in Low K$^+$ buffer (non-stimulated condition), for 5 min. Cells were then fixed in 4% paraformaldehyde, 0.1 mM MgCl$_2$, 0.1 mM CaCl$_2$ in PBS for 20 min at room temperature (RT). Cells were washed in PBS and permeabilized with 0.1% Triton X-100 for 4 min. After permeabilization, non-specific binding sites were blocked with 1% BSA in PBS for 30 min at RT. Primary and secondary antibodies were applied in blocking buffer (1% BSA in PBS) for 1 h and 30 min at RT, respectively. The following antibodies were used: anti-DDHD2 (1:100; ThermoFischer, 25203-1-AP), anti-DDHD1 (1:100; Novus biologicals, NBP2-13903), anti-Munc18-1 (STXBP1 1:100; BD Transduction Laboratories, 610336), anti-syntaxin (1:100: Origene, OTI2D11), and anti-SV2A (1:100; Abcam, ab77177).

## Secretagogue stimulation of PC12 and DKO PC12 cells

Cells transfected with STXBP1$^{WT}$, STXBP1$^{\Delta317\text{-}333}$, or STXBP1$^{F115E}$ constructs tagged with emGFP were washed with Buffer A (145 mM NaCl, 1.2 mM Na$_2$HPO$_4$, 20 mM HEPES-NaOH, 2 mM CaCl$_2$, 10 mM glucose, pH 7.4) and stimulated with either 60 mM KCl or 2 mM KCl and incubated for 15 min at 37 °C. Cells were collected, and FFAs were extracted using chloroform-methanol as described above. The FFAs were redissolved in 100 μL of acetonitrile.

## Immunoprecipitation (pull-down assay)

PC12 and STXBP1 DKO PC12 cells were maintained in DMEM (containing sodium pyruvate; Thermo Fisher Scientific), foetal bovine serum (7.5%, Gibco) and horse serum (7.5%, Gibco), and 0.5% GlutaMax (Thermo Fisher Scientific) at 37 °C and 5% CO$_2$. Cells were transfected using Lipofectamine®LTX with Plus Reagent (Thermo Fisher Scientific) as per the manufacturers' instructions with 2 μg of GFP-STXBP1 and pEGFP-C1-DDHD2$^{WT}$; respectively. Non-GFP transfected wild-type cell lines were prepared in parallel as a process control. 48 h post-transfection, PC12 cells were incubated for 5 min in isotonic buffer A (145 mM NaCl, 5 mM KCl, 1.2 mM Na$_2$HPO$_4$, 10 mM D-glucose, and 20 mM Hepes, pH 7.4) or stimulated with 2 mM BaCl$_2$ buffer A for 5 min at 37 °C. Cells were homogenised in ice-cold lysis buffer (10 mM Tris/HCl pH 7.5; 150 mM NaCl; 0.5 mM EDTA; 0.5% NP-40, and EDTA-free protease inhibitor cocktails; Roche) for 15 min on ice and the cell lysate was centrifuged at 13,000 RPM for 15 min, the supernatant was transferred to a new tube containing 25 μl of anti-GFP magnetic beads and equilibrated twice with washing buffer (10 mM Tris/HCl pH 7.5; 150 mM NaCl; 0.5 mM EDTA). During the tumble end-over-end rotation for 1 h at 4 °C, GFP-STXBP1 or pEGFP-C1-DDHD2$^{WT}$ is captured by the anti-GFP magnetic beads. The magnetic beads were then washed three times in washing buffer and magnetically separated until supernatant became clear and all unbounded protein completely washed out. Bead-bound samples for Western Blot analysis were resuspended in 100 μl 2X SDS-sample buffer (120 mM Tris/HCl pH 6.8; 20% glycerol; 4% SDS, 0.04% bromophenol blue; 10% β-mercaptoethanol), whereas sample for LC-MS/MS proteomics analysis were suspended in SDS-PAGE buffer with the following modifications buffer (50 mM Tris-HCl pH 6.8, 10 mM dithiothreitol, 2% w/v sodium dodecyl sulphate, 10% v/v glycerol). Samples were then boiled for 10 min

at 95 °C to dissociate the formed immunocomplexes from the beads and then magnetically separated to remove to beads from the protein suspensions. Samples were stored at −20 °C until further analysis.

## Immunocytochemistry

PC12 cells and STXBP1 DKO PC12 cells were stimulated with 2 mM of BaCl$_2$ buffer A for 5 min and fixed in 4% paraformaldehyde for 30 min at RT. Cells were washed in PBS containing 0.2% BSA and permeabilized with 0.1% Triton X-100 for 4 min. After permeabilization, non-specific binding sites were blocked with 1% BSA in PBS for 30 min at RT. Primary and secondary antibodies were applied in blocking buffer (1% BSA in PBS) for overnight (4 °C) and 1 h (RT); respectively. The following primary antibodies were used: anti-DDHD2 (1:50), anti-STXBP1 mouse monoclonal antibody (1:50) and anti-syntaxin mouse monoclonal antibody (1:50). Secondary antibodies: Alexa Fluor® 488 conjugated anti-mouse IgG, Alexa Fluor® 546 conjugated anti-guinea pig IgG, Alexa Fluor® 546 conjugated anti-rabbit IgG and Alexa Fluor® 647 conjugated anti-rabbit IgG, were diluted 1:1000 in blocking buffer.

## Imaging and image analysis

Confocal images were acquired with a spinning-disk confocal system consisting of an Axio Observer Z1 equipped with a CSU-W1 spinning-disk head, ORCA-Flash4.0 v2 sCMOS camera, 63× 1.3 NA C-Apo objective. Image acquisition was performed using SlideBook 6.0. Sections of each slide were acquired using a z-stack with a step of 100 nm. The exposure time for each channel was kept constant across all imaging sessions. Images were deconvolved on Huygens deconvolution software 21.10 (Scientific Volume Imaging). Both channels are thresholded using Costes method before calculating the Pearson's coefficient. For individual cells for rescues, the Costes method was applied only on the region of interest (ROI) corresponding to the cell (based on GFP-STXBP1 signal).

All images of hippocampal neurons in culture were acquired under the same conditions and processed the same way in the imaging facility of the Queensland Brain Institute. The immuno-fluorescently labelled neurons were acquired with an Olympus spinSR10 confocal microscope, using a UPLXAPO60XO (NA 1.42, WD 0.15 mm) objective, coherent OBIS lasers (405 nm LX 50 mW, 488 nm LS 150 mW, 561 nm LS 150 mW, 640 nm LX 140 mW) at 20% intensity using the SoRa spinning disk and the 3.2x magnifier with constant exposure times for each channel (405 nm: 50 ms, 488/ 561/640 nm: 200 ms) on a Hamamatsu ORCA-Fusion BT sCMOS camera. The resulting images had a final resolution of 33.85 nm per pixel xy and 200 nm in z. Each image was acquired as a sequential 4-channel stack, 5 μm in z, covering a variable number of fields of view stitched together to include at least one entire neuron's arborescence, always close to each other to optimize the number of potential synapses. Laser power and all imaging conditions were kept strictly identical across acquisitions. Five replicates per condition were acquired. The resulting VSI files (ranging from 10 to 100 Gb each) were imported into Huygens Pro 22.10 and deconvolved using the same deconvolution pipeline and the same chromatic aberration correction calculated based on acquisition of 100 nm beads Tetraspeck (molecular probes). The resulting 32-bit float images were clipped back down to 16-bit images, saved as

ICS2 before being imported into Imaris 10.0.1. The segmentation strategy was applied across all samples and conditions. Spots detection was based on automatic thresholding in all channels, detecting voxel clusters 200 nm in xy and 400 in z. Only spots located further than 50 μm away from the nucleus (defined as a surface in the 405 nm channel) were included in the analysis to focus on neurites. The values for intensity were extracted as a mean over the tens of thousands of spots detected in each replicate for each condition and marker. The colocalization with SV2A was assessed using the thresholding tools to isolate only spots within 200 nm of SV2A spots, i.e. spots with at least 50% overlap. The statistics were done using two-way ANOVA between C57BL/6J and DDHD2$^{-/-}$ neurons in stimulated and non-stimulated conditions of acquisition across the mean values of each replicate.

## Resuspension of brain lysate samples for western blotting

Brain samples previously processed for FFAST (kept in 0.1 HCl), were precipitated in ice-cold methanol (1:10) at −20 °C for 2 h, then centrifuged for 30 min at 14,000 × g. The supernatant was removed, and the brain lysate pellet was allowed to dry under a gentle stream of nitrogen. Thereafter it was resuspended in 2x LDS buffer.

## Western blotting to assess protein expression in STXBP1 samples

Brain Samples were run by SDS-PAGE (Tris/Glycine, BioRad) on 4–15% Gradient gels (BioRad) at constant 30 amp, SDS-PAGE was transferred onto FL-Immobilon PVDF (Millipore, Cat#04-1624) by wet transfer (Tris, Glycine, 15% MeOH) at 120 V for 90 min at 4 °C. Western blot was blocked in neat Odyssey® Blocking buffer (PBS) (Li-Cor, Cat#927-4000) for 20 min at room temperature and incubated with primary antibodies in blocking buffer (1:1, Odyssey® Buffer and PBST (0.1% Tween20)). Primary antibodies used: Mo-Anti-STXBP1 (Synaptic Systems, Cat#SS116 001, 1/1000), Mo-Anti- β-Actin [AC-15] (Abcam, Cat#ab6276, 1/15,000) and Rb-Anti-DDHD2 (Protein Tech, Cat#25203-1-AP, 1/1000). Membranes were washed 3× for 5–10 min with PBST (1xPBS, 0.1% Tween20) and subsequently incubated with secondary antibodies protected from light. Secondary antibodies IRDye®680RD Goat anti-Mouse IgG (H + L) (Li-Cor, Cat# LCR-312-32211 1/20,000) and IRDye®800CW Goat anti-Rabbit IgG (H + L) (Li-Cor, Cat# LCR-926-32211, 1/20,000). Membranes were washed 3× for 5–10 min with PBST and PBS before imaging on Licor Odyssey® XF imaging system. Images were analysed using Fiji (ImageJ2 v2.9.0/1.53t) and the ratio of Integrated density of STXBP1 or DDHD2 as assessed as a ratio of loading control β-Actin.

Brain tissue, PC12 and DKO PC12 cells were lysed using ice-cold lysis buffer (10 mM Tris/HCl pH 7.5; 150 mM NaCl; 0.5 mM EDTA; 0.5% NP-40 and complete EDTA-free protease inhibitor cocktails; Roche) and the lysate centrifuged at 13,000 RPM for 15 min and the supernatant transferred to a new tube. Protein concentrations were measured by colorimetry using BCA kit (Thermofisher). Briefly, 30 μg of the extracted proteins were electrophoresed on 4–20% precast polyacrylamide gel (BioRad) for 1 h with 100 fixed voltage and transferred to PVDF membranes using wet method for 90 min with 100 fixed voltage. Membranes

were then blocked with intercept blocking buffer (LI-Cor) for 1 h at room temperature. Membranes were then incubated with the DDHD1 rabbit polyclonal primary antibody (Novus Biologicals Cat#NBP2-13903) or DDHD2 rabbit polyclonal primary antibody (Proteintech #25203-AP), Munc18-1/STXBP1 mouse monoclonal primary antibody (BD, Cat# 611463), and Mouse monoclonal anti-β-actin clone AC-74 (Sigma-Aldrich) overnight at 4 °C, followed by 1hr incubation with the anti-mouse IR680 and anti-rabbit IR800 secondary antibodies. Proteins were then visualized by using LI-Cor system.

## Quantitative real-time PCR

PC12, and STXBP1/2 (Munc18-1/2) DKO PC12 cells were collected for RNA isolation. RNA was isolated using RNeasy mini kit (Qiagen). 1 μg of RNA was reverse transcribed to cDNA using High-capacity reverse transcription kit (Applied Biosystems). RT-PCR was carried out using the following primers: rat DDHD1 forward 5′-CATCGATGGAAAAGACGCTGT-3′ and reverse 5′-CCCACTGCTGGAGGCTTTAG-3′, rat DDHD2 forward 5′-CATCGATGGAAAAGACGCTGT-3′ and reverse 5′-CCCACTGCTGGAGGCTTTAG-3′, rat beta actin forward 5′- CCCGCGAGTACAACCTTCTTG-3′ and reverse 5′-GTCATCCATGGCGAACTGGTG-3′. Using PowerUpnTM SYBRTM Green Master Mix (Applied Biosystems) and using the following cycling conditions (denaturation at 95 °C for 15 s, annealing at 60 °C for 30 s, and extension at 72 °C for 30 s) for 40 cycles. The mRNA levels were normalized to β-actin mRNA levels and estimation as delta-delta cycle threshold (DDCT) was calculated as described previously (Rao et al, 2013). All Real-Time Quantitative Reverse Transcription PCR (qRT-PCR) reactions were carried out in triplicate in each biological replicate and each product size was confirmed with the agarose electrophoresis.

## NPY-hPLAP release assay

The neuropeptide-Y-human placental alkaline phosphatase (NPY-hPLAP) release assay was carried out according to a previously described procedure (Martin et al, 2013). PC12 cells and STXBP1 DKO PC12 cells were co-transfected for 72 h with NPY–hPLAP and the corresponding STXBP1 plasmids. Cells were washed and incubated for 15 min at 37 °C with PSS buffer (5.6 mM KCl, 145 mM NaCl, 0.5 mM MgCl$_2$, 2.2 mM CaCl$_2$, 15 mM Hepes-NaOH, 5.6 mM glucose, pH 7.4) or depolarized with high K$^+$ PSS buffer (2.2 mM CaCl$_2$, 70 mM KCl, 0.5 mM MgCl$_2$, 81 mM NaCl, 15 mM Hepes–NaOH, 5.6 mM glucose, pH 7.4) and incubated for 15 min at 37 °C. To measure NPY–hPLAP release, supernatant was collected, and cells were lysed with 0.2% Triton X-100. Using the Phospha-Light$^{TM}$ chemiluminescent reporter gene assay system (Applied Biosystems), the NPY-hPLAP release and total were determined.

## ALPHAScreen protein interaction assay

Amplified Luminescent Proximity Homogeneous Assay Screen (ALPHAScreen) was carried out utilising the cMyc detection kit and Proxiplate-384 Plus plates (PerkinElmer) as previously described (Martin et al, 2013; Sierecki et al, 2013). Each protein pair (one tagged with C-terminal GFP and the other with

C-terminal Cherry-cMyc) was co-expressed in 10 µl of *Leishmania tarentolae* extract (LTE) for 3 h at 27 °C using 20 and 40 nM of DNA template, respectively. LTE lysate co-expressing the proteins of interest was diluted in buffer A (25 mM HEPES, 50 mM NaCl). Each sample was subjected to a four-fold serial dilution. Anti-cMyc coated Acceptor Beads were aliquoted into each well in buffer B (25 mM HEPES, 50 mM NaCl, 0.001% NP40, 0.001% casein) for the experiment. 2 µl of diluted sample and 2 µl of biotin labelled GFP-Nanotrap were then added to buffer A. The plate was incubated at RT for 45 min. 2 µl (0.4 µg) of Streptavidin-coated Donor Beads were added, diluted in buffer A, and incubated in the dark for 45 min at RT. Using an Envision Multilabel Plate Reader (PerkinElmer), the ALPHAScreen signal was captured using the manufacturer's suggested settings (excitation: 680/30 nm for 0.18 s, emission: 570/100 nm after 37 ms). A positive interaction is indicated by the formation of a bell-shaped curve, whereas a flat line indicates a lack of contact between the proteins. Each protein pair was measured a minimum of three times using different plates each time.

## Proteomics analysis: untargeted protein identification using high-resolution tandem mass spectrometry

Sample processing of immuno-precipitates from GFP pull-downs and wild-type process controls in modified SDS-PAGE buffer consisted of protein denaturation and disulfide bond reduction by heating to 60 °C for 30 min in a heating block. Cysteine residues were then alkylated to prevent disulfide bond reformation using 50 mM iodoacetamide for 45 min at RT in the dark. Proteins were then precipitated through the addition of 10 volumes of ice-cold 1:1 methanol/isopropanol followed by overnight incubation at −20 °C. Precipitates were then pelleted by centrifugation at $14{,}000 \times g$ for 15 min in a benchtop microfuge. The supernatant was removed, and the pellet was washed with the addition of 1 mL of ice-cold 1:1 methanol/isopropanol followed by a second round of centrifugation as previously described. The pellet was resuspended in 20 µL of 50 mM ammonium bicarbonate containing 2 µg of proteomics grade trypsin (Sigma-Aldrich). Proteolytic digestion was performed overnight at 37 °C in a shaking incubator set to 200 rotations/min.

Unknown protein identification was performed by injecting 9 µL of the peptide digest onto a 5600 TripleTOF mass spectrometer (AB Sciex) with a microflow LC (Eksigent). Chromatographic conditions consisted of first injecting the sample onto a 0.3 × 10 mm C18 micro trapping column (Phenomenex) under an isocratic flow of 0.1 mM formic acid at 5 µL/min. After 10 min the trapping column was then switched in-line with the separation column (CL120 0.3 × 150 mm C18 with 3 µm particle size, Eksigent). Gradient mobile phases consisted of (A) 0.1 mM formic acid and (B) acetonitrile containing 0.1% formic acid, at a flow rate of 5 µL/min. Chromatographic separations started at 5% (B) at progressing to 32% (B) at 68 min, 40% (B) at 72 min, and 95% (B) at 76 min, plateauing until 79 min, before dropping to 3% (B) at 80 min followed by aqueous column re-equilibration for 7 min. MS acquisitions were performed using positive electrospray ionization using information-dependent data acquisition (IDA) mode. Source conditions consisted of curtain, GS1, and GS2, gases set to 30, 30, and 20 psi, respectively, source temperature at 250 °C, and ion spray and declustering potentials set to 5500 and 100 V, respectively. IDA data acquisition was performed using an initial MS survey scan for 250 ms. From this, the top 30 most abundant ions with an intensity greater than 100 cps, a mass exceeding 350 Da and a charge state between 2–5 were selected for MS/MS collision-induced dissociation (CID) fragmentation. CID collision energy for each of these ions was calculated using the rolling collision energy algorithm, scaling to ion mass. MS-MS spectra were acquired between 100 and 2000 *m/z* for 55 ms.

Data analysis and protein identification was performed using the Protein Pilot software (ABSciex) searched against the combined SwissProt and TrEMBL *Rattus norvegicus* proteomes (31,557 total entries) obtained from the Uniprot online repository (Consortium, 2021). Protein pilot parameters consisted of performing a thorough search, cystine modification with iodoacetamide, and digestion with trypsin, using an ID focus on biological modification with the default paragon method settings for modification frequency. Protein identifications were screened statistically using false discover rate (FDR) using a 1% global threshold for positive identification. Positively identified proteins were then further graded according to the number of identified peptides after 99% confidence screening. Proteins positively identified in the wild-type process controls were considered to be present due to non-specific interactions and removed from the list of proteins identified from the pulldown assays. Identified proteins with 2 or more peptides identified then underwent interaction and GO enrichment analysis using the Search Tool for Retrieval of Interacting Genes/Proteins (STRING) Database online resource (Szklarczyk et al, 2021).

## Proteomics analysis: targeted quantification of DDHD2 and STXBP1 using multiple reaction monitoring

A bespoke targeted multiple reaction monitoring (MRM) LC-MS/MS assay was employed for quantification of *Rattus norvegicus* DDHD2 and STXBP1. Assay design consisted of obtaining protein sequence information for rat isoforms of DDHD2 (D3ZJ91) and STXBP1 (P61765 and P61765-2) from the Uniprot database. Modelling of in silico tryptic digestion of these proteins, calculation of $b$ and $y$ fragment ion masses and theoretical optimal collision energies for each transition using an AB Sciex QTRAP mass spectrometer was determined using the Skyline software platform (MacCross Laboratory) Putative peptides were then screened for specificity to the target protein using blastp function of the NCBI online resource. Finally, MRM transitions for proteotyptic peptides were screened for visibility using the LC-MS/MS methodology described below using tryptic digests of whole-cell protein extracts from rat PC12 cells. The LC-MS/MS specificity of peptides yielding an unambiguous chromatographic peak was confirmed by matching MS2 fragmentation spectra obtained from enhanced product ion scans of PC12 lysates to at least three of the modelled fragment ion masses for each peptide obtained from Skyline.

MRM analysis was performed on an AB Sciex 5500 QTRAP mass spectrometer with a Shimadzu Scientific Nexera series liquid chromatography system. Chromatographic separations were performed using a 3 µL injection volume of the tryptically digested immuno-precipitates used for high-resolution protein identification onto a Aeris peptide XB-C18 2.1 × 100 mm column with a 2.5 µm particle size. Mobile phases consisted of (A) 0.1% v/v formic acid and (B) acetonitrile modified with 0.1% formic acid at a flow rate of 0.5 mL/min. Separations were performed over 20 min using gradient

conditions consisting of an initial isocratic step at 5% B for three min followed by a shallow gradient from 5% B to 50% B at 15 min followed by column washing by increasing to 100% B at 17 min, holding for 3 min and a final column re-equilibration at 5% B for 2 min. Mass Spectrometric acquisitions were performed using positive mode electrospray ionisation. Ion sources gases consisted of curtain gas, GS1 and GS2 applied at 30, 50 and 60 psi, respectively. Source temperature was set to 550 °C. Ion path potentials consisted of ion spray (5500 V), declustering potential (80 V), entrance potential (10 V) and exit potential (9V). Dwell time for each transition was 20 ms for a total cycle time of 800 ms. Precursor $m/z$, product $m/z$ and collision energy for each transition is displayed in Table S1.

### Electron microscopy

E16 hippocampal neurons from C57BL/6J and DDHD2$^{-/-}$ mice were grown on plastic dishes and fixed with 2% glutaraldehyde in 0.1 M sodium cacodylate (e.g., Sigma-Aldrich) buffer, pH 7.4, for 30 min at RT at Days in vitro 21 (DIV21). Samples were post-fixed with 1% reduced osmium tetroxide (OsO$_4$) in the same buffer, for 1 hr, dehydrated through series of ethanol washes, and flat embedded into LX-112 (Ladd Research) as described previously (Joensuu et al, 2020a).

### Cloning of pEGFP-C1-DDHD2$^{WT}$

The pEGFP-C1 vector was linearized at the XhoI and BamHI sites. The digested vectors were treated with calf intestinal phosphatase (CIP) and purified from agarose gel electrophoresis using the Qiagen Gel Extraction Kit. The PCR amplified DDHD2 from DDHD2-FLAG (a kind gift from Yuki Maemoto, School of LifeScience, Tokyo University of Pharmacy and Life Sciences, Japan) was cloned into the linearized vector using the In-Fusion® Snap Assembly Cloning Kit (Takara) and transformed into OmniMAX competent cells. The PCR amplification forward primer (sense) used was 5′-CTGTACAAGTCCGGACTCA-GATCTATGTCATCAGTGCAGTCACAAC-3′, and reverse primer (anti-sense) was 5′-TTATCTAGATCCGGTGGATCCTTACTG-TAAAGGCTGATCAAGG-3′. Purified plasmid DNA was sequenced using ABI BigDye Terminator v3.1 Australian Genome Research Facility (AGRF). Data analysis was performed using the software SnapGene® 5.3.

### Quantitative data processing and visualization

Multiquant® 3.03 (AB SCIEX) was used to quantitatively analyse FFA, DDHD2 and STXBP1, with the MQ4 peak picking/integration algorithm used to manually quantify the LC-MS/MS peak area for each analyte. Quantification of FFAs was performed using a previously established calibration curve for that species, the abundance of each FFA (FFAST-124 or FFAST-127 labelled) was calculated relative to the FFAST-138 labelled internal standard for that species. Six serial dilutions of the FFAST-124 or FFAST-127 labelled FFA calibrators (0.05 ng/ml to 25 ng/ml in acetonitrile) were coupled with 2.5 ng/ml of the corresponding FFAST-138 labelled internal standard to obtain triplicate calibration curves. The ratio between analyte and internal standard peak area was plotted as a standard curve for each FFA. Linear regression without weighting was used to produce calibration slopes (R$^2$ > 0.99). FFA quantification was carried out on experimental data using bespoke Python (python.org) programming. The concentration (ng/ml) of the analyte in the injected sample was calculated by dividing the ratio of the analyte (FFAST-124 or FFAST-127) peak area to the internal standard (FFAST-138) peak area by the calibration slope for that analyte, which was then converted to the molar concentration (pmol/μl) by dividing by the molecular weight of the labelled analyte. The concentration was then normalised to pmol/mg tissue using the weight of the tissue sample and the extraction volume.

Relative quantification techniques were used to assess DDHD2 and STXBP1 LC-MS/MS responses between treatment conditions. In brief this consisted of summing the peak areas for specific peptides across all treatment conditions to determine the relative response of that peptide for each condition. The response for each peptide of a protein were then summed and the aggregate used as a measure of protein abundance. Statistical analysis and visualisation of aggregate protein data was performed using Prism® 9.0 (Graphpad).

To visualize and assess the significance of changes in analyte concentration across brain regions and in response to experimental conditions, lipid data were further analysed using custom-written Python (python.org) scripts variously utilizing Matplotlib (matplotlib.org), Pandas (pandas.pydata.org), Numpy (numpy.org), Seaborn (seaborn.pydata.org) and Scipy (scipy.org) modules. For each analyte the measurements for all the animals were processed to remove outliers by median filtering and generate the mean and standard error of the mean (SEM). Stacked barplots of these data were generated using Pandas and Matplotlib. Heatmaps were generated using Pandas and Seaborn. For heatmaps the significance of the fold-change for each pixel was determined by Student's two-tailed $t$-test using scipy.stats.$t$-test_ind. Scatter plots were generated using Pandas and Matplotlib.

All animal behavioural data was analysed and visualized using Prism® 9.0 (Graphpad).

## Data availability

The data produced in this study is publicly available in the following databases: Imaging dataset: BioImage Archive S-BIAD910. Behavioural and quantified lipid abundance data: University of Queensland eSpace Data Collection (https://doi.org/10.48610/4a44503). The mass spectrometry proteomics data have been deposited to the ProteomeXchange Consortium via the PRIDE (Perez-Riverol et al, 2022) partner repository with the dataset identifier PXD047812. Python scripts for quantitative and multivariate data analysis and visualization are available upon request. Requests for software should be addressed to f.meunier@uq.edu.au (FAM).

## Peer review information

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

## Acknowledgements

This work was also supported by an NHMRC Ideas Grant (2010901) awarded to FAM and TPW, Clem Jones Centre for Ageing Dementia Research (CJCADR) Flagship Grant awarded to FAM and The Academy of Finland Project Funding to MJ (298124). FAM is a National Health and Medical Research Council (NHMRC) Senior Research Fellow (GNT1155794). MJ was supported by an Australian Research Council's (ARC) Discovery Early Career Research Award (DECRA) to (DE190100565). The Australian Government Research Training Program (RTP) Scholarship awarded to IOA by The University of Queensland, and a top-up scholarship courtesy of the Queensland Brain Institute (QBI) also supported this research. MX is a Caroline DeLuca Scholar. The authors thank the University of Queensland Centre for Clinical Research Mass Spectrometry Facility (UQCCR-MSF) for the provision of analytical services and technical advice. The University of Queensland Biological Resources Facility is also appreciated for the provision of support services and technical advice. We thank Alex McCann (QBI) for the critical reading of this manuscript and Vanesa Lanoue for her technical assistance with cell culture. Figures 1A and 4E were created with BioRender.com.

## Author contributions

**Isaac O Akefe**: Conceptualization; Formal analysis; Visualization; Investigation; Writing—review and editing. **Saber H Saber**: Formal analysis; Investigation; Writing—review and editing. **Benjamin Matthews**: Formal analysis; Investigation; Visualization; Writing—review and editing. **Bharat G Venkatesh**: Formal analysis; Investigation; Visualization; Writing—review and editing. **Rachel S Gormal**: Formal analysis; Investigation. **Daniel G Blackmore**: Conceptualization; Resources; Formal analysis; Investigation. **Suzy Alexander**: Formal analysis; Investigation. **Emma Sierecki**: Formal analysis; Investigation; Visualization. **Yann Gambin**: Formal analysis; Investigation; Visualization. **Jesus Bertran-Gonzalez**: Formal analysis; Investigation; Visualization. **Nicolas Vitale**: Resources; Formal analysis; Investigation. **Yann Humeau**: Resources; Formal analysis; Investigation. **Arnaud Gaudin**: Resources; Formal analysis; investigation. **Sevannah A Ellis**: Formal analysis; Investigation. **Alysee A Michaels**: Formal analysis; Investigation. **Mingshan Xue**: Resources; Formal analysis; Investigation. **Benjamin Cravatt**: Resources. **Merja Joensuu**: Conceptualization; Funding acquisition; Formal analysis; Supervision; Validation; Investigation; Visualization; Methodology; Writing—review and editing. **Tristan P Wallis**: Conceptualization; Supervision; Funding acquisition; Software; Formal analysis; Visualization; Investigation; Writing—review and editing. **Frédéric A Meunier**: Conceptualization; Resources; Supervision; Funding acquisition; Project administration; Writing—review and editing.

## Disclosure and competing interests statement

The authors declare no competing interests.

# Expanded View Figure

**Figure EV1.  Longitudinal assessment of motor function, spatial memory, and explorative behaviours in *DDHD2*$^{+/+}$ vs *DDHD2*$^{-/-}$ mice.**

(**A**) Line graphs showing monthly recordings of a motor coordination, assessed using a rotarod device by determining the latency period from when the mouse is placed on the accelerating rotating rod device to the initial fall (s), and (**B**) motor strength (N), assessed using a grip strength device. Longitudinal monitoring of mouse activity with line graphs showing (**C**) vertical counts, (**D**) jump counts, (**E**) ambulatory distance (m), (**F**), vertical time (s), and (**G**) jump time (s). (**H**) Longitudinal assessment of spatial memory performance in mice, using the Novel Object Location (NOL) paradigm, represented as a line graph. Data information: In (**A–H**), the significance of the difference between each group as determined by one-way ANOVA ($n = 20$) is indicated by asterisks $*p < 0.05$, $**p < 0.01$, $***p < 0.001$, ns = not significant. Error bars represent the cumulative standard error of the mean (SEM) for all groups and parameters.

▶

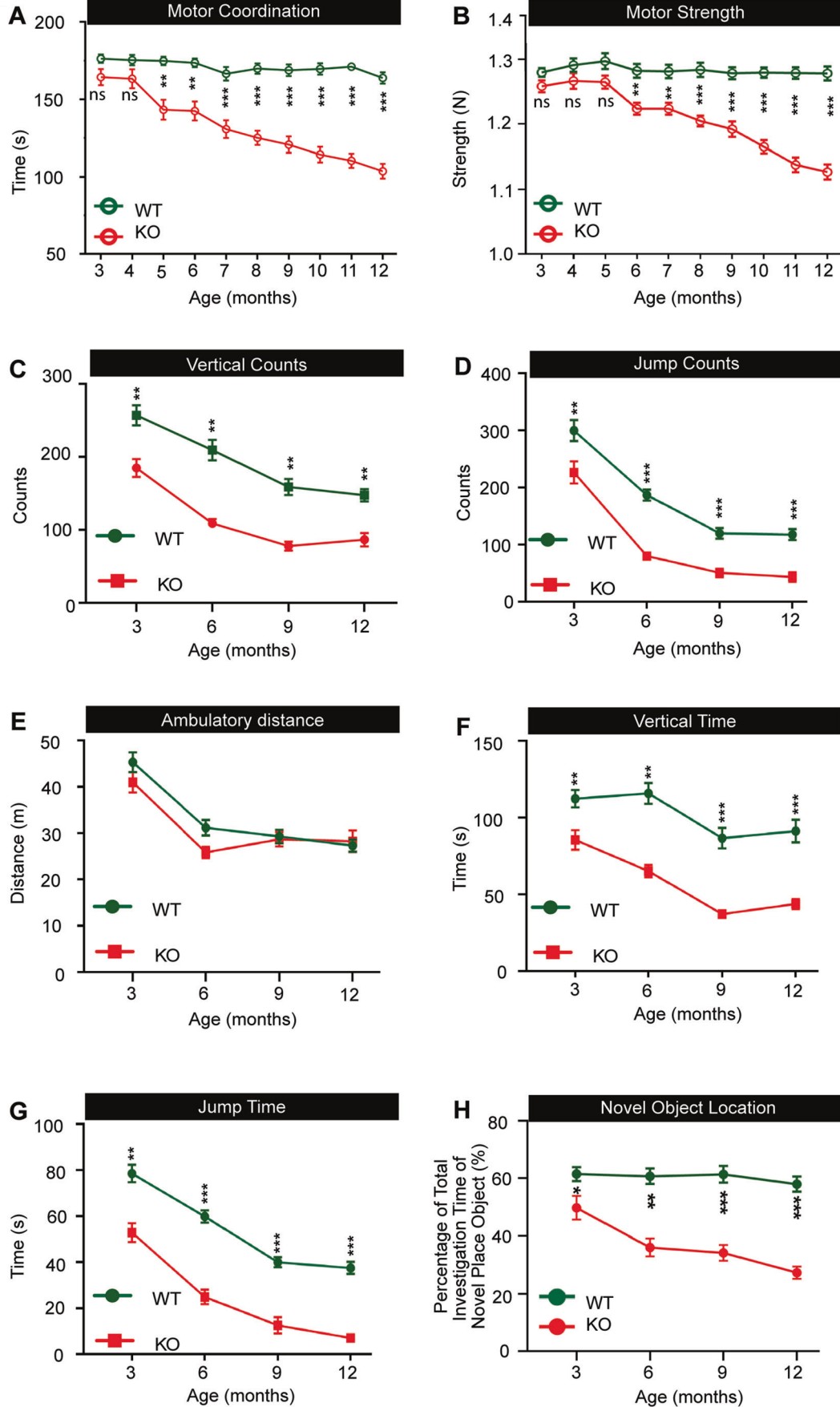

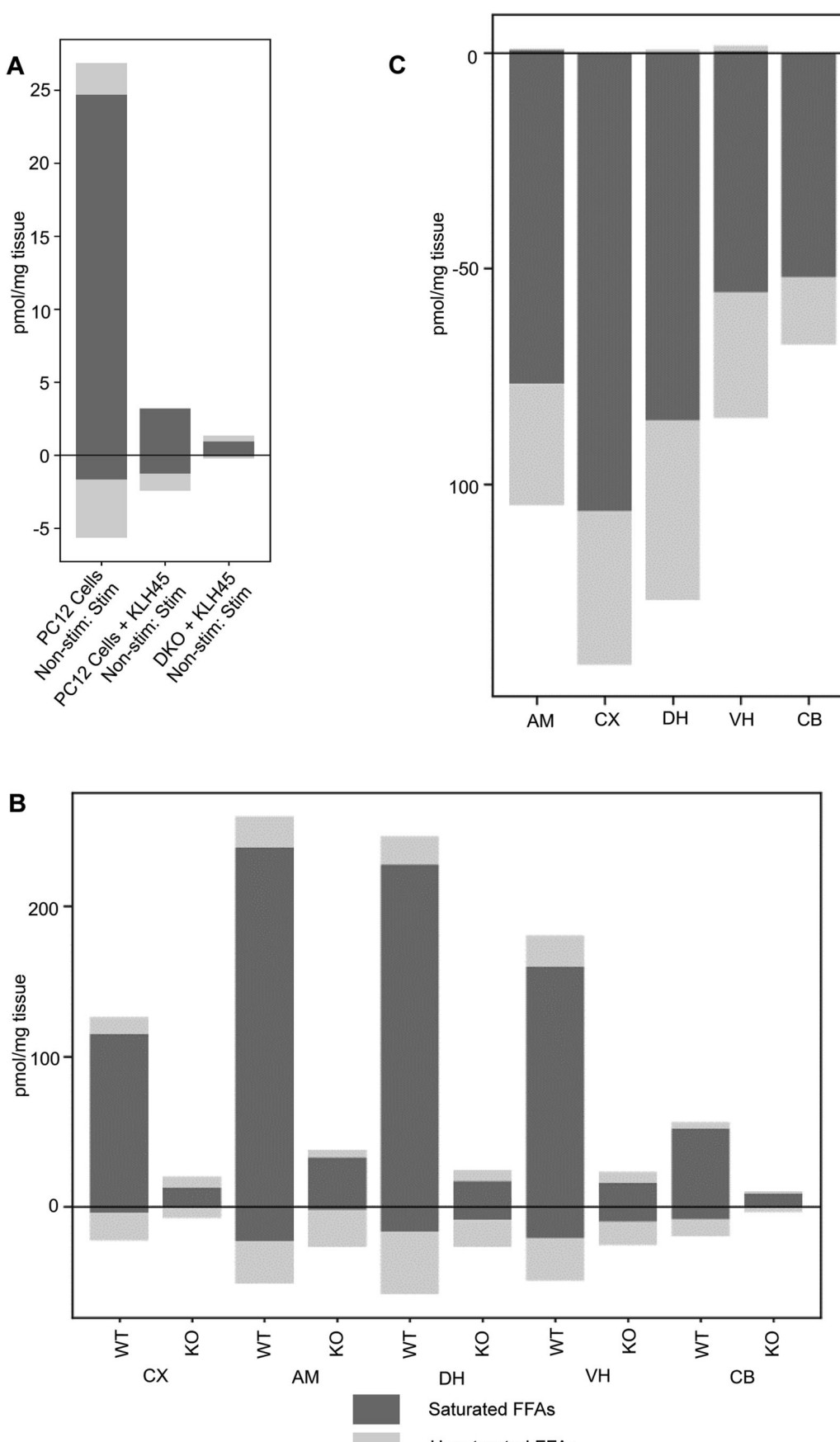

◀ **Figure EV2. Change in saturated (dark grey) vs unsaturated (light grey) FFAs in response to different conditions.**

Bar plots show the change of saturated vs unsaturated FFAs in in response to (**A**) secretagogue stimulation in PC12 cells; (**B**) instrumental conditioning across the brain of WT and DDHD2 KO mice. (**C**) STXBP1 heterozygote across the brain versus WT.

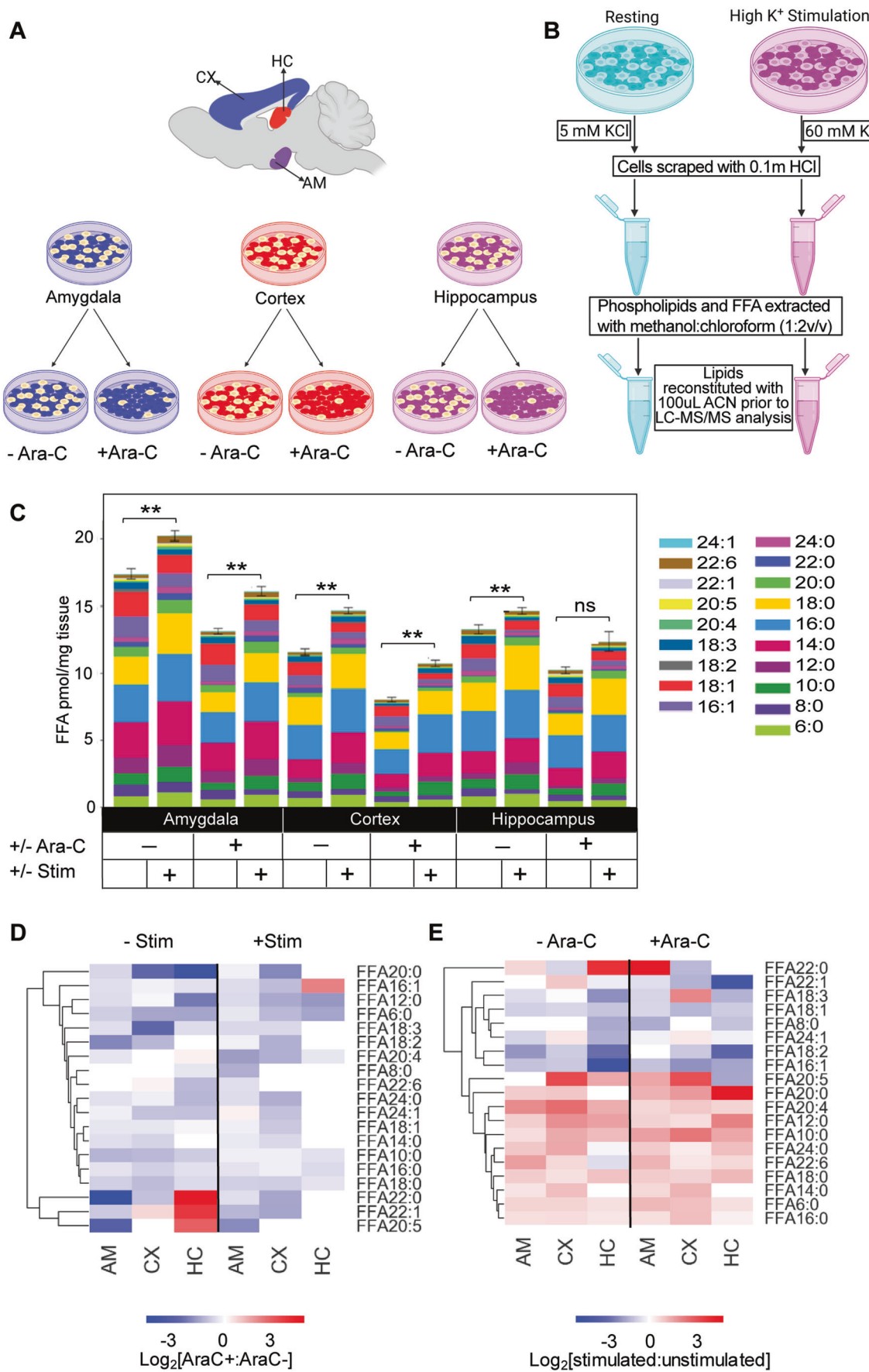

◀  **Figure EV3.  Profile of the effect of ara-C on FFAs response to stimulated neuroexocytosis.**

(A) Inhibition of glia proliferation in neuronal cultures using Ara-C. Cells from the cortex (CX), amygdala (AM), and hippocampus (HC) were isolated from embryonic-day-18 (E18) Sprague-Dawley rat embryos and seeded in separate wells from the indicated brain regions. (B) Potassium stimulation of neuroexocytosis and extraction of lipids. Cultured neurons were treated for 15 min using high potassium buffer; 60 mM $K^+$ (depolarised) or 2 mM $K^+$ (resting control). FFA and phospholipids were extracted in using methanol:chloroform using the liquid–liquid extraction protocol of Bligh and Dyer (Bligh and Dyer, 1959). (C) Bar graph showing quantification of FFA in Amygdala (AM), Cortex (CX), and Hippocampal (HC) neurons with and without ara-C treatment. (D) Hierarchical clustering heatmap showing FFA in response to ara-C treatment in stimulated versus non-stimulated cultures. (E) Hierarchical clustering heatmap showing FFA responses to stimulated neuroexocytosis with and without ara-C treatment. Data information: In (C), the significance of the difference between each group ($n = 3$ biological replicates) as determined by unpaired $t$-test with Holm–Sidak post hoc correction is indicated by asterisks **$p < 0.01$, ns = not significant. Error bars represent the cumulative standard error of the mean (SEM) for all groups and parameters.

