## [Peer Review File · The EMBO Journal]

The DDHD2-STXBP1 interaction mediates long-term memory via generation of saturated free fatty acids

Isaac Akefe, Saber Saber, Benjamin Matthews, Bharat Venkatesh, Rachel Gormal, Daniel Blackmore, Suzy Alexander, Emma Sierrecki, Yann Gambin, Jesus Bertran-Gonzalez, Nicolas Vitale, Yann Humeau, Arnaud Gaudin, Sevannah Ellis, Alysee Michaels, Mingshan Xue, Benjamin Cravatt, Merja Joensuu, Tristan Wallis, and Frederic Meunier

DOI: [10.15252/emboj.2023114489](https://doi.org/10.15252/emboj.2023114489)

Corresponding author(s): *Frederic Meunier (f.meunier@uq.edu.au)*, *Merja Joensuu (m.joensuu@uq.edu.au)*, *Tristan Wallis (t.wallis@uq.edu.au)*

Review Timeline:

Submission Date:	11th May 23
Editorial Decision:	9th Jun 23
Revision Received:	26th Oct 23
Editorial Decision:	28th Nov 23
Revision Received:	6th Dec 23
Accepted:	14th Dec 23

Editors: Karin Dumstrei / Ioannis Papaioannou

Transaction Report:

Dear Fred,

Thank you for submitting your manuscript to The EMBO Journal. Your study has now been seen by three good experts in the field.

As you can see from the comments below, the referees find the analysis interesting. However, they also state that further data is needed in order to consider publication here. In particular they find that we need further insight into the link between DDHD2-mediated lipid metabolism and synaptic function/plasticity, that key data should be repeated with primary neurons from KO mice and further support for that the effects of PLA1/DDHD2 are mediated via myristic acid.

Should you be able to address the concerns raised then I would like to invite you to submit a revised version.

let me know if you need further input from me.

I thank you for the opportunity to consider your work for publication. I look forward to your revision.

with best wishes

Karin

Karin Dumstrei, PhD
Senior Editor
The EMBO Journal

We realize that it is difficult to revise to a specific deadline. In the interest of protecting the conceptual advance provided by the work, we recommend a revision within 3 months (7th Sep 2023). Please discuss the revision progress ahead of this time with the editor if you require more time to complete the revisions.

As a matter of policy, competing manuscripts published during the revision period will not negatively impact on our assessment of the conceptual advance presented by your study. However, we request that you contact the editor as soon as possible upon publication of any related work, to discuss how to proceed.

Use the link below to submit your revision:

Referee #1:

In this manuscript, the authors investigated the link between FFAs and neuronal activity and memory acquisition. They found increases in saturated FFAs across different brain regions of mice after reward-based learning. They then showed that knockout of the DDHD2 isoform of PLA1 in mice inhibited both changes in FFAs and learning and memory. They then found that STXBP1 interacts with DDHD2, and in STXBP1/2 DKO PC12 cells, activity-stimulated increases in saturated FFAs were also abolished. The authors went on to demonstrate that STXBP1 controls the saturated FFA landscape in the brain using STXBP1^{+/-} mice. The findings showing the involvement of DDHD2-generated saturated FFAs in synaptic plasticity are interesting, and the role of STXBP1 in DDHD2 localization to the plasma membrane is a novel mechanism for activity-regulated lipid metabolism. However, there is a missing link between the animal behaviors, lipidomic changes and synaptic function/plasticity. The authors should use cultured primary neurons from the KO mice to obtain experimental data supporting the claim that STXBP1 and DDHD2 activity are involved in lipid metabolism underlying synaptic plasticity.

Major points:

1. In this study, the authors used PC12, a neurosecretory cell line, to study the mechanistic role of STXBP1 in DDHD2 localization to the plasma membrane and its role in activity-dependent FFA metabolism and synaptic plasticity. As PC12 cells are not a good cellular model for mechanistic studies of synaptic function and plasticity, to reach the conclusions that STXBP1 recruits DDHD2 to the plasma membrane for generation of saturated FFAs in response to neuronal activity, the authors should repeat the experiments in Figure 3 and 4 with neurons cultured from either the cortex or hippocampus of WT and STXBP1 KO mice, which are the brain regions with the highest FFA response to instrumental conditioning.
2. The STBP1-DDHD2 interaction should be verified by colP of endogenous proteins from mouse brain.
3. Further, to provide mechanistic insights into the causal link between DDHD2-mediated lipid metabolism and synaptic function/plasticity, the authors should do electrophysiological analysis of acute brain slices from WT and DDHD2^{-/-} mice, e.g. basal synaptic transmission and synaptic potentiation.
4. Figure 3I-K: Contrary to the authors' conclusion that "in the absence of STXBP1/2, both SSHD2 and STX1A were mislocalized away from the plasma membrane", in DKO cells, although signal intensity of both DDHD2 and STX1A greatly decreased (not explained), most of the punctate signals still localize at the plasma membrane. Moreover, given the different functions of PLA and SNARE proteins at the plasma membrane, colocalization between these two proteins is not a good assessment for STXBP1 function in PM recruitment of DDHD2. The authors should reinterpret the data with greater caution.
5. Figure 3M-P: compared with DDHD2 and STX1A, most of the overexpressed GF-STXBP1 was diffusely distributed or aggregated in the cytoplasm, raising the question whether it could recruit both DDHD2 and STX1A to the plasma membrane.
6. Figure 4C: For pharmacological inhibition of DDHD2, what is the incubation time and concentration of KLH45? And according to the results from DDHD2 KO mouse brain, inhibiting DDHD2 activity should also cause decreases in FFA levels in non-stimulated PC12 cells.
7. Concentrations of not only 14:0 but also other saturated FFAs were changed with memory training and in KO mice. Figure 1 shows even more changes in 18:0 and 16:0 than 14:0. Moreover, the causal link between generation of myristic acid or saturated FFAs and long-term memory has not yet been established in this manuscript. So the title "DDHD2 interacts with STXBP1 to mediate long-term memory via the generation of myristic acid" should be modified to match with the data.

Minor points:

1. When talking about heterogeneity in FFA concentrations across brain regions, no data was shown for the conclusion "with the highest concentrations relative to tissue weight in the prefrontal cortex, and lowest concentrations in the cerebellum".
2. The description about lipidomic data is inconsistent with data in Extended Figure 1E. Both 16:1 and 18:1 decreased and 24:0 increased following instrumental conditioning.
3. Figure 1D: Besides 18:3, how about the difference in 16:1 between WT and DDHD2 KO mice?
4. For comparison and in parallel with the FFA concentration changes in Figure 1C-F, move Extended Figure 1A-C to Figure 1 and combine with original Figure 1B and C to show the instrumental conditioning results for both 3mo and 12mo mice. Extended Figure 1 should be used exclusively to describe the workflow of lipidomic analysis and heterogeneity in FFA concentration

changes across brain regions in 12mo WT mice following instrumental conditioning.

5. Figure 4D: what do the center and right pie charts mean? FFAs decrease, not increase, in PC12 cells stimulated compared with DKO or KLH45 treated cells?

6. Figure 5C: what is the Y axis? Abundance change of FFAs?

7. On p12: under the subtitle "STXBP1 binding to DDHD2 controls FFA levels", the authors referred to Figure 4B and 5A instead of Figure 4A.

8. Figure legends:

For Extended Fig. 1B and C, I would assume the data shown are lever pressing after 14 days of training.

Extended Fig. 7: which structures do the arrowheads indicate?

Fig. 1 : f should be F.

Referee #2:

In this paper Akefe et al., extend their previous work generalizing the concept of saturated free fatty acid production in the brain during cognitive activities and identify the phospholipase A1 DDHD2 as the main source for their synthesis. This is an important novel concept. This study also uncovers for the first time an interplay between DDHD2 and STXBP1, one of the major regulators of SNARE activity during synaptic vesicle exocytosis. But the mechanism by which saturated free fatty acids modulate memory consolidation remains to be elucidated.

The following points need to be considered by the authors:

- Validation of DDHD2 knock-out should be illustrated at the expression level using brain extracts. Furthermore, evaluation of the impact on DDHD1 expression would also be helpful to validate the specificity of the findings reported here.
- Potential long-term impact on the lipidome profile of DDHD2 knockout mice and its possible effect on the behavioral tests performed here may be validated by the acute use of the DDHD2 inhibitor KLH45. It is indeed of note that Inloes et al., 2014 reported accumulation of TAG and lipid droplets in DDHD2^{-/-} cells illustrating major lipidic alterations.
- p9 the statement "The fact that saturated FFA increases are activity-dependent makes it likely that the activity of this enzyme is localized at the synapse." is neither supported by solid evidence at this stage of the manuscript nor by the literature (Inloes et al., 2014) and should thus be more speculative.
- In vitro competition experiments could be performed to better describe the interaction of STXBP1 with STX1 and DDHD2 and establish whether there is some competition or whether simultaneous binding occurs. The ability of STXBP1 to interact and possibly transport to the cell periphery additional phospholipases can be discussed.
- MALDI-imaging mass spectrometry if available would significantly improve the significance of the lipidomics data.

Additional points:

- There is probably a mistake in the legend of Figure 3D. Green line should be DDHD2 instead of STXBP1.
- Correct few typos.

Referee #3:

The manuscript from Akefe et al, entitled "DDHD2 interacts with STXBP1 to mediate long term memory via the generation of myristic acid", addresses the important role of lipids and lipid modulating enzymes in learning and memory. The authors tackle the role of PLA1/DDHD2 using a combination of animal and cellular models, with behavior, imaging and biochemistry approaches. However, based on the presented results, there are a number of key issues that should be addressed.

Major general points

- The authors propose that PLA1/DDHD2 could be playing a role in memory mechanisms, through myristic acid (based on the title). However, in the results section, this specificity for myristic acid is not as clear as the title implies. Also, it is not clear if the results could be driven by changes in other pools of FFAs or, importantly, in the pools of glycerophospholipids, which are substrates of PLA1, such as phosphatidic acid (PA). In fact, PA, and PA synthesizing enzymes, such as PLD1 have been extensively implicated in neuroendocrine chromaffin cells secretion, similarly to PC12 cells (Tanguy et al, 2020), PA has also been shown to bind to STX1A (Lam et al, 2008) and PA levels have shown to be linked with plasticity and hippocampal behavior alterations (Santa-Marinha et al, 2020). Therefore, and in line with what has been done in Wallis et al, 2020, it would have been relevant to test for levels of PA, or other glycerophospholipids modulated by PLA1, since PA, or other glycerophospholipid levels, could be involved in learning and memory, as a way to complement FFA assessments. An alternative would be to tackle the potential role of PA in a cell system, with a PA binding probe.

- Functional experiments linking DDHD2 to synapse function would be favourable for this work. For instance, changes in synaptic strength and synapse plasticity are at the basis of memory processes. Therefore, showing that DDHD2 KO has an

impact on these functional properties would be of interest. Alternatively, the authors could induce chemical LTP or LTD and inquire if synapse numbers are altered using primary cultures from DDHD2 KO.

- In different experimental settings the authors explore the role of STXBP1, using a double knock-out model with STXBP2. Can the authors clarify what could be the differential role of STXBP1 vs STXBP2?

- As it is written in the Results section and the organization of the Figures, it is very hard to follow the connection between the Figures and Results. Overall the flow of the results is not very linear, in particular - Fig1, Extended Fig1 & Extended Fig2. For instance, if the message that is being conveyed is that instrumental conditioning drives region-specific changes in the brain lipidome, it makes it hard to follow when DDHD2 KO is mixed up in this message. The authors should re-write and re-organize the figure display in order to match the text. For instance, Extended Fig. 1E is apparently not relative to DDHD2 KO, but panels 1A-C are.

Major specific points

- The authors should be clearer in the first sentence of the abstract - are they referring to fatty acyl composition of glycerophospholipids in membranes, or are they referring to FFA?

- In page 5, it should be avoided to mention "lipidome" if only FFAs were assessed. Unless a more extensive lipidomic analysis is indeed performed.

Fig1, Extended Fig1 & Extended Fig2

- Based on Fig1 the authors claim that learning increases the production of FFA. However, in Fig1, they compare WT vs KO only, both from the learning group. Therefore, the only explanation possible for these results is that ablation of DDHD2 has an impact on FFA. The authors do not clarify if these results are normalized to the control group (no learning).

- Based on Extended Fig 2 the authors claim that there are memory defects, however, it is not convincing that the performances of these mice in these tests is not due to motor impairment.

- Even though the authors have used a similar graphic representation of the data as in Wallis, 2021, for Fig 1C-F, the lines are superimposed, which makes them hard to interpret the underlying representation of each group. It would be more effective to show the data in columns, or alternatively heatmap, comparing each group, and mentioning which comparisons that are statistically significant and specify in the legend (and methods section) the statistical analysis.

- Color assignment for each FFA appears to be somewhat random. A visualization suggestion could be to use Saturated FFAs vs Unsaturated FFAs with a more differentiating color code.

- In Extended Data Fig. 1E - which ones of the alterations are indeed significant? Authors should highlight only significant results and clarify statistical analysis employed.

Figure3

- Since the authors bring up a possibility that a complex between STX1A, STXBP1 and DDHD2 might occur, it would be interesting to perform co-localization studies between these proteins (DDHD2-STXBP1 and DDHD2-STX1A) before and after stimulation. Based on the methods it appears the authors used Pearson co-efficient, but this should be clearer if it was indeed the case. As a complement or alternative for co-localization assessment, the authors can also use proximity-ligation assays.

Figure4

- PC12 vs DKO non-stim is a condition both present in Fig. 4A and 4B - were these independent experiments? Statistical analysis should be done for Fig. 4A.

- In Fig4 - Re-expression of WT STXBP1 in these DKO PC12 cells completely rescued the FFA basal levels - it is interesting, but once again, what is then the role of STXBP2? Does it also rescue? Is it any different that STXBP1?

- The way Fig 4D is presented is not clear - instead bar graphs should be used to understand exactly which comparisons are performed and the proper statistical analyses mentioned.

Figure5

- What happens to the levels of DDHD2 and STXN1A in these brain regions?

- Is this a matter of regulation of DDHD2 activity or simply without the complex it is more degraded?

- Once again, what happens to PA or other glycerophospholipids?

- Figure 5C is hard to read with all lipids superimposed - it is interesting that one FFA is significantly increased - 24:0 this should be discussed. Also, various unsaturated and saturated FAs are decreased - should this argue against a specific effect on saturated FFAs?

Minor Comments:

- 24:0 is always in a bottom position and not grouped with the other saturated FFAs in the legend.

- Extended Figs 1B and 1C should be mentioned in the text.

- Extended Fig 3 - in the y axis - do the authors mean "tissue" instead of "cells"?

- Fig 2B-C-D some graphs have data points but not others - this should be standardized.

- Extended Fig 5 - it should be mentioned the names of the proteins from the top 10
- Is the legend in panel D, Fig3, incorrect?
- No legend is shown with the different colors in Fig4
- In page 12, do the authors mean Fig. 4A instead of 5A?
- In Fig 5B and C should it be mentioning Het instead of KO?
- Fig5B appears to be unnecessary - particularly as a main figure.

Point-by-point rebuttal

Referee #1:

In this manuscript, the authors investigated the link between FFAs and neuronal activity and memory acquisition. They found increases in saturated FFAs across different brain regions of mice after reward-based learning. They then showed that knockout of the DDHD2 isoform of PLA1 in mice inhibited both changes in FFAs and learning and memory. They then found that STXBP1 interacts with DDHD2, and in STXBP1/2 DKO PC12 cells, activity-stimulated increases in saturated FFAs were also abolished. The authors went on to demonstrate that STXBP1 controls the saturated FFA landscape in the brain using STXBP1^{+/-} mice. The findings showing the involvement of DDHD2-generated saturated FFAs in synaptic plasticity are interesting, and the role of STXBP1 in DDHD2 localization to the plasma membrane is a novel mechanism for activity-regulated lipid metabolism. However, there is a missing link between the animal behaviors, lipidomic changes and synaptic function/plasticity. The authors should use cultured primary neurons from the KO mice to obtain experimental data supporting the claim that STXBP1 and DDHD2 activity are involved in lipid metabolism underlying synaptic plasticity.

Major points:

1. In this study, the authors used PC12, a neurosecretory cell line, to study the mechanistic role of STXBP1 in DDHD2 localization to the plasma membrane and its role in activity-dependent FFA metabolism and synaptic plasticity. As PC12 cells are not a good cellular model for mechanistic studies of synaptic function and plasticity, to reach the conclusions that STXBP1 recruits DDHD2 to the plasma membrane for generation of saturated FFAs in response to neuronal activity, the authors should repeat the experiments in Figure 3 and 4 with neurons cultured from either the cortex or hippocampus of WT and STXBP1 KO mice, which are the brain regions with the highest FFA response to instrumental conditioning.

Response: We thank the reviewer for this fair criticism. We considered his request but the mice STXBP1 het mice are located in the lab Dr Mingshan Xue (Baylor College, Texas) and the logistics of bringing them to Australia to perform these experiments is difficult and would have taken a long time. However, we have carried out additional experiments that address this point. Our data demonstrate that DDHD2 binds STXBP1 (Fig 4). We have also used neurosecretory cells to demonstrate that in STXBP1 and STXPB1 and 2, the transport of DDHD2 was completely blocked and that this phenotype was fully rescued upon re-expression of STXBP1 (new Fig 5). We have now confirmed that in DDHD2 KO neurons, the anterograde trafficking was significantly impacted, with strong disruption of the ERGIC system and to a lesser extent the Golgi (new Fig 6). Further, using electron microscopy we found that the number of synaptic vesicles was reduced, which was confirmed by super-resolution microscopy using SV2 as a synaptic vesicle marker (new Fig 7). The cortex of DDHD2 KO mice was analysed by western blotting to assess the levels of SNAP25, Synaptotagmin 1, and DDHD1 and these were normal (extended data S7). Conversely, DDHD2 was reduced although not significantly in STXBP heterozygote hippocampal tissue (new Supporting data Figure S10) These data point to a clear-cut transport deficit that is nicely correlated with the impacted saturated FFA levels as follows: In neurosecretory cells the levels of sFFA were strongly inhibited by both STXBP1 knockout and STXPB1 and 2 knockout and rescued upon re-expression of STXBP1 (Fig 8). Further, we found that this rescue was not related to the priming function of STXPB1, suggesting that the transport deficit is directly responsible of the inhibition sFFAs (Fig 8). Finally, we demonstrate that in all regions of the STXBP heterozygote tested the levels of sFFAs are much reduced (Fig 9). Our additional experiments *in vitro* and *in vivo* demonstrate that STXBP1 controls DDHD2 transport and activity.

2. The STBP1-DDHD2 interaction should be verified by colP of endogenous proteins from mouse brain.

The GFP trap system we used from DDHD2-GFP expressing PC12 cells, pulls down STXBP-1 (Fig 4). The GFP trap system we used from STXBP-GFP expressing DKD-PC12 cells pulls down DDHD2 (Fig 4). These were detected by western blotting and mass spectrometry with large peptide coverage (Fig 4). Further, this interaction was found *in vitro* using the expressed proteins (Fig. 4). We did try the pull-down experiment but unfortunately, none of the Abs we used were sufficiently sensitive and despite many trials we could not generate this data. Although it is disappointing, the GFP trap combined with the ALPHAScreen and all the functional data provide, in our view, strong evidence that this interaction is direct and functionally relevant.

3. Further, to provide mechanistic insights into the causal link between DDHD2-mediated lipid metabolism and synaptic function/plasticity, the authors should do electrophysiological analysis of acute brain slices from WT and DDHD2^{-/-} mice, e.g. basal synaptic transmission and synaptic potentiation.

This is a great suggestion and an important experiment to do. However, given the time and the fact that we are not equipped to carry out electrophysiology we had to drop this idea, which we hope can be done in the near future.

4. Figure 3I-K: Contrary to the authors' conclusion that "in the absence of STXBP1/2, both SSHD2 and STX1A were mislocalized away from the plasma membrane", in DKO cells, although signal intensity of both DDHD2 and STX1A greatly decreased (not explained), most of the punctate signals still localize at the plasma membrane. Moreover, given the different functions of PLA and SNARE proteins at the plasma membrane, colocalization between these two proteins is not a good assessment for STXBP1 function in PM recruitment of DDHD2. The authors should reinterpret the data with greater caution.

We are sorry that we had not made this figure and associated result clearer. It has been well documented by several laboratories including Rory Duncan, Shuzo Sugita, and mine that STXBP1/Munc18-1 controls the transport of Syntaxin1A to the plasma membrane. Our point is that DDHD2 follows the same anterograde pathway. In PC12 cells, both DDHD2 and Syntaxin1A appear on the plasma membrane. In the absence of Munc18, in DKO-PC12 cells, both proteins are no longer localised to the plasma membrane. Their transport to the plasma membrane can however be rescued upon re-expression of STXBP1/Munc18-1. We have altered the analysis and representation of Fig 4 to make this point clearer in the result section and the new Fig 4. The fact that the levels of both Syntaxin1A and DDHD2 are much reduced has been attributed to the trafficking block interference with protein expression (Kasula *et al*, 2016; Martin *et al*, 2013), (Han *et al*, 2010; Han *et al*, 2011; Malintan *et al*, 2009) Rickman *et al.*, 2007). The co-localisation between Syntaxin1A and DDHD2 was expected but is of less importance in the context of the transport phenotype. We have therefore replaced this panel by the analysis of the DDHD2 transport defect and its rescue upon re-expression of STXBP-1 in DKD-PC12 cells. We hope our message is now clearer.

5. Figure 3M-P: compared with DDHD2 and STX1A, most of the overexpressed GF-STXBP1 was diffusely distributed or aggregated in the cytoplasm, raising the question whether it could recruit both DDHD2 and STX1A to the plasma membrane.

See response above.

6. Figure 4C: For pharmacological inhibition of DDHD2, what is the incubation time and concentration of KLH45? And according to the results from DDHD2 KO mouse brain, inhibiting DDHD2 activity should also cause decreases in FFA levels in non-stimulated PC12 cells.

KLH45 was used at 25 nM for 4 h. This information is now included in the Fig legend. The lower basal FFA levels in DDHD2 KO mice are likely due to a lifetime of compounding effects of altered early secretory trafficking of proteins that are directly or indirectly involved in FFA metabolism. The acute treatment of PC12 cells with KLH45 clearly alters the activity-dependent catalytic activity of DDHD2 in the presynapse, but most likely does not allow sufficient time for secretory dysregulation to manifest itself in respect to basal FFAs as seen in the DDHD2 KO neurons.

7. Concentrations of not only 14:0 but also other saturated FFAs were changed with memory training and in KO mice. Figure 1 shows even more changes in 18:0 and 16:0 than 14:0. Moreover, the causal link between generation of myristic acid or saturated FFAs and long-term memory has not yet been established in this manuscript. So the title "DDHD2 interacts with STXBP1 to mediate long-term memory via the generation of myristic acid" should be modified to match with the data.

The reviewer is correct, and we have amended the title to reflect this valid point: "DDHD2 interacts with STXBP1 to mediate long-term memory via the generation of saturated free fatty acids".

Minor points:

1. When talking about heterogeneity in FFA concentrations across brain regions, no data was shown for the conclusion "with the highest concentrations relative to tissue weight in the prefrontal cortex, and lowest concentrations in the cerebellum".

This data was provided in Extended Figure S3. We realise that this was important data and have now relocated it to the main figure (Fig 2). We have also included more background data related to Wallis et al. Nat Comm (Wallis et al, 2021).

2. The description about lipidomic data is inconsistent with data in Extended Figure 1E. Both 16:1 and 18:1 decreased and 24:0 increased following instrumental conditioning.

This has been corrected.

3. Figure 1D: Besides 18:3, how about the difference in 16:1 between WT and DDHD2 KO mice?

We have now addressed this point in the result section.

4. For comparison and in parallel with the FFA concentration changes in Figure 1C-F, move Extended Figure 1A-C to Figure 1 and combine with original Figure 1B and C to show the instrumental conditioning results for both 3mo and 12mo mice. Extended Figure 1 should be used exclusively to describe the workflow of lipidomic analysis and heterogeneity in FFA concentration changes across brain regions in 12mo WT mice following instrumental conditioning.

We have made these changes and consolidated these figures as requested. We thank the reviewer for this important input.

5. Figure 4D: what do the center and right pie charts mean? FFAs decrease, not increase, in PC12 cells stimulated compared with DKO or KLH45 treated cells?

This has now been replaced with bar graphs and moved to Extended Fig 8. In essence, we pulled all the saturated FFA in dark grey and all the unsaturated FFA in light grey. This gives a fair account of the role of DDHD2 in generating saturated FFAs and the impact of its inhibition by KLH45.

6. Figure 5C: what is the Y-axis? Abundance change of FFAs?

The Y-axis represents the significance of the change – Log 10 of the p-value generated by an unpaired Student's t-test.

7. On p12: under the subtitle "STXBP1 binding to DDHD2 controls FFA levels", the authors referred to Figure 4B and 5A instead of Figure 4A.

We thank the reviewer for noticing this mistake. This has been corrected.

8. Figure legends:

For Extended Fig. 1B and C, I would assume the data shown are lever pressing after 14 days of training.

The data shows the total mean lever presses/min. This has now been included in new Figure 1.

Extended Fig. 7: which structures do the arrowheads indicate?

This has now been indicated in the legends.

Fig. 1 : f should be F.

This has been corrected.

Referee #2:

In this paper Akefe et al., extend their previous work generalizing the concept of saturated free fatty acid production in the brain during cognitive activities, and identify the phospholipase A1 DDHD2 as the main source for their synthesis. This is an important novel concept. This study also uncovers for the first time an interplay between DDHD2 and STXBP1, one of the major regulators of SNARE activity during synaptic vesicle exocytosis. But the mechanism by which saturated free fatty acids modulate memory consolidation remains to be elucidated.

The following points need to be considered by the authors:

- Validation of DDHD2 knock-out should be illustrated at the expression level using brain extracts. Furthermore, evaluation of the impact on DDHD1 expression would also be helpful to validate the specificity of the findings reported here.

The impact of DDHD2 KO on DDHD2, DDHD1 and the synaptic markers SNAP-25, synaptotagmin1 and STXBP1/Munc18-1, has now been included in the result section: Appendix Figure S7.

- Potential long-term impact on the lipidome profile of DDHD2 knockout mice and its possible effect on the behavioral tests performed here may be validated by the acute use of the DDHD2 inhibitor KLH45. It is indeed of note that Inloes et al., 2014 reported accumulation of TAG and lipid droplets in DDHD2^{-/-} cells illustrating major lipidic alterations.

We agree with the reviewer that using WT mice treated with the KLH45 inhibitor should reduce their ability to form memories but at this early stage we have not performed toxicity nor pharmacokinetic analyses and are therefore unsure whether this inhibitor can cross the blood brain barrier. We are sorry to report that we could not carry out this interesting experiment but have plan to do so in the future.

- p9 the statement "The fact that saturated FFA increases are activity-dependent makes it likely that the activity of this enzyme is localized at the synapse." is neither supported by solid evidence at this stage of the manuscript nor by the literature (Inloes et al., 2014) and should thus be more speculative.

This is a very important point raised by the referee. We have now demonstrated by immunocytochemistry and super-resolution microscopy that DDHD2 is in fact present at the synapse, see new Figure 7A.

- In vitro competition experiments could be performed to better describe the interaction of STXBP1

with STX1 and DDHD2 and establish whether there is some competition or whether simultaneous binding occurs. The ability of STXBP1 to interact and possibly transport to the cell periphery additional phospholipases can be discussed.

Our study revealed that STXBP1 binds DDHD2 and transport it to the plasma membrane where it performs its function and generate saturated FFAs that are important for memory acquisition. The reviewer is correct in that STX1A could compete with DDHD2 for STXBP1/Munc18-1 binding or display cooperative binding. Although we feel that this question is outside the scope of our study, we nonetheless attempted several times to perform colP experiments in the view of testing this hypothesis but unfortunately the Abs used were not optimal for IP. Other strategies will have to be employed to test this valid question.

- MALDI-imaging mass spectrometry if available would significantly improve the significance of the lipidomics data.

This is a great suggestion which is currently being evaluated for a PhD project. We have generated preliminary data from MALDI imaging which clearly shows brain region specificity of various phospholipids (see below: 597Da potentially PE12:0_12:0). However, imaging of FFAs is particularly challenging as they ionise poorly and are difficult to differentiate from other low MW metabolic species. We are therefore considering adapting our FFAST derivatisation for MALDI imaging. This will take considerable amount of time and is outside the scope of this study, but we plan to carry this out in the future.

Additional points:

- There is probably in mistake in the legend of Figure 3D. Green line should be DDHD2 instead of STXBP1.

- Correct few typos.

These have now been corrected.

Referee #3:

The manuscript from Akefe et al, entitled "DDHD2 interacts with STXBP1 to mediate long term memory via the generation of myristic acid", addresses the important role of lipids and lipid modulating enzymes in learning and memory. The authors tackle the role of PLA1/DDHD2 using a combination of animal and cellular models, with behavior, imaging, and biochemistry approaches. However, based on the presented results, there are a number of key issues that should be addressed.

Major general points

- The authors propose that PLA1/DDHD2 could be playing a role in memory mechanisms, through myristic acid (based on the title). However, in the results section, this specificity for myristic acid is not as clear as the title implies.

The reviewer is correct, and the title has now been changed accordingly: "DDHD2 interacts with STXBP1 to mediate long-term memory via the generation of saturated free fatty acids".

Also, it is not clear if the results could be driven by changes in other pools of FFAs or, importantly, in the pools of glycerophospholipids, which are substrates of PLA1, such as phosphatidic acid (PA). In fact, PA, and PA synthesizing enzymes, such as PLD1 have been extensively implicated in neuroendocrine chromaffin cells secretion, similarly to PC12 cells (Tanguy et al, 2020), PA has also been shown to bind to STX1A (Lam et al, 2008) and PA levels have shown to be linked with plasticity and hippocampal behavior alterations (Santa-Marinha et al, 2020). Therefore, and in line with what has been done in Wallis et al, 2020, it would have been relevant to test for levels of PA, or other glycerophospholipids modulated by PLA1, since PA, or other glycerophospholipid levels, could be involved in learning and memory, as a way to complement FFA assessments. An alternative would be to tackle the potential role of PA in a cell system, with a PA binding probe.

This is a particularly relevant question pertaining to the selectivity of the phospholipase A1 involved. We have extensively explored the possible substrate contribution of various PLs. We have carried out multiples experiments to tackle this important question. First, we have analysed the response to instrumental conditioning to 4 of the main phospholipid classes (see Figure 3). We report a very heterogenous response for the phospholipids tested which makes it arduous to establish with confidence a clear substrate-product relationship. In particular, PA species do not stand out as favoured substrate. Second, Gil Di Paolo's group has previously demonstrated that PLD1 KO mice did not impact on the fear memory response suggesting that the PA substrate generated by PLD1 are not essential for memory acquisition (Santa-Marinha *et al*, 2020). We obtained the PLD1 KO (Appendix Figure S4) to perform the same behavioural analysis and examine the lipid response. Our results fully corroborate Di Paolo's group finding by showing no effect on fear memory acquisition even though PA levels were greatly reduced in these animals as anticipated (See new appendix Figure S3 and S4).

- Functional experiments linking DDHD2 to synapse function would be favourable for this work. For instance, changes in synaptic strength and synapse plasticity are at the basis of memory processes. Therefore, showing that DDHD2 KO has an impact on these functional properties would be of interest. Alternatively, the authors could induce chemical LTP or LTD and inquire if synapse numbers are altered using primary cultures from DDHD2 KO.

We agree with the reviewer and these experiments are ongoing in the lab in the view to build up a comprehensive new manuscript on the mechanism by which DDHD2 controls synaptic plasticity. We have however, included new sets of data that strongly suggest a link between DDHD2 and synaptic function.

- 1- We show that DDHD2 is present and enriched at the synapse by IF (new Figure 7).
- 2- We show that DDHD2 KO significantly reduces the number of synaptic vesicles at the presynapse (new Figure 7).
- 3- The activity-dependent increase in saturated FFAs is completely blocked in DDHD2 KO hippocampal neurons (new Figure 8).
- 4- The activity-dependent increase in saturated FFAs is not affected by treatment with Ara-C that stops astrocytic proliferation. Indeed, we profiled the FFA changes in neuronal cell cultures from the amygdala, cortex and hippocampus. In all cultures, inhibition of glial proliferation reduced the overall FFA levels but did not change the activity-dependent increase in the saturated FFA (new Appendix Fig S9C-E). These data demonstrate that glia contribute to some extent to the basal levels of FFAs but that synapses likely drive the activity-dependent response.

- In different experimental settings the authors explore the role of STXBP1, using a double knock-out model with STXBP2. Can the authors clarify what could be the differential role of STXBP1 vs STXBP2?

We have now added the FFA profile for MKO cells in which only STXBP1 is ablated (new Appendix Figure S9). This data suggests that STXBP1 is primarily responsible for the maintenance of basal FFA levels.

- As it is written in the Results section and the organization of the Figures, it is very hard to follow the connection between the Figures and Results. Overall the flow of the results is not very linear, in particular - Fig1, Extended Fig1 & Extended Fig2. For instance, if the message that is being conveyed is that instrumental conditioning drives region-specific changes in the brain lipidome, it makes it hard to follow when DDHD2 KO is mixed up in this message. The authors should re-write and re-organize the figure display in order to match the text. For instance, Extended Fig. 1E is apparently not relative to DDHD2 KO, but panels 1A-C are.

We thank the reviewer for highlighting these inconsistencies in our presentation. The figures have now been re-arranged accordingly, as per the concerns raised by reviewer 1.

Major specific points

- The authors should be clearer in the first sentence of the abstract - are they referring to fatty acyl composition of glycerophospholipids in membranes, or are they referring to FFA?

We are specifically referring to the free fatty acid. We have modified the text of the manuscript accordingly.

- In page 5, it should be avoided to mention “lipidome” if only FFAs were assessed. Unless a more extensive lipidomic analysis is indeed performed.

We have modified the text to clarify our use of the word lipidome. Please note that we have also subsequently added a significant amount of phospholipid data to the revised manuscript. When discussing only the FFA part of the lipid analysis we have referred to the FFA lipidome.

Fig1, Extended Fig1 & Extended Fig2

- Based on Fig1 the authors claim that learning increases the production of FFA. However, in Fig1, they compare WT vs KO only, both from the learning group. Therefore, the only explanation possible for these results is that ablation of DDHD2 has an impact on FFA. The authors do not clarify if these results are normalized to the control group (no learning).

The reviewer is correct. Fig 1 demonstrates the cognitive impact of DDHD2 knockout, whilst Fig 2 demonstrates the impact of DDHD KO in the FFA response to instrumental conditioning. Our text has been revised to state that Fig 1 demonstrates that DDHD2 impacts on learning. The results are presented as a comparison, not normalised to the control.

- Based on Extended Fig 2 the authors claim that there are memory defects, however, it is not convincing that the performances of these mice in these tests is not due to motor impairment.

It is important to note that during the planning stages of this project, instrumental conditioning was specifically chosen as the preferred paradigm which was less sensitive to motor dysfunction than the fear conditioning which we had used in our previous study (Wallis *et al.*, 2021). The observation of no significant difference in the ambulatory distance of *DDHD2^{+/+}* and *DDHD2^{-/-}* mice suggests that the impairment in motor function was not severe enough to impact on the mice ability to press the lever, and hence the reduced lever presses can be attributed to authentic memory deficits as explained in page 19. Also, complementary data from Novel Object Location

testing confirms that the reduced memory performances of these mice is not due to motor impairment.

- Even though the authors have used a similar graphic representation of the data as in Wallis, 2021, for Fig 1C-F, the lines are superimposed, which makes them hard to interpret the underlying representation of each group. It would be more effective to show the data in columns, or alternatively heatmap, comparing each group, and mentioning which comparisons that are statistically significant and specify in the legend (and methods section) the statistical analysis.

Linked scatter plots were specifically chosen as they enable the viewer to determine at a glance both the absolute and fold change of the analytes in two conditions. In fact, in our previous publication, reviewers specifically asked us to present the data in a format other than bar plots. Accordingly, the data is now shown as stacked bar plots in Fig 2, linked scatter plots in Appendix Figure S1 and as heatmaps of significant change in Appendix Figure S2.

- Color assignment for each FFA appears to be somewhat random. A visualization suggestion could be to use Saturated FFAs vs Unsaturated FFAs with a more differentiating color code.

These colours have been chosen carefully so that they are clearly differentiated in the stacked bar plots. The colour code is also identical to our previous publication (Wallis *et al.*, 2021). The new Appendix Figure S8 now clearly differentiates saturated and unsaturated FFAs as dark and light grey respectively.

- In Extended Data Fig. 1E - which ones of the alterations are indeed significant? Authors should highlight only significant results and clarify statistical analysis employed.

We have now moved Appendix Figure S1E to form a stand alone Appendix Figure S2, which includes the statistical analysis used in the legend. Pixels are shown in white if the fold change was not significant (t test).

Figure3

- Since the authors bring up a possibility that a complex between STX1A, STXBP1 and DDHD2 might occur, it would be interesting to perform co-localization studies between these proteins (DDHD2-STXBP1 and DDHD2-STX1A) before and after stimulation. Based on the methods it appears the authors used Pearson co-efficient, but this should be clearer if it was indeed the case. As a complement or alternative for co-localization assessment, the authors can also use proximity-ligation assays.

The colocalization study we performed was criticised by Reviewr #1 point 4. We therefore concentrated our efforts on showing that STX1 and DDHD2 follow the same anterograde pathway. In PC12 cells, both DDHD2 and Syntaxin1A appear on the plasma membrane. In the absence of Munc18, in DKO-PC12 cells, both proteins are no longer localised to the plasma membrane. Their transport to the plasma membrane can however be rescued upon re-expression of STXBP1/Munc18-1. We have altered the analysis and representation of Fig 4 to make this point clearer in the result section and the new Fig 4. The fact that the levels of both Syntaxin1A and DDHD2 are much reduced has been attributed the trafficking block interference with protein expression (Kasula *et al.*, 2016; Martin *et al.*, 2013) (Han *et al.*, 2010; Han *et al.*, 2011; Malintan *et al.*, 2009) Rickman *et al.*, 2007). The co-localisation between Syntaxin1A and DDHD2 was expected but is of less importance in the context of the transport phenotype. We have therefore replaced this panel by the analysis of the DDHD2 transport defect and its rescue upon re-expression of STXBP-1 in DKD-PC12 cells. We hope our message is now clearer.

Figure4

- PC12 vs DKO non-stim is a condition both present in Fig. 4A and 4B - were these independent experiments? Statistical analysis should be done for Fig. 4A.

The data in 4A and 4B were acquired independently. The number of independent experiments has been added to the figure legends.

-In Fig4 - Re-expression of WT STXBP1 in these DKO PC12 cells completely rescued the FFA basal levels - it is interesting, but once again, what is then the role of STXBP2? Does it also rescue? Is it any different that STXBP1?

As we answered in the "major general points" above, we have now added the FFA profile for MKO cells in which only STXBP1 is ablated (new Figure 8). This data suggests that STXBP1 is primarily responsible for the maintenance of basal FFA levels.

- The way Fig 4D is presented is not clear - instead bar graphs should be used to understand exactly which comparisons are performed and the proper statistical analyses mentioned. We thank the reviewer for this observation. We have now presented this data as bar plots, in Fig S8. Statistical analyses were not performed as this figure is merely intended to show that saturated FFAs are predominant in the response.

Figure5

- What happens to the levels of DDHD2 and STXN1A in these brain regions?

We have now added western blot data quantifying the levels of DDHD2 and STBP1 in the cortex (Fig 9) and the dorsal hippocampus (S10)

- Is this a matter of regulation of DDHD2 activity or simply without the complex it is more degraded?

We have now added data in FigS7A and B which show that DDHD2 levels remain constant in the absence of STXBP1/2. This indicates that lower levels of DDHD2 are due to specific regulation.

- Once again, what happens to PA or other glycerophospholipids?

A new complete set of phospholipid data has now been added to DDHD2 (Fig3) and PLD1 (FigS4) as detailed in our earlier response.

- Figure 5C is hard to read with all lipids superimposed - it is interesting that one FFA is significantly increased - 24:0 this should be discussed. Also, various unsaturated and saturated FFAs are decreased - should this argue against a specific effect on saturated FFAs?

We have now included a new analysis of the percentage of saturated versus unsaturated FFA changes in STXBP1 heterozygote brain regions. This data is now in Appendix Figure S8 and clearly shows that STXBP1 majorly impact saturated FFAs and to a much lower extent unsaturated FFAs.

Minor Comments:

We thank the reviewer for the attention to detail in tracking down the following issues for correction:

- 24:0 is always in a bottom position and not grouped with the other saturated FFAs in the legend. This has now been corrected

- Extended Figs 1B and 1C should be mentioned in the text. This has now been corrected

- Extended Fig 3 - in the y axis - do the authors mean "tissue" instead of "cells"? This has now been corrected

- Fig 2B-C-D some graphs have data points but not others – this should be standardized.
Bar plot presentation and colours have now been standardised.

- Extended Fig 5 - it should be mentioned the names of the proteins from the top 10
This protein name data is included as part of Table 3

- Is the legend in panel D, Fig3, incorrect?
This has now been corrected

- No legend is shown with the different colors in Fig4
This has now been corrected

- In page 12, do the authors mean Fig. 4A instead of 5A?
This has now been corrected to 4A

- In Fig 5B and C should it be mentioning Het instead of KO?
This has now been corrected

- Fig5B appears to be unnecessary - particularly as a main figure.

This has now been corrected

References

- Han GA, Malintan NT, Collins BM, Meunier FA, Sugita S (2010) Munc18-1 as a key regulator of neurosecretion. *J Neurochem* 115: 1-10
- Han GA, Malintan NT, Saw NM, Li L, Han L, Meunier FA, Collins BM, Sugita S (2011) Munc18-1 domain-1 controls vesicle docking and secretion by interacting with syntaxin-1 and chaperoning it to the plasma membrane. *Mol Biol Cell* 22: 4134-4149
- Kasula R, Chai YJ, Bademosi AT, Harper CB, Gormal RS, Morrow IC, Hosy E, Collins BM, Choquet D, Papadopoulos A *et al* (2016) The Munc18-1 domain 3a hinge-loop controls syntaxin-1A nanodomain assembly and engagement with the SNARE complex during secretory vesicle priming. *Journal of Cell Biology* 214: 847-858
- Malintan NT, Nguyen TH, Han L, Latham CF, Osborne SL, Wen PJ, Lim SJ, Sugita S, Collins BM, Meunier FA (2009) Abrogating Munc18-1-SNARE complex interaction has limited impact on exocytosis in PC12 cells. *J Biol Chem* 284: 21637-21646
- Martin S, Tomatis VM, Papadopoulos A, Christie MP, Malintan NT, Gormal RS, Sugita S, Martin JL, Collins BM, Meunier FA (2013) The Munc18-1 domain 3a loop is essential for neuroexocytosis but not for syntaxin-1A transport to the plasma membrane. *J Cell Sci* 126: 2353-2360
- Santa-Marinha L, Castanho I, Silva RR, Bravo FV, Miranda AM, Meira T, Morais-Ribeiro R, Marques F, Xu Y, Point du Jour K *et al* (2020) Phospholipase D1 Ablation Disrupts Mouse Longitudinal Hippocampal Axis Organization and Functioning. *Cell Reports* 30: 4197-4208.e4196
- Wallis TP, Venkatesh BG, Narayana VK, Kvaskoff D, Ho A, Sullivan RK, Windels F, Sah P, Meunier FA (2021) Saturated free fatty acids and association with memory formation. *Nature Communications* 12: 3443

Dear Frederic,

Thank you for the submission of your revised manuscript to The EMBO Journal. We have now received the comments of the referees that were asked to re-evaluate your study (included below). As you will see, the referees appreciate the additional data and textual improvements, acknowledge that all previously raised concerns have been successfully addressed, and they now recommend publication. There is only one comment of referee #1 regarding the legend of Appendix Figure S7 that should be corrected before acceptance of the manuscript.

From the editorial side, there are also a few things that we need from you before we can proceed with publication of your manuscript:

- All co-corresponding authors are required to supply their ORCID IDs; please provide the ORCID ID of Dr. Wallis.
- Please enter all relevant funding information in our online manuscript handling system. It should match exactly the information provided in the Acknowledgements section of your manuscript.
- Please note that all primary datasets produced in the study need to be deposited in appropriate public databases and listed in the "Data availability" section of the manuscript. We would kindly ask you to provide public access to the mass spectrometry datasets that were produced in this study and include their access information in your "Data availability" section, following the examples/model:
- RNA-seq data: Gene Expression Omnibus GSE46843 (<https://www.ncbi.nlm.nih.gov/geo/query/acc.cgi?acc=GSE46843>)
- [data type]: [name of the resource] [accession number/identifier/doi] ([URL or identifiers.org/DATABASE:ACCESSION])
- Please change the heading of your competing interests statement to "Disclosure and competing interests statement".
- The author contributions statement should be removed from the manuscript. Instead, we now use CRediT to specify the contributions of each author in the journal submission system (you can use the free text box to provide more detailed descriptions). See also our guide to authors: <https://www.embopress.org/page/journal/14602075/authorguide#authorshipguidelines>
- Figure callouts for Appendix Figure S4 are missing; please note that all figures and their panels should be called out in your revised manuscript (in alphabetical order).
- Callouts for Appendix Figure 1A-B should be corrected to Appendix Figure S1A-B.
- Appendix tables should be called out as Appendix Table S1-S3 instead of Table S1-S3.
- Supplementary Tables 4 and 5 are called out, but no such tables are uploaded.
- Main and EV Figures should be uploaded individually in high resolution. See also our guide to authors: <https://www.embopress.org/page/journal/14602075/authorguide#figureformat>
- A brief Table of Contents, including page numbers, should be provided on the first page of the Appendix.
- The synopsis image should be uploaded individually (in JPG or PNG format). Please note that the final dimensions should be 550 x 300-600 pixels (width x height) and all text should be easily readable when the image is resized to these dimensions.
- Please define the annotated p values ***/**/* in the legends of figures 1c, e; 8b; EV3c as appropriate.
- Please indicate the statistical test used for data analysis in the legends of figures 1c, e; 3j-k; 6c, e; EV3c.
- Please note that information related to the sample size (n) and the nature of the replicates (e.g. biological or technical replicates) is missing in the legends of figures 1c, e; 2b; 5r; 6c, e; 8a, c-d; 9c; EV3c.
- Please note that the error bars are not defined in the legends of figures 1b-e; 5r; 6c, e; 7c, e; EV3c.
- In each Figure legend, information related to data representation, statistical tests, statistical significance (p values) etc. should be listed in a "Data information" section at the end of the legend. Please see also our authors' guide: <https://www.embopress.org/page/journal/14602075/authorguide#figureformat>

Please also note that as part of the EMBO publications' Transparent Editorial Process, The EMBO Journal publishes online a Peer Review File along with each accepted manuscript. This File will be published in conjunction with your paper and will include the referee reports, your point-by-point response and all pertinent correspondence relating to the manuscript. You can opt out of this by letting the editorial office know (contact@embojournal.org). If you do opt out, the Peer Review File link will point to the following statement: "No Peer Review File is available with this article, as the authors have chosen not to make the review process public in this case."

We look forward to seeing a final version of your manuscript as soon as possible. Please use this link to submit your revision:
<https://emboj.msubmit.net/cgi-bin/main.plex>

Best regards,

Ioannis

Referee #1:

The authors have responded in detail to all my comments and concerns with dedicated experiments, careful discussion, and editorial modifications. There is, however, one more mistake the authors must fix: the legend to Appendix Figure S7 does not match well with the labeled panels in the figure. Otherwise, the manuscript has been improved significantly and can be accepted for publication.

Regarding authors' response to Reviewer #2's remarks, I have gone through the rebuttal point-by-point. The authors also made great efforts to address every question or suggestion with either explanation or experimentation. I am particularly impressed by their effort to demonstrate localization of DDHD2 to synapses by high-resolution fluorescence microscopy, and the involvement of DDHD2 in intracellular trafficking by electron microscopy in new Figure 7. Overall, I believe the authors have responded adequately to Reviewer #2's comments and concerns.

Referee #3:

The authors have addressed the comments previously raised in a satisfactory manner.

Point-by-point response to reviewers' comments

We thank the reviewers and editorial team for their valuable comments and suggestions. We have now revised the manuscript and implemented the recommendations accordingly.

There is only one comment of referee #1 regarding the legend of Appendix Figure S7 that should be corrected before acceptance of the manuscript.

Response: The legend for Appendix Figure S7 has now been revised accordingly.

From the editorial side, there are also a few things that we need from you before we can proceed with publication of your manuscript:

- All co-corresponding authors are required to supply their ORCID IDs; please provide the ORCID ID of Dr. Wallis.

Response: The ORCID ID of Dr. Wallis has now been provided.

- Please enter all relevant funding information in our online manuscript handling system. It should match exactly the information provided in the Acknowledgements section of your manuscript.

Response: All relevant funding information has been updated on the online manuscript handling system.

- Please note that all primary datasets produced in the study need to be deposited in appropriate public databases and listed in the "Data availability" section of the manuscript. We would kindly ask you to provide public access to the mass spectrometry datasets that were produced in this study and include their access information in your "Data availability" section, following the examples/model:

- RNA-seq data: Gene Expression Omnibus GSE46843

(<https://www.ncbi.nlm.nih.gov/geo/query/acc.cgi?acc=GSE46843>)

- [data type]: [name of the resource] [accession number/identifier/doi] ([URL or identifiers.org/DATABASE:ACCESSION])

Response: The data produced in this study is publicly available in the following databases:

- Imaging dataset: BioImage Archive S-BIAD910

(<https://www.ebi.ac.uk/biostudies/bioimages/studies/S-BIAD910>).

- Behavioural and quantified lipid abundance data: University of Queensland eSpace Data Collection (<https://doi.org/10.48610/4a44503>)

- Please change the heading of your competing interests statement to "Disclosure and competing interests statement".

Response: The heading of your competing interests statement has now been changed to "Disclosure and competing interests statement".

- The author contributions statement should be removed from the manuscript. Instead, we now use CRediT to specify the contributions of each author in the journal submission system (you can use the free text box to provide more detailed descriptions). See also our guide to authors:
<https://www.embopress.org/page/journal/14602075/authorguide#authorshipguidelines>

Response: The author contributions statement has now been removed from the manuscript and included using the online submission system.

- Figure callouts for Appendix Figure S4 are missing; please note that all figures and their panels should be called out in your revised manuscript (in alphabetical order).

Response: The figure callouts for Appendix Figure S4 have been included in the manuscript.

- Callouts for Appendix Figure 1A-B should be corrected to Appendix Figure S1A-B.

Response: The figure callouts for Appendix Figure 1A-B have been corrected to Appendix Figure S1A-B.

- Appendix tables should be called out as Appendix Table S1-S3 instead of Table S1-S3.

Response: Appendix tables call out have been changed to Table S1-S3.

- Supplementary Tables 4 and 5 are called out, but no such tables are uploaded.

Response: Supplementary Tables 4 and 5 call out has been removed from the manuscript.

- Main and EV Figures should be uploaded individually in high resolution. See also our guide to authors:

<https://www.embopress.org/page/journal/14602075/authorguide#figureformat>

Response: Main and EV Figures have now been uploaded individually in high resolution.

- A brief Table of Contents, including page numbers, should be provided on the first page of the Appendix.

Response: As requested, a brief Table of Contents, including page numbers, has now been provided on the first page of the Appendix.

- The synopsis image should be uploaded individually (in JPG or PNG format). Please note that the final dimensions should be 550 x 300-600 pixels (width x height) and all text should be easily readable when the image is resized to these

dimensions.

Response: The synopsis image has been provided as requested.

- Please define the annotated p values *****/**/*** in the legends of figures 1c, e; 8b; EV3c as appropriate.

Response: The annotated p values *****/**/*** in the legends of figures 1c, e; 8b; EV3c have now been appropriately defined.

- Please indicate the statistical test used for data analysis in the legends of figures 1c, e; 3j-k; 6c, e; EV3c.

Response: The statistical test used for data analysis has now been defined in the legends of the specified figures.

- Please note that information related to the sample size (n) and the nature of the replicates (e.g. biological or technical replicates) is missing in the legends of figures 1c, e; 2b; 5r; 6c, e; 8a, c-d; 9c; EV3c.

Response: The information related to the sample size (n) and the nature of the replicates have now been defined in the legends of the specified figures.

Fig 1

Data information: In (C, E), the significance of the difference between each group (n=20 in each cohort of animals) as determined by unpaired t-test is indicated by asterisks ****** p<0.01, ******* p<0.001. Error bars represent the cumulative standard error of the mean (SEM) for all groups and parameters.

Fig 2

Data information: In (B), the significance of the difference between each group (n = 6 biological replicates) as determined by unpaired t-test with Holm-Sidak post hoc correction is indicated by asterisks ****** p<0.01, ******* p<0.001, ns = not significant. Error bars represent the cumulative standard error of the mean (SEM) for all groups and parameters.

Fig 3

Data information: In (A-I), each pixel in the heatmap represents the average change in total abundance of all lipids of a given class across the 6 measured brain regions (n = 6 biological replicates). In (J-K), each dot on the volcano plot represents the average change in abundance of a single lipid analyte across 6 measured brain regions in control vs instrumentally conditioned DDHD2+/+ vs DDHD2-/- mice. Analytes below the red line represent those whose change in abundance was not statistically significant (two-tailed t-test p > 0.05).

Fig 4

Data information: In (A), the thickness and hatching of the lines connecting each protein represents the confidence score of the strength of the interaction as determined by the STRING knowledge base matching algorithm. In (B-D), the significance of the change between the different experimental conditions as

determined by two-tailed unpaired Student's t-test, * $p < 0.05$, *** $p < 0.001$. The significance of the change between the different experimental conditions as determined by two-tailed unpaired student's t-test.

Fig 5

Data information: In (R), the significance is tested using ordinary one-way ANOVA multiple comparison test. Scale bars are 5 μm . $n = 10-22$ cells from 3 independent experiments and error bars represent the cumulative standard error of the mean (SEM) for all groups and parameters

Fig 6

Data information: In (A), statistical testing of non-normally distributed data was done using Mann-Whitney U test. $n = 30-53$ ROIs from 3 independent experiments, In (C, E), the significance of the difference between each group as determined by unpaired t-test is indicated as < 0.0001 , and the error bars represent the cumulative standard error of the mean (SEM) for all groups and parameters. $n = 30-55$ cells from 3 independent experiments.

Fig 7

Data information: In (C, E), $n = 5$ acquisitions per condition from 2 independent experiments in (C), with 94 and 147 presynapses quantified from C57BL/6J and DDHD2^{-/-}, respectively from 3 independent experiments in E. Statistical testing of normally distributed data was done using Student's t-test.

Fig 8

Data information: In (A-D), the significance of the change in FFA abundance between the different experimental conditions ($n = 3$ biological replicates) as determined by one-way ANOVA with Holm-Sidak post-hoc correction is indicated by asterisks * $p < 0.05$, ** $p < 0.01$, ns = not significant. Error bars represent the cumulative standard error of the mean (SEM) for all groups and parameters.

Fig 9

Data information: In (B), values are presented as mean \pm SEM. Student's t-test was used to compare protein expression from WT to STXBP1^{-/+} brains, $n=4$ mice each. In (C), the significance of the change in FFA abundance between the different experimental conditions ($n = 5$ biological replicates) as determined by one-way ANOVA with Holm-Sidak post-hoc correction is indicated by asterisks, **** $p < 0.0001$. In (E), each dot on the volcano plot represents the average change in abundance of a single analyte across 5 measured brain regions. Analytes below the red line represent those whose change in abundance was not statistically significant (two-tailed t-test $p > 0.05$). Error bars represent the cumulative standard error of the mean (SEM) for all groups and parameters.

Fig EV1

Data information: In (A-H), the significance of the difference between each group as determined by 1-way ANOVA ($n=20$) is indicated by asterisks * $p < 0.05$, ** $p < 0.01$, *** $p < 0.001$, ns = not significant. Error bars represent the cumulative standard error of the mean (SEM) for all groups and parameters.

Fig EV3

Data information: In (C), the significance of the difference between each group (n = 3 biological replicates) as determined by unpaired t-test with Holm-Sidak post hoc correction is indicated by asterisks ** p<0.01, ns = not significant. Error bars represent the cumulative standard error of the mean (SEM) for all groups and parameters.

Appendix Fig S3

Data information: In (B), two-way ANOVA: WT/KO 0h: *** p<0.0001; WT/KO context A 0h: ** p=0.0014; WT/KO context B 24hrs: ns = Not significant p= 0.3533, Sidak's multiple comparison test. n = 5 independent brain samples.

Appendix Fig S6

Data information: In (B, D, E), statistical analysis was performed using Student's t-test. All data are represented as mean ± SEM from 3 independent experiments. One-way ANOVA with Tukey's correction for multiple comparisons, *p<0.05, **p<0.01, ns = not significant.

Appendix Fig S7

Data information: In (S7), the significance of the change in average FFA abundance between PC12 Non-stim/Stim, MKO Non-stim/Stim, and DKO Non-stim/Stim as determined by t-test with Holm-Sidak post-hoc correction is indicated by asterisks * p<0.01. ns = not significant. The error bars represent the cumulative standard error of the mean (SEM). n = 3 biological replicates.

Appendix Fig S7

Data information: In (B), Values are presented as mean ± SEM. Student's t-test was used to compare protein expression from WT (orange bars) vs STXBP-1-/+ brains (blue bars). n = 4 mice for each condition.

- Please note that the error bars are not defined in the legends of figures 1b-e; 5r; 6c, e; 7c, e; EV3c.

Response: The error bars have now been defined in the legends of the specified figures.

- In each Figure legend, information related to data representation, statistical tests, statistical significance (p values) etc. should be listed in a "Data information" section at the end of the legend. Please see also our authors' guide:

<https://www.embopress.org/page/journal/14602075/authorguide#figureformat>

Response: The "Data information" section has now been included at the end of the legends. Please see detailed response above

Please also note that as part of the EMBO publications' Transparent Editorial Process, The EMBO Journal publishes online a Peer Review File along with each accepted manuscript. This File will be published in conjunction with your paper and will include the referee reports, your point-by-point response and all pertinent correspondence relating to the manuscript. You can opt out of this by letting the

editorial office know (contact@embojournal.org). If you do opt out, the Peer Review File link will point to the following statement: "No Peer Review File is available with this article, as the authors have chosen not to make the review process public in this case."

Referee #1:

The authors have responded in detail to all my comments and concerns with dedicated experiments, careful discussion, and editorial modifications. There is, however, one more mistake the authors must fix: the legend to Appendix Figure S7 does not match well with the labeled panels in the figure. Otherwise, the manuscript has been improved significantly and can be accepted for publication.

Response: The legend for Appendix Figure S7 has now been revised accordingly. Appendix Figure S7. Profile of FFAs in PC12 cells, STXBP1 single knockout (MKO) PC12 cells, and STXBP1/2 double knockout (DKO) PC12 cells following stimulation. PC12, MKO, and DKO cells were stimulated (Stim) by depolarization for 15 min in high K⁺ (60 mM) buffer. Control unstimulated (Non-stim) cells were treated for 15 min in low K⁺ (2 mM) buffer.

Data information: In (S7), the significance of the change in average FFA abundance between PC12 Non-stim/Stim, MKO Non-stim/Stim, and DKO Non-stim/Stim as determined by t-test with Holm-Sidak post-hoc correction is indicated by asterisks * $p < 0.01$. ns = not significant. The error bars represent the cumulative standard error of the mean (SEM). $n = 3$ biological replicates.

Referee #2:

Regarding authors' response to Reviewer #2's remarks, I have gone through the rebuttal point-by-point. The authors also made great efforts to address every question or suggestion with either explanation or experimentation. I am particularly impressed by their effort to demonstrate localization of DDHD2 to synapses by high-resolution fluorescence microscopy, and the involvement of DDHD2 in intracellular trafficking by electron microscopy in new Figure 7. Overall, I believe the authors have responded adequately to Reviewer #2's comments and concerns.

Referee #3:

The authors have addressed the comments previously raised in a satisfactory manner.

Dear Fred,

I am very pleased to inform you that your manuscript has been accepted for publication in the EMBO Journal.

Best regards,

Ioannis
